# Interplay of cell dynamics and epithelial tension during morphogenesis of the *Drosophila* pupal wing

**Raphaël Etournay[1†], Marko Popović[2†], Matthias Merkel[2†], Amitabha Nandi[2†], Corinna Blasse[1], Benoît Aigouy[3], Holger Brandl[1], Gene Myers[1], Guillaume Salbreux[2,4]\*, Frank Jülicher[2]\*, Suzanne Eaton[1]\***

[1]Max Planck Institute of Molecular Cell Biology and Genetics, Dresden, Germany; [2]Max Planck Institute for the Physics of Complex Systems, Dresden, Germany; [3]Institut de Biologie du Développement de Marseille, Marseille, France; [4]Lincoln's Inn Fields Laboratories, The Francis Crick Institute, London, United Kingdom

**Abstract** How tissue shape emerges from the collective mechanical properties and behavior of individual cells is not understood. We combine experiment and theory to study this problem in the developing wing epithelium of *Drosophila*. At pupal stages, the wing-hinge contraction contributes to anisotropic tissue flows that reshape the wing blade. Here, we quantitatively account for this wing-blade shape change on the basis of cell divisions, cell rearrangements and cell shape changes. We show that cells both generate and respond to epithelial stresses during this process, and that the nature of this interplay specifies the pattern of junctional network remodeling that changes wing shape. We show that patterned constraints exerted on the tissue by the extracellular matrix are key to force the tissue into the right shape. We present a continuum mechanical model that quantitatively describes the relationship between epithelial stresses and cell dynamics, and how their interplay reshapes the wing.

**\*For correspondence:** salbreux@pks.mpg.de (GS); julicher@pks.mpg.de (FJ); eaton@mpi-cbg.de (SE)

†These authors contributed equally to this work

## Introduction

The dynamic choreography of tissue shape changes that occur during development dramatically illustrates the fact that morphogenesis depends on organized cellular force generation. The mechanisms that control the orientation and patterning of these active processes and the corresponding tissue stresses are beginning to be explored in a variety of developmental systems, for review (*Lecuit and Lenne, 2007*; *Keller, 2012*; *Heisenberg and Bellaiche, 2013*). However, a complete understanding of the mechanical basis of morphogenesis will require not only a description of cell autonomously generated forces, but also quantitative insights into how cells respond to tissue stresses. Cells exert forces on extracellular matrices, but also on each other—this is particularly true of epithelial cells, which are tightly connected by specialized adhesive junctions. Thus, stresses generated by one epithelial cell can be transmitted to others throughout the tissue. In vitro experiments have shown that tissues respond to stress elastically over short time scales but that they can plastically remodel when subjected to stress over longer times (*Beloussov et al., 2000*; *Harris et al., 2012*). This can occur as a result of cell shape changes, cell rearrangements or both, and appears to vary with the cell types examined. Furthermore, experiments with cultured epithelial cells suggest that tissue compression can limit cell proliferation in vitro (*Puliafito et al., 2012*; *Streichan et al., 2014*). How these cellular responses might influence tissue size and shape in vivo is not clear. Nevertheless, these in vitro observations suggest that a complete and quantitative understanding of tissue morphogenesis will require new insights into tissue viscoelasticity in vivo and the cellular mechanisms that give rise to it.

**eLife digest** The individual cells in a developing animal embryo organize themselves into tissues with specific and reproducible shapes, which requires the cells to communicate with one another. Cells in tissues exert forces on their neighbors, and respond to being pushed and pulled by the cells around them.

In the fruit fly *Drosophila melanogaster*, each wing consists mainly of a framework of proteins and other molecules that is built by epithelial cells. These epithelial cells divide and grow during the life of a fly larva, and then reorganize themselves into the shape of the wing after it forms into a pupa. During this reshaping, epithelial cells in some regions of the wing experience powerful contractions. Previous work had suggested that and these forces produced tension in the rest of the wing to pull it into its final elongated shape. But it wasn't clear what exactly these contractions were pulling against to produce the tension. Nor was it understood exactly how wing epithelial cells responded to tension to reorganize themselves into a different wing shape.

Now, Etournay, Popović, Merkel, Nandi et al. have analyzed the forces acting across the entire wing blade and how these forces shape the wing. All cell divisions, cell neighbor exchanges and changes in cell shape in the developing wing blade were tracked under a microscope; this revealed how each one of them contributed to the change in wing shape. Further experiments revealed that localized contractile forces produce tension in the wing because it is connected around its edge to surrounding structures via an extracellular protein called Dumpy. Releasing these contacts, by severing them with a laser or by mutating Dumpy, caused the wing to develop into abnormal shapes, showing that the tension in the wing blade has an important role in determining wing shape. Furthermore, by tracking cells in wings that had been severed by a laser, or mutated for Dumpy Etournay, Popović, Merkel, Nandi et al. could figure out exactly which cellular processes were guided by epithelial tension.

Etournay Popović, Merkel, Nandi et al. also present a theoretical model that describes how the interplay between active force generation and the response of cells to the resulting tension shapes the wings of fruit flies. They propose that epithelial tension provides a mechanism through which cells can communicate with each other to ensure that together the combined behavior of these cells generates reproducible shapes. Further studies are required to analyze how active force generation is patterned and cells sense and respond to external forces during development.

*Drosophila* pupal wing morphogenesis is an ideal system in which to study the interplay of cellular force generation and tissue material properties in vivo. During pupal stages, anisotropic stresses along the proximal-distal (PD) axis of the wing blade epithelium help guide anisotropic tissue flows that reshape the blade—elongating it in the PD axis and narrowing it in the anterior-posterior (AP) axis, for review (*Eaton and Julicher, 2011*). The mechanisms that produce PD-oriented stresses in the wing blade are not fully understood. They are generated in part by contraction of cells in the wing hinge, which connects to the wing blade on its proximal side. However, we do not understand the origin of counterforces that restrain movement of the wing blade at the margin.

Analyzing cells in a subregion of the wing blade showed that tissue flows are associated with cell shape changes, cell divisions and cell rearrangements that are oriented along the PD axis (*Aigouy et al., 2010*). To quantitatively understand the cellular basis of this tissue shape change, we must determine the global patterns of these cellular events throughout the wing blade. Furthermore, while hinge contraction contributes to PD tissue stresses in the blade, cells in the wing blade might also contribute autonomously to tissue flows and stresses. Thus, to understand the mechanical basis of pupal wing morphogenesis, we must understand the emergence of PD-oriented stresses in the wing blade, and distinguish stresses autonomously generated by wing epithelial cells from the response of epithelial cells to these stresses.

Here, we combine several quantitative methods to investigate how cell flows and global tissue shape changes emerge from the collective behavior and mechanical properties of many wing epithelial cells. We develop image analysis methods to track the majority of cells in the wing throughout morphogenesis, and analyze cell shapes and rearrangements of the junctional network. Furthermore, we develop theoretical methods to quantify the cellular contributions to tissue shear and area homeostasis in the wing blade.

We show that localized apical extracellular matrix connections to the cuticle at the wing margin provide the counterforce to hinge contraction, and are required for the development of normal stresses in the wing blade. These stresses are essential to reshape the pupal wing while maintaining wing area homeostasis. We distinguish autonomously controlled from stress-driven cellular events, and present a continuum mechanical model that quantitatively explains wing shape changes on the basis of the relationship between tissue stress, cell elongation and cell rearrangements.

## Results

### Dumpy-dependent physical constraints at the margin maintain epithelial tension in the wing

The emergence of two-dimensional stresses in the plane of the wing blade suggests that there are physical constraints on the movement of wing epithelial cells near the margin. We wondered whether there might be a matrix connecting the wing epithelium to the overlying pupal cuticle in this region. To investigate this, we used a laser to destroy the region between the margin of the E-Cadherin:GFP expressing wing epithelium and the cuticle after the two had separated as a consequence of molting. Although this treatment does not apparently damage either the wing or the cuticle, it causes the wing epithelium to rapidly retract away from the cuticle within seconds (*Figure 1A–B′′*, *Video 1*). Laser ablation causes epithelial retraction when performed at any region along the wing blade margin—anteriorly, posteriorly or distally. During tissue flows, the now disconnected margin moves even further away from the cuticle, producing abnormal wing shapes (*Figure 1C–F*). This shows that the wing is physically restrained by apical extracellular matrix connections to the overlying cuticle, and that these connections are required to shape the wing during tissue flows.

We wondered whether the large apical extracellular matrix protein Dumpy might contribute to these connections. Dumpy is a 2.5 MDa protein that is predicted to form filaments at least 1 μm long (*Wilkin et al., 2000*). It forms an elastic matrix in the embryonic tracheal lumen, and provides mechanical resilience of tendon cell attachments to the overlying cuticle (*Dong et al., 2014*). While *dumpy* null mutations are lethal, some hypomorphs produce wings that are short and misshapen—a defect that arises during pupal development (*Waddington, 1939*, *1940*). To ask whether shape defects in *dumpy* wings might arise during pupal tissue flows, we imaged *dumpy^{ov1}* pupal wings that expressed E-Cadherin:GFP. The shape of *dumpy^{ov1}* wings is normal at 16 hr after puparium formation (APF), before molting occurs (*Figure 2A,B*). Shortly afterwards, when hinge contraction begins, the shape of the *dumpy^{ov1}* mutant wing blade begins to differ from wild type (WT). The wing blade epithelium retracts abnormally far from the distal cuticle and fails to elongate in the PD axis. By the time tissue flows have ended, the characteristic abnormal shape of the *dumpy^{ov1}* wing is apparent (*Video 2* and *Figure 2A–B′′*).

To examine Dumpy distribution, we imaged wings from flies harboring a protein trap construct that expresses YFP:Dumpy from the endogenous chromosomal locus. YFP:Dumpy is present on the apical surface of epithelial cells throughout the wing, and within the overlying cuticle (*Figure 2—figure supplement 1A*, *Video 3*). Interestingly, Dumpy is also present in a fibrous-appearing matrix that connects the wing to the overlying cuticle in specific places. This matrix lies between the cuticle and the margin of the wing (*Figure 2C–E,H*), as well as in stripes that run on the dorsal surface of the wing between longitudinal veins L3 and L4, and over veins L2 and L5 (*Figure 2F,G*, *Figure 2—figure supplement 1B*). Dumpy-containing matrix also connects a subregion of the wing hinge to the overlying cuticle (*Figure 2F,G*, *Figure 2—figure supplement 1B*, *Video 3*).

To investigate the extent to which wing margin constraints had been relieved by the *dumpy^{ov1}* mutation, we performed laser-severing experiments in the *dumpy^{ov1}* mutant background. Cutting a *dumpy^{ov1}* wing between the cuticle and the distal wing blade revealed almost undetectable retraction, suggesting that distal attachments of the wing blade to the cuticle are severely compromised. However retraction was still observed when severing was performed between the cuticle and the anterior or posterior margins (*Figure 2—figure supplement 2*). Furthermore, severing the matrix around the entire margin causes *dumpy^{ov1}* mutant wings to develop even more dramatic wing shape abnormalities during tissue flows (compare *Figure 2—figure supplement 2G* to *Figure 2A′′*). This suggests that apical matrix connections to the cuticle are not completely abrogated in *dumpy^{ov1}* wings.

To ask how the *dumpy^{ov1}* mutation influenced PD tension in the wing blade, we performed circular laser cuts covering about 5–10 cells in different regions of WT and *dumpy^{ov1}* wings

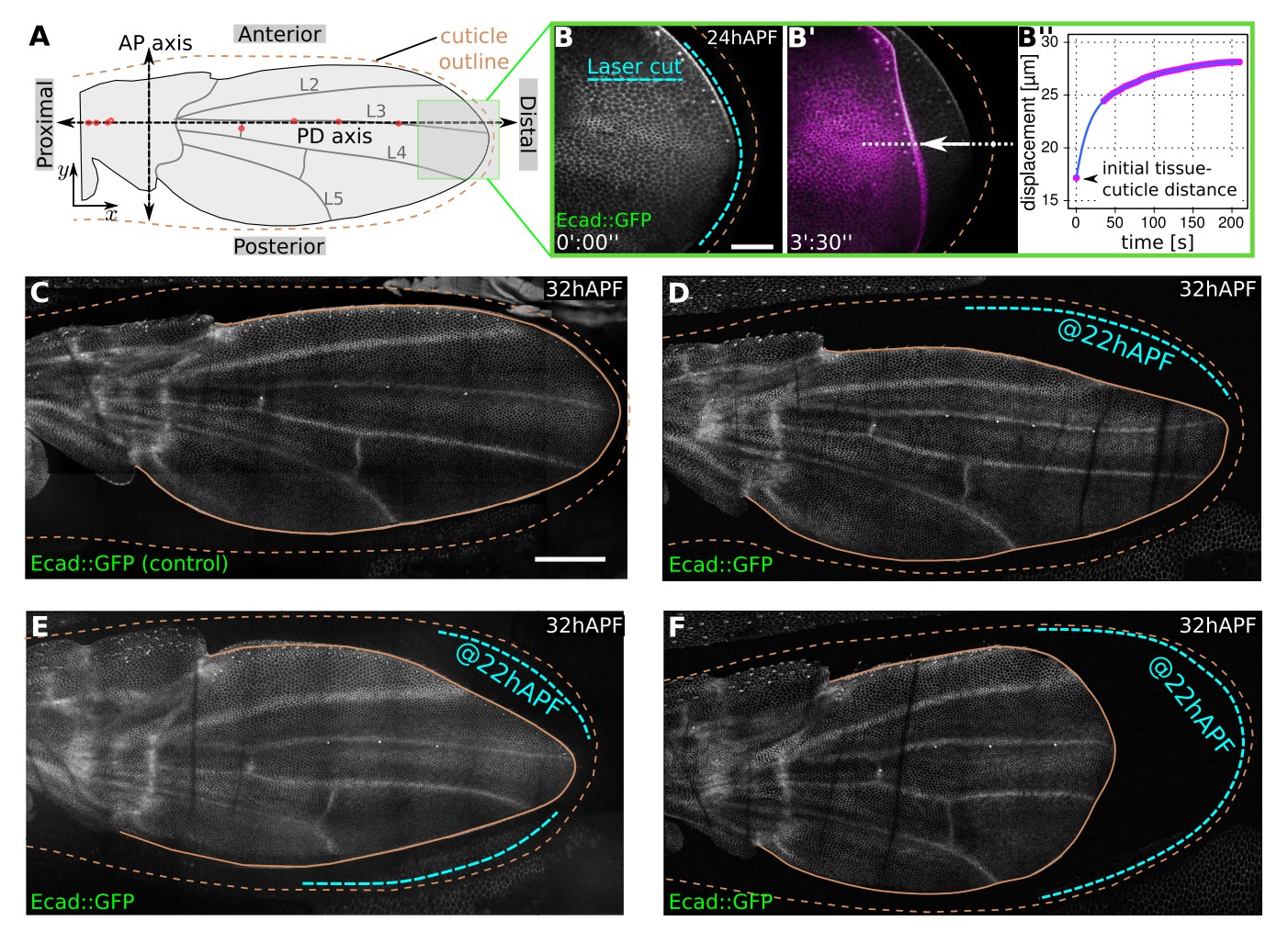

**Figure 1**. Physical constraints at the margin maintain epithelial tension in the wing. (**A**) Cartoon depicting a pupal wing at 32 hAPF. Dashed double-sided arrows depict the proximal-distal (PD) and anterior-posterior (AP) axes. The PD axis is defined by a regression line passing through selected sensory organs (red dots) that are easily identifiable in Ecad::GFP expressing wings. The x axis is defined to correspond to the PD axis pointing distally, and the y axis is defined to correspond to the AP axis pointing anteriorly. L2–L5 indicate longitudinal veins 2–5. Brown dashed line outlines the cuticular sac surrounding the wing epithelium. Scale bar 20 μm. (**B**, **B'**) Show the distal end of a wild-type (WT) Ecad::GFP-expressing wing at 24 hAPF (greyscale in **B**, **B'**) and the same wing 3.5 min after laser ablation in the space between wing margin and cuticle (magenta in **B'**). The blue dashed line indicates the site of laser ablation. (**B''**) Shows wing margin displacement measured with respect to the cuticle (brown dashed line in **B'**) along the white dotted line in (**B'**). Experimental points (magenta) were interpolated by a polynomial (blue line). (**C**–**F**) Show 32 hAPF wings that were unperturbed (**C**) or subjected to laser ablation at 22 hAPF (**D**–**F**). Ablation of the connections between the wing margin and the cuticle were performed in different regions, indicated by blue dashed lines in (**D**–**F**), and lead to altered wing shapes at 32 hAPF compared to the unperturbed control (**C**). Scale bar 100 μm.

(*Figure 2I,I'*, *Figure 2—figure supplement 3*). We observed a recoil of the ablated region, indicating that the blade epithelium is under tension. From the recoil, we can compare both the isotropic and the anisotropic components of epithelial stress in WT and *dumpy^{ov1}* mutant wings (*Figure 2I,I'* and *Figure 2—figure supplement 3*). These stress patterns differ between WT and *dumpy^{ov1}* wings. The orientation of anisotropic stress in *dumpy^{ov1}* is somewhat splayed and not as well aligned with the PD axis. Furthermore, anisotropic tension in *dumpy^{ov1}* wings tends to be reduced in the central region and increased anteriorly and posteriorly.

Overall, Dumpy-dependent elastic connections are key to the emergence of the stress pattern during morphogenesis. This suggests that these stresses play an important role in guiding tissue flows.

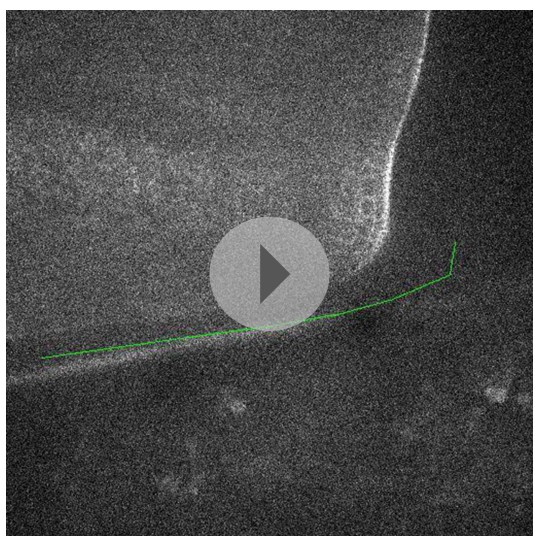

**Video 1.** Laser ablations of the apical extracellular matrix present in the space between the tissue and the cuticle. Green lines indicate the site of ablation right before ablation.

## Quantifying wing morphogenesis

What are the cellular events that shape the wing blade during tissue flows? To quantitatively address this question, we developed methods to quantify cell shape changes, cell divisions, cell rearrangements and cell extrusions during wing morphogenesis. We imaged three E-Cadherin:GFP-expressing wings at cellular resolution every 5 min between 16 and 32 hr APF. We then extracted and projected the planes containing the apical adherens junctions, automatically detected cell edges, and tracked each cell in the wing over the course of the videos (*Figure 3A,A′* and *Videos 4, 5*). We designed a relational database (DB) to store information pertaining to all cells in a given video ('Materials and methods', Long-terms time-lapse imaging and Data handling and image processing). Querying these DBs provides information about individual cellular properties such as shape, area, and associated cell boundaries. It also provides information about neighbor and lineage relationships, identifying neighbor exchanges (T1 transitions), cell divisions and cell extrusions (T2 transitions).

## Cellular contributions to wing area changes

We showed previously that the area of the wing blade remains fairly constant during pupal morphogenesis, despite the fact that cells are dividing (*Aigouy et al., 2010*). To understand how the wing blade maintains area homeostasis, we quantified wing blade area, cell divisions, cell area changes and cell extrusions between 16 and 32 hAPF in three different WT wings (*Figure 3A–E*). We analyzed a large region of the wing blade in which all cells could be tracked during the whole video (green in *Figure 3A,C*).

The tissue area expansion rate is $v = \frac{1}{A}\frac{dA}{dt}$, where A is the tissue area and d/dt denotes the total time derivative. The tissue expansion rate can be decomposed into its cellular contributions as

$$v = \frac{1}{a}\frac{da}{dt} + k_d - k_e, \tag{1}$$

where $a$ is average cell area, $k_d$ and $k_e$ are cell division and extrusion rates, respectively. The cumulative area expansion rate of the whole wing blade is $\ln(A/A_0)$, where $A_0$ is the tissue area when recording starts (~16 hAPF), can be obtained by integrating the area expansion rate $v$ over time.

*Figure 3D* shows both the tissue area expansion rate (dark blue) and the contributions to this expansion rate from cell area changes, cell divisions and extrusions. These are averages over three WT wings. *Figure 3E* shows the corresponding cumulative quantities. The dynamics of wing area changes in the 3 WT wing blades are extremely similar—after contracting slightly during the first half of morphogenesis, blade area gradually returns to very close to its original value (dark blue lines in *Figure 3D,E*). This almost constant area reflects a balance between cell divisions (orange) on the one hand, and cell area changes (green) and extrusions (light blue) on the other. Interestingly, blade area in three analyzed wings is more reproducible than would be expected from the variation in each cellular contribution, if they were independent of each other (see shaded regions depicting standard deviations in *Figure 3D,E*). To quantify this observation we compared the variance of overall relative area change with the sum of the variances of the cumulative cellular contributions. We find that sum of variances is about 20 times larger than variance of the sum. This shows that cellular contributions are not independent and that normal variations in the rate of cell division can be compensated by changes in cell area and/or extrusion to maintain wing blade area (*Figure 3—figure supplement 1*).

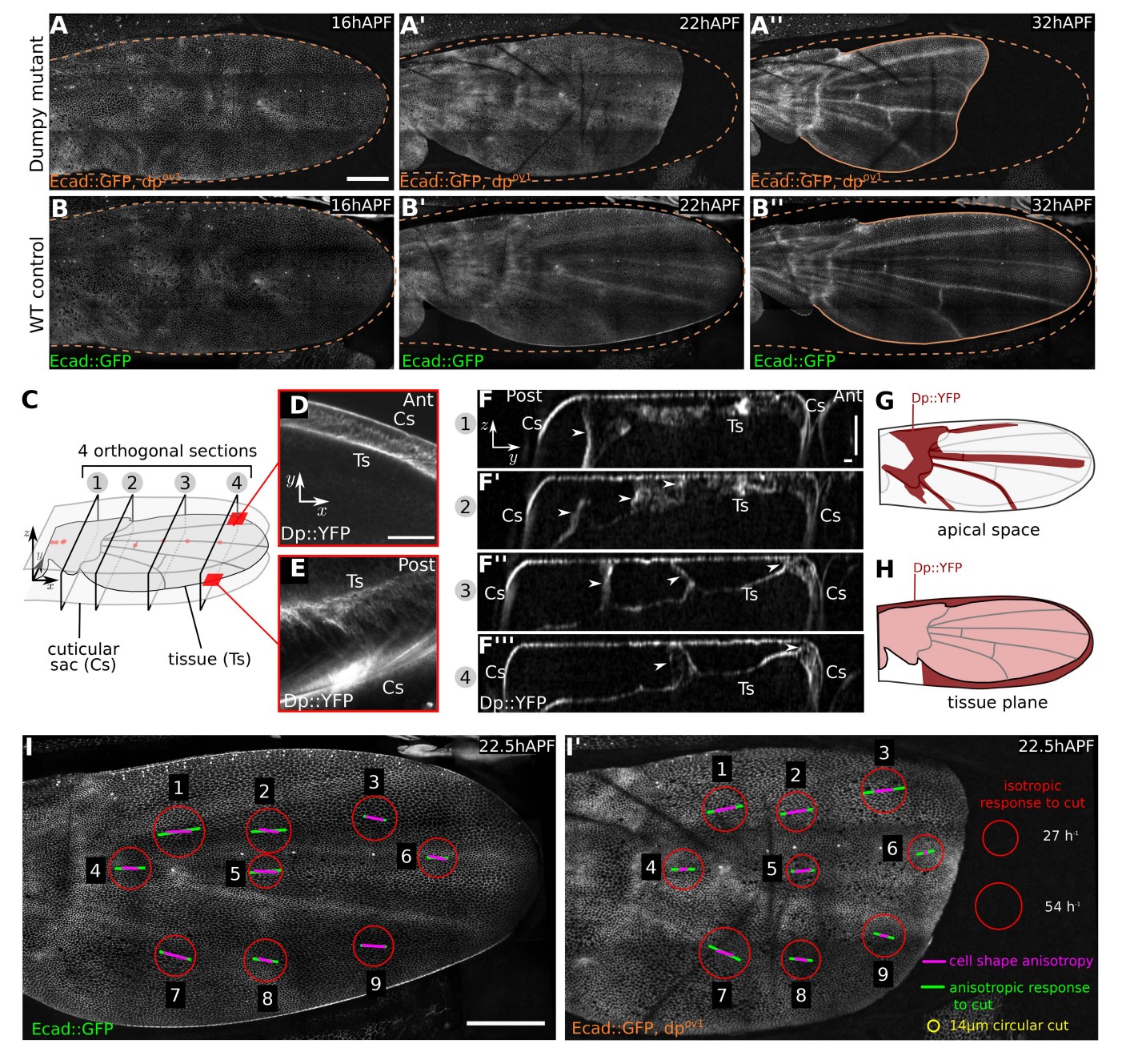

**Figure 2**. Dumpy-dependent apical attachments of wing tissue to the cuticle act as a counter-force to hinge contraction. (**A–B''**) Show individual frames from a time-lapse video of *dumpy^ov1* mutant and control WT wings expressing Ecad::GFP, and depict wings at 16 hAPF (**A**, **B**), 22 hAPF (**A'**, **B'**), and 32 hAPF (**A''**, **B''**). The position of the cuticle is indicated by a brown dashed line. Scale bar 100 μm. (**C**) Cartoon depicting the pupal wing and surrounding cuticular sac, labeled to indicate the optical sections shown in panels (**D–F**). (**D–F'''**) Show optical sections through a 22 hAPF wing from a pupa harboring a Dumpy::YFP protein trap at the endogenous locus. (**D–E**) Show sections in the plane of the wing epithelium near the anterior (**D**) and posterior (**E**) margins. Dumpy::YFP is present in the space between the wing margin and the cuticle. Scale bar 20 μm. (**F–F'''**) Show optical sections orthogonal to the wing epithelium at different proximal-distal positions (indicated in **C**). Dumpy::YFP connects the dorsal wing surface and the cuticle in specific positions (arrowheads). Scale bars: 20 μm. (**G**) Summarizes the pattern of Dumpy::YFP connections between the dorsal wing surface and overlying cuticle—these lie over the hinge and vein regions. (**H**) Shows the pattern of Dumpy::YFP connections between the wing margin and cuticle—these extend around the wing margin except in the posterior/proximal regions. (**I–I'**) Summarize quantifications of circular laser ablation experiments performed in nine specific regions of five WT and five *dumpy^ov1* mutant wings. The size of the red circles indicates the initial rate of area expansion of the perimeter of the circular cut, which reflects the isotropic tissue stress. Green bars represent the direction and magnitude of the elliptical deformation of the initially

*Figure 2. continued on next page*

*Figure 2. Continued*

circular cut, reflecting the anisotropy of tissue stress (see 'Materials and methods', Analysis of circular laser ablations). Magenta bars depict the orientation and magnitude of local cell elongation. Scale bar: 100 μm.

The following figure supplements are available for figure 2:

**Figure supplement 1**. Dumpy::YFP cuticle imprint and Dumpy apical connections.

**Figure supplement 2**. *dumpy<sup>ov1</sup>* weakens distal attachments between wing margin and cuticle.

**Figure supplement 3**. Method to determine stresses in WT and *dumpy<sup>ov1</sup>* mutant.

To ask whether connections to the cuticle were required to maintain wing blade area, we examined blade area changes and the underlying cellular contributions in *dumpy$^{ov1}$* mutant wings (***Figure 3F,G***). As a complementary approach, we quantified the cellular contributions to area changes in laser-severed wings. We severed the wing either between the hinge and the blade, or at the very distal tip before hinge contraction occurred (***Figure 3—figure supplement 2***). We also severed connections between the wing margin and the cuticle (***Figure 1D,F***) at about 22 hAPF. In contrast to unperturbed WT wings, total wing area decreases dramatically when connections at the margin are weakened by *dumpy$^{ov1}$* mutation or by laser severing (***Figure 3G*** and ***Figure 3—figure supplement 2***, see dark blue curve). Thus, connections to the cuticle are required to maintain wing blade area during morphogenesis. These connections provide mechanical linkages that permit the buildup of tensile stresses while maintaining wing blade area.

How does epithelial stress influence the cellular events contributing to area homeostasis? To answer this question, we first compared cellular contributions to area change during morphogenesis of WT and *dumpy$^{ov1}$* mutant wings. Wing blade area decrease in *dumpy$^{ov1}$* mutant wings is not a consequence of fewer cell divisions—cells actually divide more than in WT (***Figure 3—figure supplement 2D–F,I***). Cells in *dumpy$^{ov1}$* mutant wings have a similar maximum division rate but divide over a longer period of time, resulting in more cells at the end of morphogenesis (***Figure 3G*** and ***Figure 3—figure supplement 2D–F***, yellow curve). The reduced wing blade area in *dumpy$^{ov1}$* is quantitatively explained by reductions in cell area and by cell extrusions, which more than compensate the increased proliferation. Thus, reduced epithelial stresses in *dumpy$^{ov1}$* wings perturb the balance between cell divisions, area changes and extrusions seen in WT.

All laser-severing perturbations decrease the final wing area, similar to *dumpy$^{ov1}$* mutant wings (***Figure 3—figure supplement 2***). In these wings, the analysis of cellular contributions to wing area changes is complicated by the delay between cutting and the initiation of time-lapse imaging (about 45 min). During this intervening time, the wing responds acutely to reduced tension, and both wing and cell area decrease to values below those expected for WT wings of the same stage (***Figure 3—figure supplement 2D–F,H***). While we can estimate changes to cell area during this time, we cannot know the rates of cell division and extrusion. Nevertheless, several interesting conclusions can be drawn by analyzing final cell

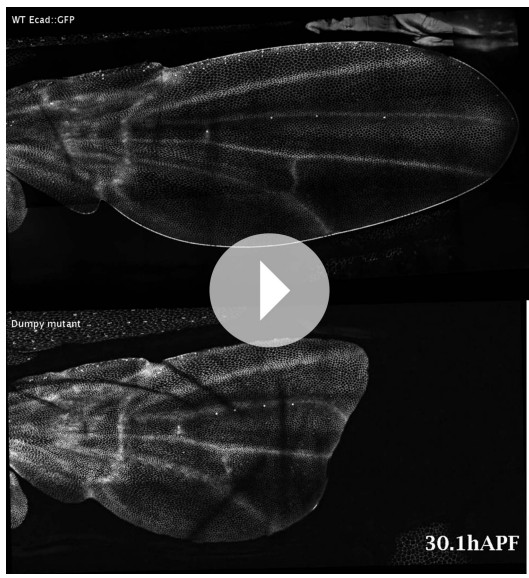

**Video 2.** Synchronized time-lapses of wild-type (WT) and *dumpy$^{ov1}$* wings. The synchronization is based on the time when histoblast nests merge at ~26.5 hAPF.

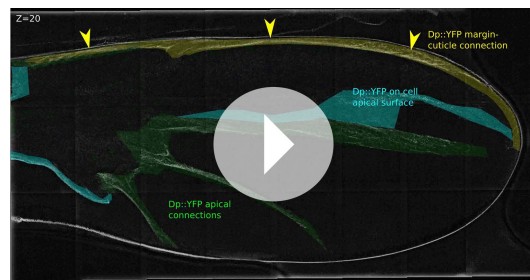

**Video 3.** Dumpy::YFP distribution in a 40 μm deep z-stack that was manually annotated to identify the regions where the protein is present.

area, and the rates of division, area change and extrusion after time-lapse imaging begins. Wings that have been severed before hinge contraction (whether at the hinge-blade interface or at the distal tip) behave similarly to *dumpy^ov1* mutant wings. After an initial delay, the rate of cell division increases and cells divide more than in unperturbed wings. However, cell extrusions and decreasing cell area more than compensate for increased cell division to produce smaller wings. When cuticle connections are severed later at 22 hAPF, most cell divisions have already occurred, and this treatment does not increase proliferation. In this case, the number of cell extrusions increases, and the final cell area is smaller than that of unperturbed wings. Taken together, analyzing *dumpy^ov1* and laser-severed wings shows that epithelial stresses are required to balance cell divisions with cell extrusions and cell area changes to maintain area homeostasis during morphogenesis. This is consistent with observations in the thorax, where overcrowding drives delamination (*Marinari et al., 2012*).

## A method to quantify cellular contributions to wing blade deformation

In the preceding section, we discussed how cellular processes contributed to wing blade area changes. These area changes correspond to the isotropic component of a tensor characterizing the tissue strain (*Figure 4A*). Now, we discuss shape changes of the wing blade, which correspond to the anisotropic part of this tensor and characterize the process of elongation along an axis (i.e., pure shear). The rate of change of pure shear is described by the pure shear rate tensor $\tilde{v}$ (*Figure 4B*) with $\tilde{v}_{xx}$ characterizing the rate of elongation along the PD axis (*Figure 1A*). Note that *pure* shear, that is, convergence-extension flow, is different from so-called *simple* shear, which results from a superposition of pure shear and a rotation (*Figure 4C*). In the following, we use the term shear to refer to pure shear. We now discuss how tissue shear can be decomposed into contributions from cell shape changes and topological changes. These include cell divisions, cell neighbor exchanges (T1 transitions), and cell extrusions (T2 transitions) (*Figure 4D–G*).

To understand the cellular contributions to the overall shear of the wing blade, we developed a method to distinguish and quantify shear caused by cell shape changes and shear caused by topological changes. In a piece of tissue where no topological changes occur within the cellular network, the deformation of the individual cells defines the deformation of the whole piece of tissue. However, when topological changes occur, deformation of individual cells no longer completely accounts for the overall shear (*Figure 4D–H*). The triangle method we outline below represents an exact geometrical formalism to decompose large-scale deformations of the wing blade into contributions by cell deformation and by each kind of topological change.

First, we tile the cellular network with triangles as follows. Each vertex of the cellular network that touches three cells (red dot in *Figure 4J*) gives rise to a single triangle (red), whose corners are defined by the centers of the three cells (green dots). Vertices that touch more than three cells are treated as described in Appendix 1, 'Triangulation procedure'. The resulting set of triangles tiles the cellular network without gaps or overlaps (*Figure 4K*). We choose a tiling into triangles, because the deformation of a single triangle between two frames of the video can be uniquely characterized by a single 2 × 2 tensor describing a linear transformation (see Appendix 1, 'Triangle deformation'). Note that such a characterization by a 2 × 2 tensor is in general not possible for polygons with more than three sides. For each triangle and time point, this tensor describes relative area changes, rotation, and shear of the triangle. The average shear rate of all triangles in the tissue corresponds to the overall tissue shear rate. To connect the tissue shear rate to cell elongation changes, we define a nematic tensor **Q** characterizing the state of triangle elongation (see Appendix 1, 'Triangle elongation'). Then, the change of triangle elongation corresponds exactly to triangle shear. Cell elongation is obtained as the average of the elongation tensors **Q** of those triangles that belong to a given cell. Hence, in the absence of topological changes, we find an exact relation between cell elongation change and overall tissue shear (Appendix 1, 'Large-scale shear in the absence of topological transitions').

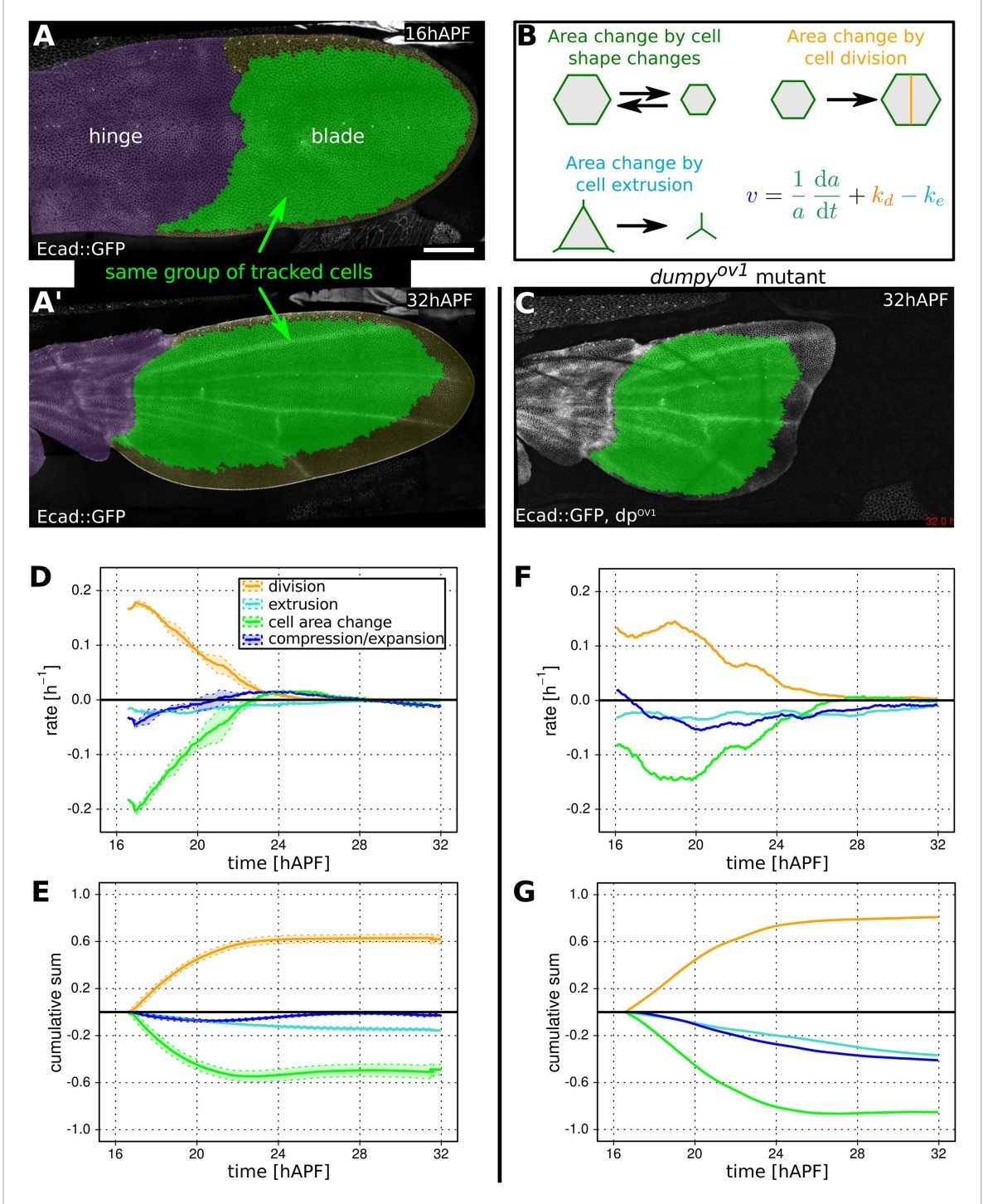

**Figure 3**. Cellular contributions to wing blade area changes. (**A**, **A'**) WT Ecad::GFP expressing wing at 16 hAPF (**A**) and (**A'**) at 32 hAPF. The hinge is colored purple. Green shaded region indicates the region of the blade in which segmented cells could be tracked from the beginning to the end of the video. This region was used for further analysis (**D**, **E**). (**B**) Cartoon illustrating cellular contributions to wing blade area change: cell area change, cell division and cell extrusion. The equation decomposes the relative area change of the entire wing blade (*v*) into the relative area changes due to each cellular contribution throughout the wing blade. (**C**) *dumpy*^ov1^ mutant wing at 32 hAPF. Green shaded region indicates tracked region used for analysis in (**F**, **G**). (**D**, **E**) Relative cellular contributions to wing blade area change over time, averaged over three WT wings. The rates of relative area change are shown in (**D**) and their cumulative sums are shown in (**E**). Lighter shaded regions in indicate standard deviations between wings. (**F**, **G**) Cellular contributions to wing blade area change in a *dumpy*^ov1^ mutant wing. Cumulative plots (**G**) were generated starting at 16.5 hAPF, the earliest time common to all compared videos. Scale bar 100 μm.

*Figure 3. continued on next page*

*Figure 3. Continued*

The following figure supplements are available for figure 3:

**Figure supplement 1**. Reproducibility of cell number and cell area in three WT wings.

**Figure supplement 2**. Epithelial tension is required to maintain wing blade area.

If topological changes occur, then average cell elongation change is not equal to tissue shear. To include the effect of topological transitions, we write (Appendix 1, 'Contributions to shear by topological changes'):

$$\tilde{v} = \frac{D\boldsymbol{Q}}{Dt} + \boldsymbol{R}, \tag{2}$$

where $\tilde{v}$ is the overall tissue shear rate tensor, $D\boldsymbol{Q}/Dt$ is a corotational rate of change in average triangle elongation, and $\boldsymbol{R}$ is the shear rate tensor due to topological changes. Contributions to $\boldsymbol{R}$ include T1 transitions ($\boldsymbol{T}$), cell divisions ($\boldsymbol{C}$), T2 transitions ($\boldsymbol{E}$), as well as correlated cell shape changes and cell rotations ($\boldsymbol{D}$): $\boldsymbol{R} = \boldsymbol{T} + \boldsymbol{C} + \boldsymbol{E} + \boldsymbol{D}$ (see *Figure 4I*).

How can we define the contributions by topological changes ($\boldsymbol{T}$, $\boldsymbol{C}$, and $\boldsymbol{E}$) to tissue shear? During a topological transition, the triangulation changes and thus the average triangle elongation changes (*Figure 4L–O*). However, at the moment the topological change occurs there is no tissue shear. Therefore, tissue shear and triangle elongation are no longer the same. This can be compensated by introducing a contribution to tissue shear by topological transitions. This contribution corresponds to the negative change in the average triangle elongation caused by the change in the triangulation (*Figure 4M* and Appendix 1, 'Intermediate network states' and Appendix 1, 'Contributions to shear by topological changes').

In the definition of $\boldsymbol{R}$ in *Equation 2*, we have also introduced the contribution $\boldsymbol{D}$ to tissue shear, which accounts for collective cellular events that combine to increase average triangle elongation in the absence of tissue shear and topological transitions. This occurs when several triangles have fluctuating shapes, such that the instantaneous elongation and the rotation rate or area expansion rate of triangles are correlated. Note that this effect does not occur when several triangles undergo equal deformations and rotations. One example of a cellular network deformation that produces the contribution $\boldsymbol{D}$ to tissue shear is shown in *Figure 4H*. Here, cells in neighboring rows slide relative to each other in alternating directions, such that no net pure shear occurs. However, there are alternating rows of *simple* shear and a net change in triangle elongation. We introduce the contribution $\boldsymbol{D}$ in the definition of $\boldsymbol{R}$ in order to compensate for the increase in the average triangle elongation $D\boldsymbol{Q}/Dt$ stemming from such correlations, should they exist in the wing blade.

## Patterns of cellular contributions to tissue shear

We first used the triangle method to calculate the patterns of tissue shear and cellular contributions to this tissue shear in WT (*Figure 5*, *Video 6*, and Appendix 1, 'Spatially resolved shear patterns') and *dumpy^{ov1}* mutant wings (*Figure 5—figure supplement 1* and *Video 7*). To visualize these patterns we averaged all quantities within squares of about 26 × 26 μm². *Figure 5* shows shear patterns in WT at early, intermediate, and late time points during pupal wing morphogenesis. Shear is indicated by a line whose orientation represents the shear axis and whose magnitude corresponds to the shear rate.

Tissue shear in the WT wing blade is oriented in a fan-shaped pattern with a strong PD component (*Figure 5A–A''*). At about 21 hAPF, the shear pattern develops a sharp reorientation between veins L3 and L4, where shear is oriented along the AP axis. This region corresponds to a stripe of Dumpy-containing matrix that attaches the blade

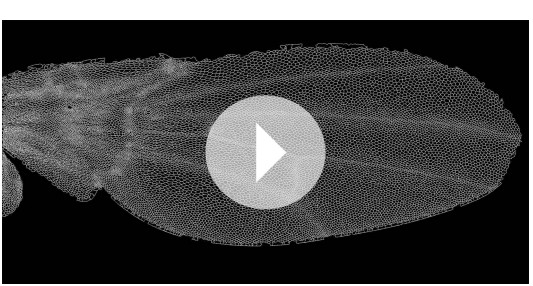

**Video 4.** Cell outline obtained from the segmentation procedure.

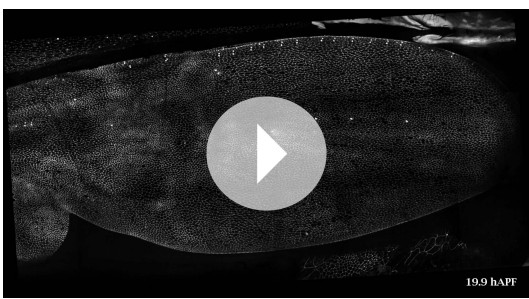

**Video 5.** High resolution video of a WT wing expressing Ecad::GFP.

to the cuticle (*Figure 2—figure supplement 1B*). Shear decreases over time, beginning distally and proceeding proximally, and finishes by about 32 hAPF.

What are the patterns of cellular contributions to the tissue shear? These patterns reveal a surprising complexity that changes with time. Shear caused by cell elongation and cell rearrangements (T1 transitions) display significant contributions that are antagonistic. Unexpectedly, T1 transitions cause shear along the AP axis early in the process (*Figure 5B–B''*). This shear is opposed by increasing cell elongation along the PD axis (*Figure 5C–C''*). At intermediate and late stages, T1 transitions shift their average orientation to cause PD shear—at the same time, cells reduce their elongation along the PD axis, causing AP shear. The contribution of cell extrusions to tissue shear is negligible (not shown) and cell divisions result in significant shear only during the early stages of wing morphogenesis (*Figure 5D–D''*). Finally, plotting the contribution $D$ to shear due to correlation effects reveals that these effects do exist in the wing (*Figure 5E–E''*). Thus, patterns of cell elongation, cell rearrangement, cell division and correlation effects make dynamically changing contributions to tissue shear that are sometimes antagonistic.

To investigate the cause of altered wing shape in *dumpy^{ov1}*, we performed a similar analysis. The patterns of cellular contributions to tissue shear in a *dumpy^{ov1}* mutant wing display subtle abnormalities (*Figure 5—figure supplement 1*). However, a more quantitative analysis is required to understand the origin of the altered *dumpy^{ov1}* mutant wing shape.

## Total cellular contributions to tissue shear

To better understand the quantitative relationships between the cellular processes contributing to tissue shear, we studied spatially averaged shear in the wing blade projected on the PD axis. We quantified average tissue shear, and shear caused by each cellular contribution over time in three different WT videos. These averages were taken over the tracked region shown in *Figure 3A* and the resulting quantities were further averaged over the three videos. These averages and the standard deviations between the videos are shown in *Figure 6A,B*. Positive values indicate shear along the PD axis and negative values indicate shear orthogonal to the PD axis (i.e., shear in the AP axis). Adding the contributions of shear caused by cell divisions, cell rearrangements, cell shape changes, and correlation effects (light-pink line in *Figure 6—figure supplement 1A*) reproduces the independently calculated total shear curve (blue line in *Figure 6A* and *Figure 6—figure supplement 1A*). Small differences between these two curves (about 3%) stem from small inaccuracies (see Appendix 1, 'Decomposition of the large-scale tissue shear rate'). Thus, we can decompose tissue shear into its individual cellular contributions.

Does the cumulative tissue shear we calculate account for the shape change of the wing blade? To verify this, we characterize the blade shape by a nematic determined by the outline of the tracked region (see Appendix 1, 'Characterization of wing blade anisotropy'). The change of this nematic with respect to its initial value over time is shown together with the cumulative tissue shear obtained by the triangle method (*Figure 6—figure supplement 1B*). These quantities agree well, indicating that the shear projected on the PD axis accounts for the main features of tissue shape changes. Thus, we can now use the shear decomposition method to discuss how different cellular events contribute to shape change of the wing over time.

On average, the wing shears smoothly along its PD axis between 17 and 32 hAPF as the hinge contracts. In contrast, the cellular processes that combine to produce tissue shear change over time. During the first 6 hr of our videos, shear caused by cell elongation in the PD axis (green curves in *Figure 6A,B*) is even larger than PD tissue shear (blue curves). This shows that cell elongation increases more than the tissue elongates suggesting that active cellular processes also contribute to PD cell elongation. Subsequently, starting at about 22 hAPF, cell elongation decreases although the

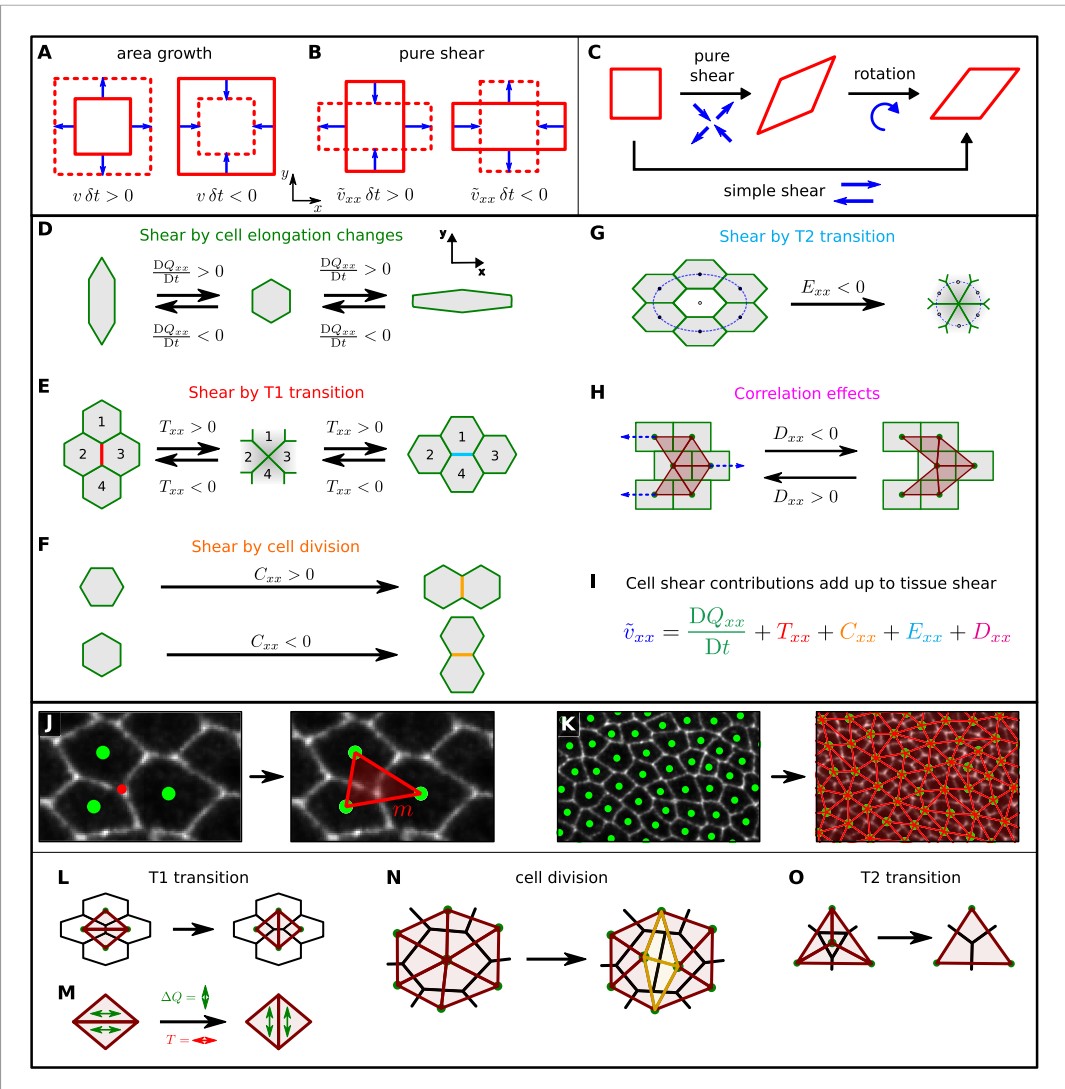

**Figure 4**. A method to quantify cellular contributions to wing blade deformation. (**A**, **B**) Isotropic part of tissue deformation, that is, tissue area growth (**A**), and anisotropic part of tissue deformation, that is, tissue pure shear, along the x axis (**B**). The deformation occurs during a time interval $\delta t$. (**B**) The sign of the pure shear rate component $\tilde{v}_{xx}$ indicates the shear direction. Positive means that the shear deformation occurs along the PD axis (also referred to as x axis). Negative corresponds to shear along the AP axis (or y axis). (**C**) A simple shear deformation corresponds to a superposition of a pure shear deformation and a rotation. (**D**–**H**) Cartoons depicting how changes in cell shape, cell rearrangements, cell divisions, cell extrusions and correlation effects could produce tissue shear. (**I**) The equation decomposes the tissue shear rate into shear contributed by each of these cellular processes (color code as in **D**–**H**). For simplicity, the tensorial equation was projected onto the PD axis as most of the deformation occurs along the PD axis. (**J**, **K**) Triangulation method. (**J**) The cellular network is tiled with triangles: each vertex (red dot) of the cellular network that touches three cells gives rise to a single triangle (red), whose corners are defined by the centers of the three cells (green dots). (**K**) The resulting set of triangles tiles the cellular network without gaps or overlaps. (**L**–**O**) Triangle network modifications upon topological changes due to cell rearrangements (**L**), cell divisions (**N**), and extrusions (**O**). (**M**) The discontinuous change in average triangle elongation during a given topological change is used to calculate the shear induced by the topological change.

blade continues to elongate. These discrepancies between cell shape changes and tissue shape changes require topological changes in the cell network. To more clearly discuss these events, we define two distinct phases of wing morphogenesis (phases I and II) that are separated by the peak of cell elongation occurring at about 22.5 hAPF.

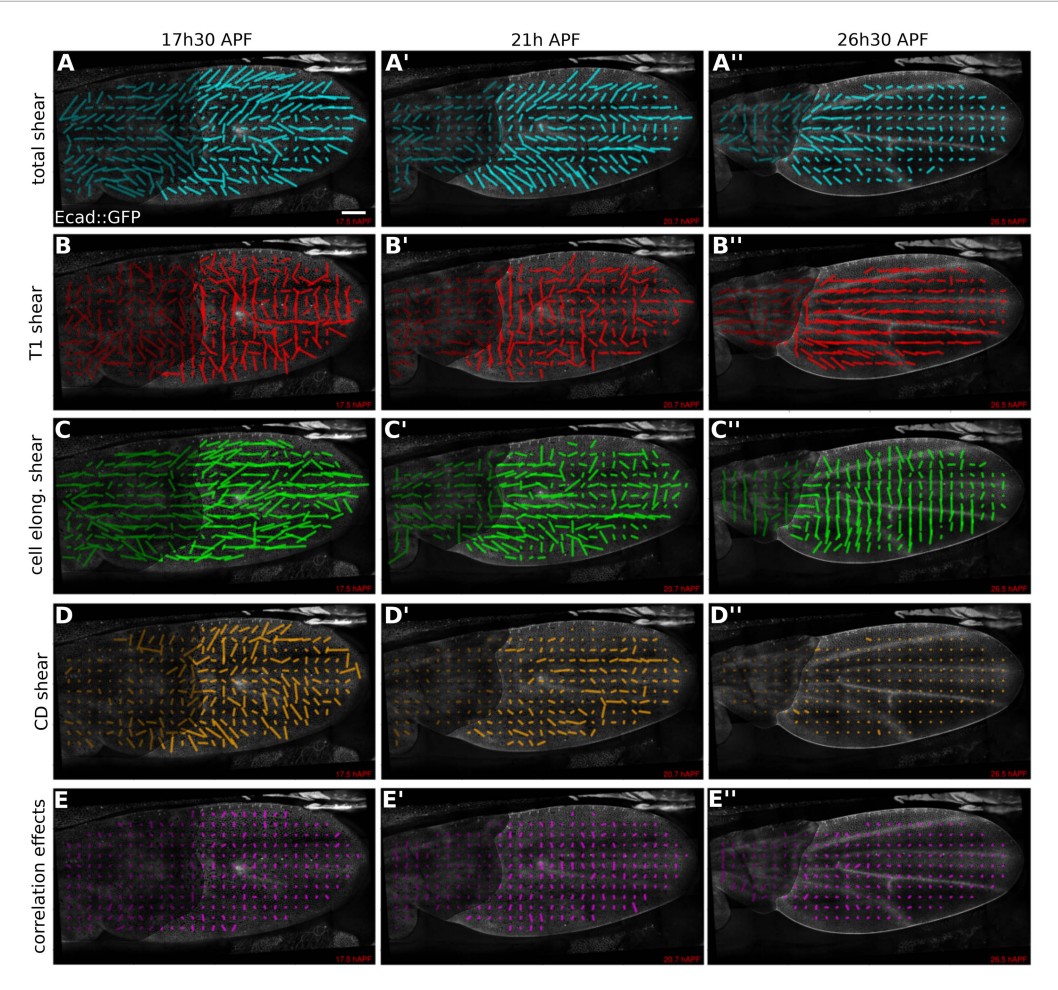

**Figure 5**. Patterns of cellular contributions to tissue shear in unperturbed WT wings. (**A–E''**) Patterns of local tissue shear rates (**A–A''**), local shear rates contributed by cell rearrangements (T1 transitions, **B–B''**), cell shape changes (**C–C''**), cell divisions (**D–D''**), and correlation effects (**E–E''**), in a WT wing at 17.5 hAPF (phase I), 21 hAPF (intermediate phase) and 26 hAPF (phase II). The shear rate and shear rate contribution tensors were locally averaged within 26 × 26 μm$^2$ square elements (25–50 cells) of a fixed grid. A 45 min time window was used to smooth the shear values within each grid element. The resulting nematic tensors are represented by line segments whose length and direction correspond to the norm and orientation of the tensor, respectively. Scale bar: 100 μm.

The following figure supplement is available for figure 5:

**Figure supplement 1**. Patterns of cellular contributions to tissue shear in *dumpy*$^{ov1}$ mutant.

Quantifying shear caused by T1 transitions and by cell elongation reveals that they change dynamically throughout morphogenesis with a striking reciprocal relationship (green and red curves in *Figure 6A,B*). This reciprocal relationship accounts to a large extent for the discrepancies between cell elongation and tissue shear. It further suggests that the active contribution to cell elongation (i.e., the amount of cell elongation that exceeds tissue shear) may be linked to AP-oriented T1 transitions—the orientation of these T1s, which work against the observed tissue shear, suggests that they are autonomously controlled. Active AP-oriented T1 transitions could produce PD cell elongation if mechanical constraints prevent the wing from shearing. In principle, it is also possible that active PD-oriented cell shape changes could produce AP-oriented T1 transitions under the same constraints. Cell divisions also contribute to PD shear

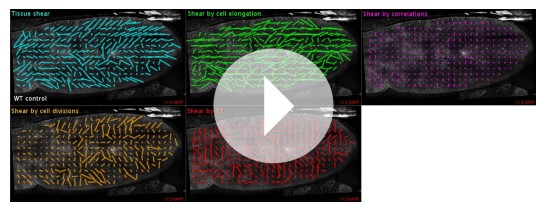

**Video 6.** Dynamic patterns of tissue shear and of its cellular contributions in a WT wing. Line segments are nematic representations of shear.

in the wing blade. Although cell divisions initially cause a small amount of AP shear, their direction changes during phase I such that their net contribution shears the wing in the PD axis. In addition, correlation effects produce significant shear in the AP direction and contribute most strongly at the time that T1 transitions are changing their orientation.

In summary, the continuous large-scale deformation of the wing blade emerges from complex patterns of cell dynamics on small scales. During phase I, cells undergo AP-oriented T1 transitions while elongating in the PD axis. Cell divisions during phase I contribute shear along the PD axis. During phase II, the orientation of T1 transitions shifts to the PD axis and cells relax their shape. Correlation effects contribute AP shear, and peak roughly at the time that T1's shift their orientation.

To ask whether the cellular contributions to tissue shear were independent of each other, we compared the sum of the variances of the final cellular contributions to tissue shear with the variance of final tissue shear itself. The ratio of these values is about 25, indicating that the cellular contributions to tissue shear—like the contributions to area change—are not independent of each other. Thus, the overall tissue shear is more reproducible than would be expected from the variations of the cellular contributions.

## Alternating directions of shear and rotation in neighboring cell rows contribute to tissue shear

What exactly are cells doing that results in correlation effects? We found that shear due to correlation effects was mainly generated by correlations between local elongation and rotations (*Figure 6C–I*). To investigate this further, we determined the magnitude and orientation of shear and the rotation rate associated with each triangle for each frame of the video. We observed that triangles rotated and sheared in striking spatial patterns that rapidly fluctuate in time (*Figure 6C,F,G*). These patterns correspond to rows of correlated shear and rotation that are distributed throughout the wing blade. To characterize these correlated patterns, we calculated the spatial power spectrum of the local tissue rotation rate (*Figure 6D*, Appendix 1, 'Power spectrum of local tissue rotation'). This revealed that shear and rotation are correlated in regions corresponding to PD-oriented rows that were about 3–7 triangles long. These rows consist of triangles that all rotate and shear in the same direction. The rows are interspersed with other rows of similar length with mirrored patterns of shear and rotation (note blue and red rows in *Figure 6F,G*). Such a pattern of rotation and pure shear is characteristic of neighboring rows of triangles and cells undergoing simple shear in alternating directions (*Figure 6H*). This would occur if PD-oriented rows of cells slide with respect to each other. As discussed above, such rearrangements can indeed contribute to correlation terms (see cartoon in *Figure 4H*).

If rows of cells slide past each other, cells typically engage in T1 transitions. Since the peak of AP correlation effects coincided with a shift in the net orientation of T1 transitions (i.e., when the red curve in *Figure 6A* crosses 0), we wondered whether correlation effects could be associated with T1 transitions at this time (see also *Figure 6—figure supplement 1C*). We therefore examined whether correlation effects were associated with a particular type of topological change. Indeed, correlation effects are mostly accounted for by those cells that are about to undergo a T1 transition within the next 9 frames, although they cover only a small fraction of the total area (*Figure 6I*, Appendix 1, 'Role of T1 transitions in the correlation-induced shear').

For rows of cells to slide past each other, cells would have to undergo a peculiar type of T1

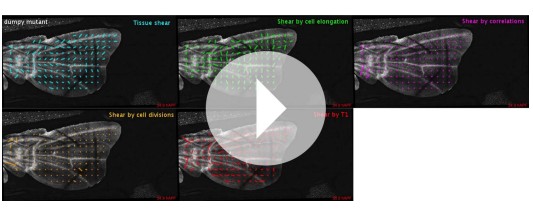

**Video 7.** Dynamic patterns of tissue shear and of its cellular contributions in a *dumpy*[ov1] mutant. Line segments are nematic representations of shear.

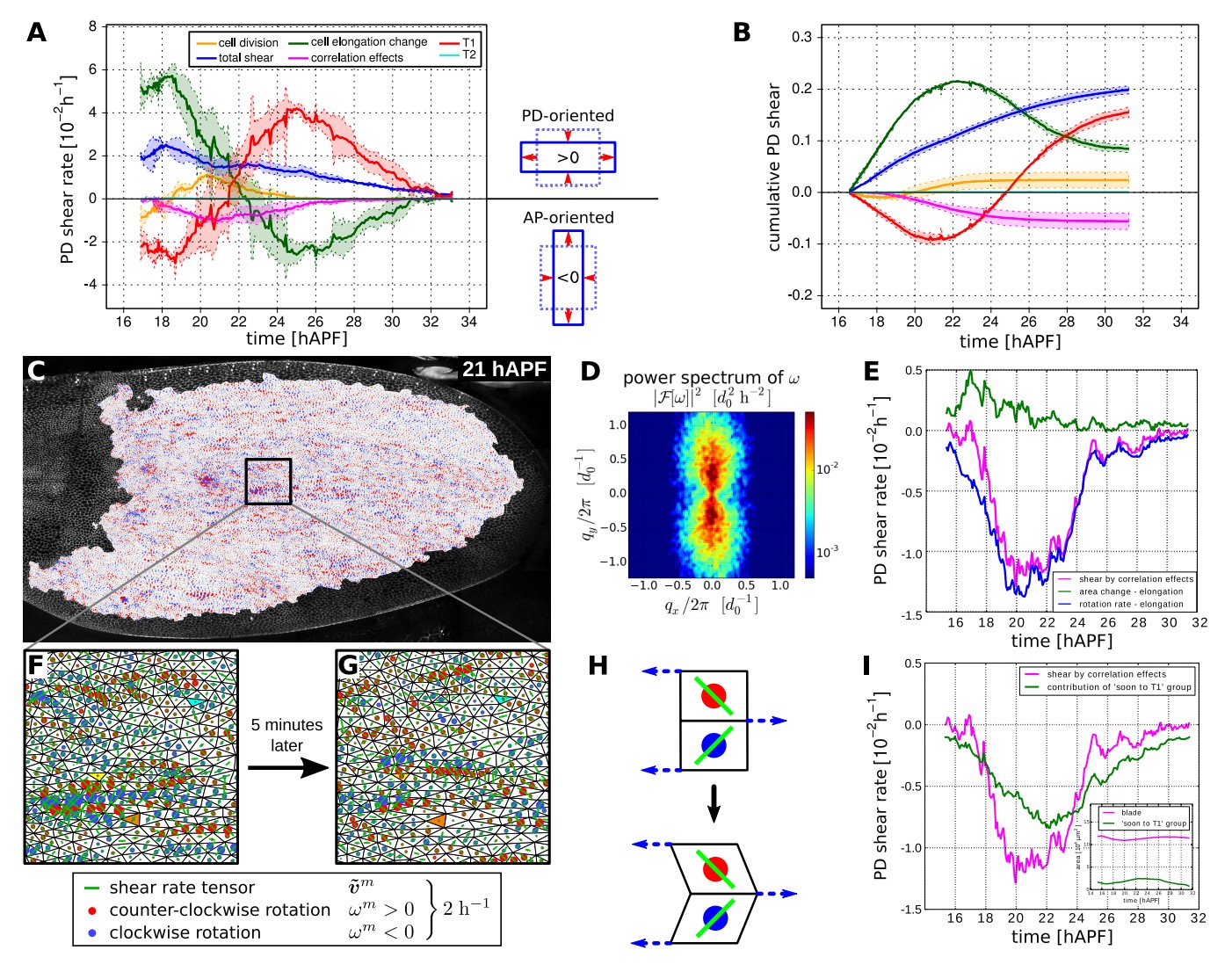

**Figure 6**. Total cellular contributions to tissue shear throughout the WT wing blade. (**A**) Shows the tissue shear rate (blue) over time, and shear rates contributed by cell rearrangements (red), cell shape changes (green), cell divisions (orange), and correlation effects (magenta), averaged throughout the wing blade. These averages were taken over the tracked region shown in **Figure 3A** by averaging nematic tensors throughout the wing blade. The resulting quantities were further projected onto the PD axis and averaged over the three WT videos. Ribbons indicate the standard deviation between wings. The sign of the shear rate defines its orientation (>0 is PD-oriented and <0 is AP-oriented). (**B**) Shows the accumulated tissue shear over time throughout the blade, and the accumulated contributions of each cellular process (color code as in **A**). (**C**) Pattern of local tissue rotation rate at 21 hr after puparium formation (APF). The local tissue rotation rate $\omega^m$ is plotted separately for each triangle $m$. Red circles correspond to a counter-clockwise rotation and blue circles correspond to a clockwise rotation. The area of each circle scales with the absolute rotation rate. (**D**) The spatial power spectrum of the local tissue rotation rate corresponding to the pattern in panel **C** (see Appendix 1, 'Power spectrum of local tissue rotation'). The power spectrum is a function of a wave vector $q = (q_x, q_y)$, which is measured in units of a typical cell diameter $d_0 = 4$ μm. The two peaks in the power spectrum at $q_{peak} \approx (0, \pm 0.3 d_0/2\pi)$ correspond to the existence of horizontal bands of alternating tissue rotation that are separated by about 1.5 cell diameters (compare panels **F**, **G**). (**E**) Correlation effects contributing to shear along the PD axis, $D_{xx}$ (magenta curve). $D_{xx}$ can be decomposed into an area expansion part $D_{xx}^e$ (green curve), which corresponds to a correlation between the local area expansion rate $v^m$ and local triangle elongation $Q_{xx}^m$: $D_{xx}^e = -(\langle v^m Q_{xx}^m \rangle - \langle v^m \rangle \langle Q_{xx}^m \rangle)$, and into a rotational part $D_{xx}^r$ (blue curve), which corresponds to a correlation between the local tissue rotation rate $\omega^m$ and local triangle elongation: $D_{xx}^r \approx 2(\langle \omega^m Q_{xy}^m \rangle - \langle \omega^m \rangle \langle Q_{xy}^m \rangle)$ (see Appendix 1, 'Large-scale shear in the absence of topological transitions'). The rotational part dominates the shear by correlation effects. (**F**) Enlargement of the rotation pattern in panel **C** with an additional indication of the pattern of the local shear rate tensor by green bars. Length and orientation of a bar correspond to magnitude and axis of the local shear rate, respectively. The axis of local shear is correlated with the sign of local rotation (indicated by red and blue circles). (**G**) The same region of the wing in the subsequent frame (about 5 min later). Three corresponding triangles in panels **F** and **G** are colored in cyan, yellow and orange, respectively. The patterns of local shear and rotation change on time scales of minutes. (**H**) A correlation of local rotation and local shear within bands as shown in panels **F**, **G** corresponds to bands of alternating simple shear. (**I**) Contribution to the shear due to correlation effects of the group of triangles that are going to disappear due to a T1 transition within nine video frames (<45 min) (Appendix 1,

*Figure 6. continued on next page*

*Figure 6. Continued*

'Role of T1 transitions in the correlation-induced shear'). Inset: the area of this group is small compared to the total blade area, although it accounts for a significant amount of shear due to correlation effects in the blade.
The following figure supplement is available for figure 6:

**Figure supplement 1**. The shear decomposition method effectively describes tissue deformation and cellular contributions to tissue shear.

transition in which the orientation of the boundaries gained and lost are similar. Boundary orientations around T1 transitions are difficult to measure, because boundary length is small. The triangulation method provides us with a more robust measure of a bond orientation. As a proxy for cell boundary orientation, we use the orientation of the line connecting the centers of the two corresponding triangles (*Figure 6—figure supplement 1D*). We calculated the proportion of connection losses or acquisitions occurring in different directions over the course of morphogenesis, and compared them with the distribution of connection angles in general. We observe that at early times, when T1-induced shear is mainly along the AP axis, connections are typically lost along the PD axis and gained along the AP axis (*Figure 7A–D*). At intermediate times, when the correlation term is maximal, both lost and gained connections were oriented along the PD axis (*Figure 7B*). This is consistent with the unusual T1 transitions associated with sliding cell rows. Finally, as T1 transitions begin to cause net PD shear, AP connections are preferentially lost and PD connections are gained (*Figure 7D*). Thus, as T1 transitions shift from an AP- to a PD-oriented shear, they pass through an intermediate state where connection gains and losses are still oriented but do not cause shear. Interestingly, at the same time, the correlation term has maximal magnitude. This suggests that the correlation effects are related to these unusual patterns of T1 transitions at intermediate times.

## Interplay of cell dynamics and tissue stresses during tissue shear

To investigate which cellular events depended on epithelial stresses, we quantified shape changes and cellular contributions to tissue shear in the *dumpy^{ov1}* mutant wing blade (*Figure 8A,B*). As a complementary approach, we also studied these events in wings that had been subjected to laser severing (*Figure 8C–F* and *Figure 8—figure supplement 1*). In *dumpy^{ov1}* wings, cells experience hinge contraction but the counterforces exerted by cuticle connections seem to be reduced (*Figure 2A–A''*). Tissue shear is dramatically altered in *dumpy^{ov1}* mutant wings as compared to WT—instead of shearing in the PD axis, these wings shear on average along the AP axis (*Figure 8A,B*). Examining the different contributions to tissue shear in *dumpy^{ov1}* wings shows that the rates of AP shear caused by T1 transitions and by correlation effects are similar to WT and persist for longer times. Thus, these processes are likely to be autonomously driven. By the end of the video, they cause more accumulated AP-oriented shear than in WT. Analogously, cell divisions cause more cumulative PD shear than in WT—consistent with the increased number of cell divisions in the *dumpy^{ov1}* mutant wing. In contrast, cell elongation during phase I causes less PD shear than in WT. Thus, PD-oriented epithelial stresses must contribute to PD cell elongation. Interestingly, the increase of cell elongation in the PD axis still exceeds the increase of elongation of the blade in the *dumpy^{ov1}* mutant wing. This suggests that autonomous cellular processes cause the residual PD cell elongation in *dumpy^{ov1}* mutant wings. Finally, PD shear by T1 transitions in phase II is smaller than in WT. This is not due to a premature cessation of T1's—indeed quantifying the rate of T1 transitions (regardless of orientation) shows that they occur at a higher rate and for a longer time than in WT wings (*Figure 8—figure supplement 2*). Rather, T1 transitions fail to orient as effectively with the PD axis in the *dumpy^{ov1}* mutant wing (see shear patterns in *Figure 5—figure supplement 1B–B''*).

As a complementary approach, we asked how tissue shear and cell dynamics were altered in wings that had been subjected to laser severing. We first examined wings that had undergone laser severing before hinge contraction at the distal tip of the wing blade (*Figure 8C,D* and *Figure 8—figure supplement 1A,B*). These wings experience mechanical conditions similar to those in *dumpy^{ov1}* wings in that they undergo hinge contraction but cannot attach properly to the distal cuticle. Consistent with this, the final shape of distally severed wings resembles that of *dumpy^{ov1}* wings. Furthermore, distally severed wings show similar tissue shear and cellular contributions to shear as *dumpy^{ov1}* mutant wings.

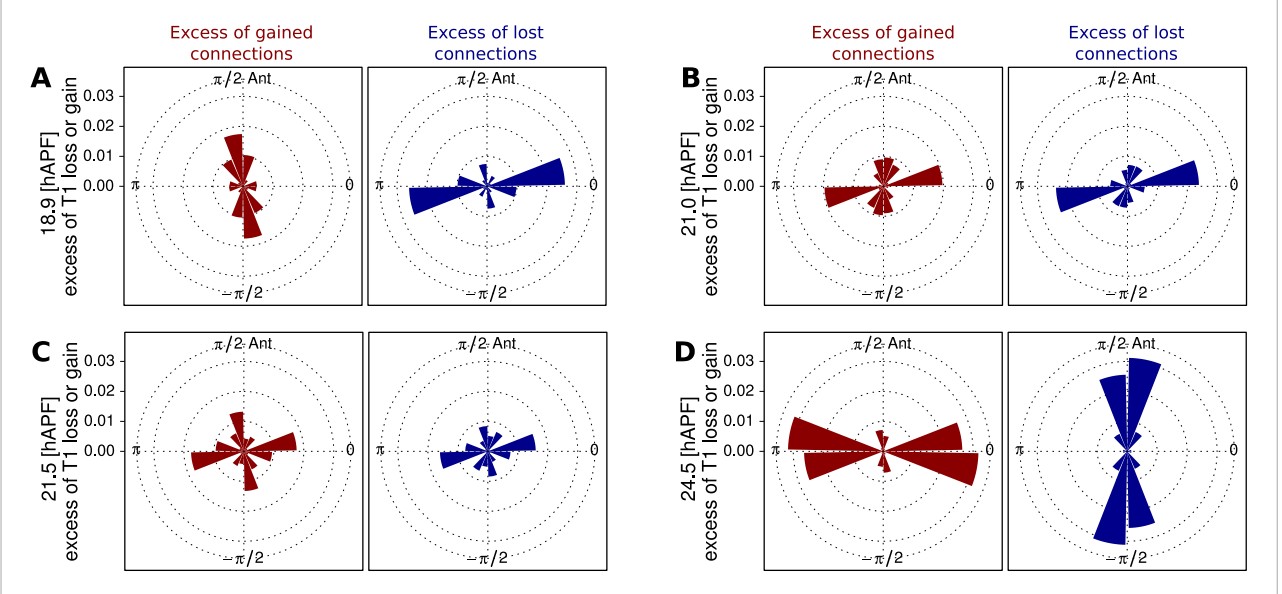

**Figure 7**. Changes in angular distribution of lost and newly formed cell–cell junctions. (**A–D**) Effective proportions of cell–cell connections that are lost (blue) or gained (red) in different directions as a consequence of cell rearrangements in a WT wing. To calculate these effective proportions, we subtracted the angular distribution of all cell boundaries from the angular distribution of cell–cell connections that were lost or gained by cell rearrangements, revealing the orientation of cell boundaries with a disproportionate tendency to be lost or gained. Rose diagrams show angles of cell boundaries that are more likely to be gained (red) or lost (blue) at specific times during the video corresponding to important changes in cell dynamics: (**A**) 18.9 hAPF (peak of negative shear rate by cell rearrangements), (**B**) 21 hAPF (peak of correlation effects), (**C**) 21.5 hAPF (shear rate by cell rearrangements crosses zero) and (**D**) 24.5 hAPF (peak of positive shear rate by cell rearrangements).

These observations suggest that weakened connections to the distal cuticle are key to the altered wing shape of *dumpy^ov1* mutants.

To perturb PD stresses even more strongly, we severed the region between the hinge and the blade before hinge contraction occurred (*Figure 8E,F*). These wings are not subjected to externally generated PD stresses at all, although they may still experience autonomously-induced stresses. We expected that cellular events that depended on externally generated stresses would be even more strongly perturbed in proximally severed wings than in *dumpy^ov1* or distally-severed wings. The proximally severed wing undergoes significant but short-lived PD shear during the molting process until 18 hAPF. Subsequently, PD shear stops and becomes even negative by 20 hAPF. Cells do not elongate in the PD axis as much as in WT wings, although PD cell elongation still exceeds PD tissue shear. Again, this suggests that only a fraction of PD cell elongation is normally caused by external stresses, and that autonomous cellular events must produce the residual PD cell elongation in proximally severed wings. PD shear due to cell division is larger than in WT (as it is in *dumpy^ov1* and distally severed wings) confirming that these divisions do not depend on external stresses. Furthermore, like *dumpy^ov1* and distally severed wings, proximally severed wings undergo greater AP shear resulting from correlation effects. Thus the cellular events underlying correlation effects produce even more shear when stresses are reduced. However later, T1 transitions fail to generate significant PD shear in proximally severed wings. The reduction in T1-dependent PD shear is much stronger than in either *dumpy^ov1* or distally severed wings, confirming that reorientation of T1 transitions in phase II is dependent on externally generated PD stress. At the very beginning of the video, PD-oriented connections are preferentially lost and AP-oriented connections are preferentially gained, consistent with the AP shear caused by T1 transitions at this time (*Figure 8—figure supplement 2C–F*). However, unlike WT, the preferential loss of connections never shifts towards the AP axis. Loss of connections remains biased towards the PD axis throughout the video—despite the fact that net shear caused by T1 transitions becomes very small. Shear caused by T1 transitions becomes small because the preferred orientation of gained connections gradually shifts from the AP to the PD axis. By the end of the video, both the assembly and disassembly of cell boundaries are preferentially oriented along the PD axis. These observations suggest that PD-oriented cell boundaries

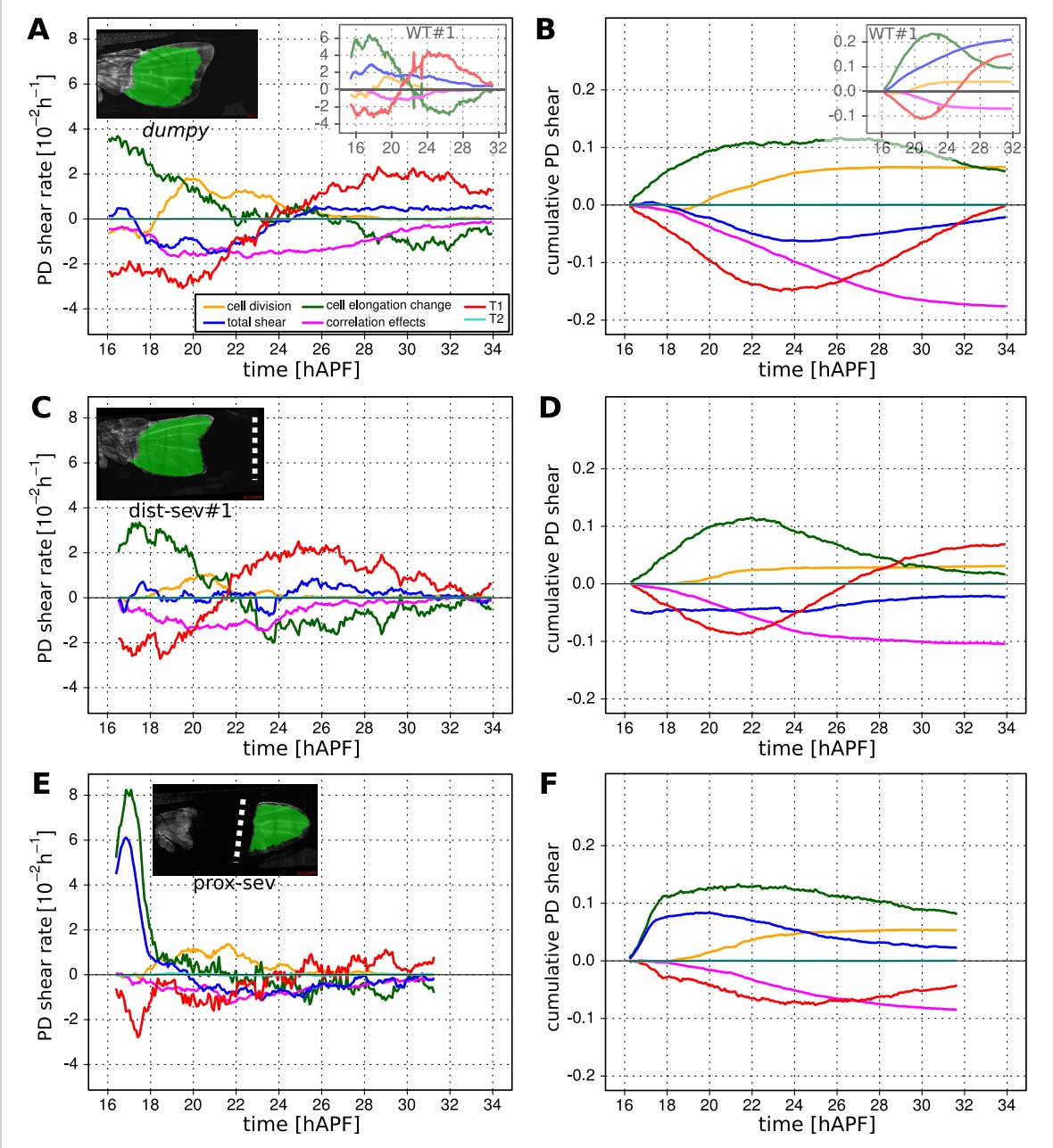

**Figure 8**. Total cellular contributions to tissue shear throughout perturbed wing blades. (**A–F**) Show total shear rates (**A**, **C**, **E**) and total accumulated shear (**B**, **D**, **F**), along with their cellular contributions, in the *dumpy*$^{ov1}$ wing blade (**A**, **B**), and wing blades severed either distally (**C**, **D**) or proximally (**E**, **F**) before hinge contraction (~16 hAPF). Blue = total tissue shear, Red = shear due to T1 transitions, Green = shear due to cell elongation change, Orange = shear due to cell division, Magenta = shear due to correlation effects. Corresponding plots for WT wings (identical to those in *Figure 6*) are inset in the upper right corners of (**A**, **B**) for the purposes of comparison. Insets in left corners of (**A**, **C**, **E**) show the tracked regions of each wing in green at 32 hAPF. To plot accumulated tissue shear in laser ablated wings, we offset the calculated accumulated tissue shear (blue) by a value corresponding to the difference in blade elongation before ablation and at start of recording (see *Figure 3—figure supplement 2*). All videos were aligned in time by taking the histoblast nests fusion time as a reference at about 26.5 hAPF. All cumulated shear curves start at 16.2 hAPF, which is the earliest common time point registered in all compared videos, including the dist-sev#2 video shown in *Figure 8—figure supplement 1A,B*.

The following figure supplements are available for figure 8:

**Figure supplement 1**. Total cellular contributions to tissue shear throughout perturbed wing blades.

*Figure 8. continued on next page*

*Figure 8. Continued*

**Figure supplement 2**. Effect of mechanical perturbations on T1 transitions.

**Figure supplement 3**. Total cellular contributions to tissue shear in wing blades after laser-severing of the extracellular matrix.

have a greater tendency to disassemble than those oriented at other angles, and that this is an autonomous, planar polarized feature of wing epithelial cells.

To disturb connections to the overlying cuticle without damaging the wing epithelium we disrupted apical extracellular matrix between the wing margin and the cuticle shortly before the onset of phase II, when this region becomes accessible (see *Videos 8, 9*). When anterior connections are severed, the wing blade shears even more in the PD axis while the area decreases slightly (*Figure 8—figure supplement 3*). This suggests that these connections restrain the narrowing of the wing blade in the AP axis. Increased PD tissue shear is mainly a consequence of T1 transitions. Since severing anterior connections reduces AP stress while PD stress does not change, this supports the idea that T1 events during phase II are oriented by anisotropic stress.

When distal connections are severed, cells rapidly reduce their elongation and area (*Figure 8—figure supplement 3*). Surprisingly, the rate of PD shear caused by T1 transitions continues to increase as it does in WT at this stage. However, after about 4 hr the PD shear rate due to T1 transitions decreases prematurely. Since the analysis of *dumpy$^{ov1}$* and proximally severed wing shows that PD T1 transitions are stress dependent, this suggests that there is a time delay between the change in tissue stress and the resulting T1 transitions.

Taken together, the analyses of *dumpy$^{ov1}$* wing blades and laser-severed wing blades distinguish autonomously driven cellular processes from those induced by tissue stresses. PD cell elongation and PD-oriented T1 transitions clearly depend on tissue stresses, whereas PD-oriented cell divisions are autonomously driven. AP-oriented T1 transitions and a corresponding fraction of PD-oriented cell elongation are also driven autonomously, as are the cellular events underlying correlation effects.

The overall picture that emerges is that changes of wing blade shape arise due to force balances that involve stresses exerted at the boundary of the tissue, and internal tissue stresses. Boundary stresses are due to hinge contraction and to the resistance of extracellular matrix attachments to the cuticle. Internal tissue stresses are generated by cell autonomous processes, like T1 transitions, and by elastic cell deformations. Tissue mechanics depends strongly on elastic connections of the wing to the cuticle. We must now understand how these isotropic and anisotropic mechanical stresses in the tissue, combined with boundary stresses, lead to cell and tissue remodeling. The interplay between boundary stresses and forces generated in the tissue is complex and requires a physical approach. We now present a continuum mechanical theory to understand these force balances and to calculate both tissue and cell shape changes.

We first define tissue material properties, starting with elastic properties, adding cell autonomous stresses and tissue shear due to topological changes. We then introduce elastic linkers to the surrounding cuticle. Finally, we compare predictions of this theory to the experimentally measured cell and tissue shape changes and determine key biophysical parameters characterizing tissue material properties.

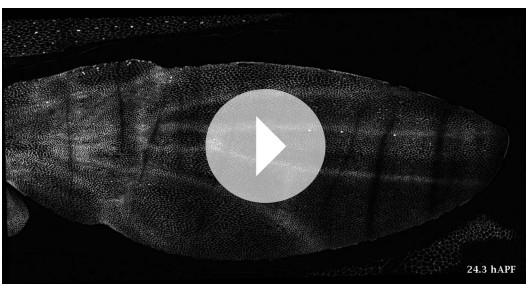

**Video 8.** Video of a WT wing in which the extracellular matrix was laser-ablated anteriorly.

## Relationship between tissue stress and cell elongation

We first investigated stresses present in the tissue. In an elastic tissue, tissue deformations stem from cell shape changes and cell elasticity is responsible for tissue elasticity. For small deformations, Hooke's law states that the mechanical stress in the material is proportional to its deformation. We write the isotropic part of the stress

$$P = -\bar{K} \ln \frac{a}{a_0}, \qquad (3)$$

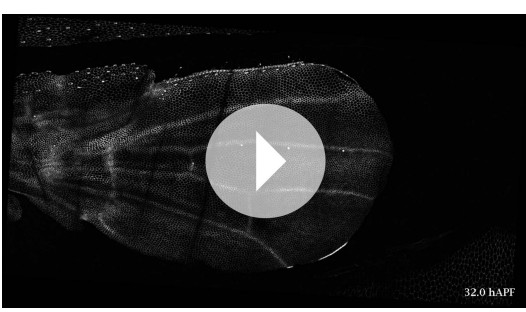

**Video 9.** Video of a WT wing in which the extracellular matrix was laser-ablated distally.

where $P$ denotes two-dimensional tissue pressure, $a$ and $a_0$ are cell area and preferred cell area respectively, and $\bar{K}$ is the area compressibility. The preferred cell area $a_0$ changes when cells divide. For small $(a - a_0)/a_0$, *Equation 3* corresponds to Hooke's law. *Equation 3* implicitly contains the active contribution to pressure, which influences the preferred cell area $a_0$ (see Appendix 2, 'Constitutive equation for the tissue stress').

We now focus on the anisotropic part of the stress $\tilde{\sigma}$, also called the shear stress. For simplicity, we write the elastic anisotropic stress in the form of Hooke's law $\tilde{\sigma}_e = 2KQ$, where $K$ is a shear elastic modulus and $Q$ is the cell elongation. In this expression, the cell shape is isotropic in the absence of anisotropic stresses. However, in tissues, planar polarized cells may spontaneously elongate. Therefore, we postulate the following constitutive equation for the anisotropic tissue stress

$$\tilde{\sigma} = 2KQ + \zeta, \tag{4}$$

where $\zeta$ is a tensor that can be interpreted as a cell autonomous active stress related to spontaneous cell elongation.

To test whether *Equation 4* accounts for anisotropic stresses present in the tissue, we further analyzed the circular laser ablations performed in different regions of WT and *dumpy^{ov1}* wings at 22 hAPF (*Figure 2I,I'*). We defined a shear rate $\tilde{v}^{cut}$ that characterizes the anisotropic recoil of the circular cut boundary into an elliptic shape (see 'Materials and methods', Analysis of circular laser ablations). We plotted the projection of this shear rate on the PD axis $\tilde{v}^{cut}_{xx}$ as a function of the projected average cell elongation $Q_{xx}$ (*Figure 9A*). Cell elongation was determined as the average cell elongation within the corresponding region in unperturbed wings. We found that the anisotropic part of the shear rate varied linearly with cell elongation (*Figure 9A*). The positive slope of this linear relation indicates that the shear modulus $K$ is positive. In this argument, we use the recoil shear rate $\tilde{v}^{cut}$ as a measure proportional to tissue stress. A linear fit to this data also has a positive intercept, corresponding to a positive $\zeta_{xx}$ in *Equation 4*. This implies that wing blade cells would spontaneously elongate along the AP axis in the absence of stress. Equivalently, in the absence of cell deformation ($Q = 0$) wing blade cells are exerting higher contractile stress in the PD axis than in the AP axis. The relative contributions of elastic stress and stress associated with spontaneous cell elongation can be quantified from this linear fit. The ratio of the intercept and the slope of the linear fit equals $\zeta_{xx}/2K$, which leads to the estimate $\zeta_{xx}/K = 0.333 \pm 0.003$ in WT and $\zeta_{xx}/K = 0.316 \pm 0.004$ in *dumpy^{ov1}* wing. Experimental data obtained in WT and *dumpy^{ov1}* wings fall on similar lines (*Figure 9A*), suggesting that internal mechanical properties of the tissue are not perturbed in *dumpy^{ov1}* mutant wings. Since *dumpy* mutant cells are less elongated than those of WT (*Figure 9—figure supplement 1A*), their similar mechanical properties imply that they are under less anisotropic stress consistent with the loosened connections to the overlying cuticle.

If we perform the same analysis for the isotropic part of the tissue stress, we use the area expansion rate of the cut circle during recoil as a proxy for negative tissue pressure $-P$. Plotting this rate as a function of average cell area in a given region, we find that $a_0$ is smaller than cell area $a$, which reveals that the tissue is under contractile tension or negative P (*Figure 9B*). This is consistent with the observed tissue area contraction in *dumpy^{ov1}* mutant and severed wings. However, parameter values $a_0$ and $\bar{K}/K$ cannot be reliably estimated by this method because the average cell area $a$ varies too little, and because cell divisions may introduce heterogeneity in the preferred cell area $a_0$.

Overall, laser ablation experiments indicate that there are two contributions to anisotropic stress in the PD axis. First, blade cells are elastically deformed along the PD axis, in response to hinge contraction and margin attachments. Second, these cells would tend to spontaneously shorten in the PD axis even in the absence of external stresses.

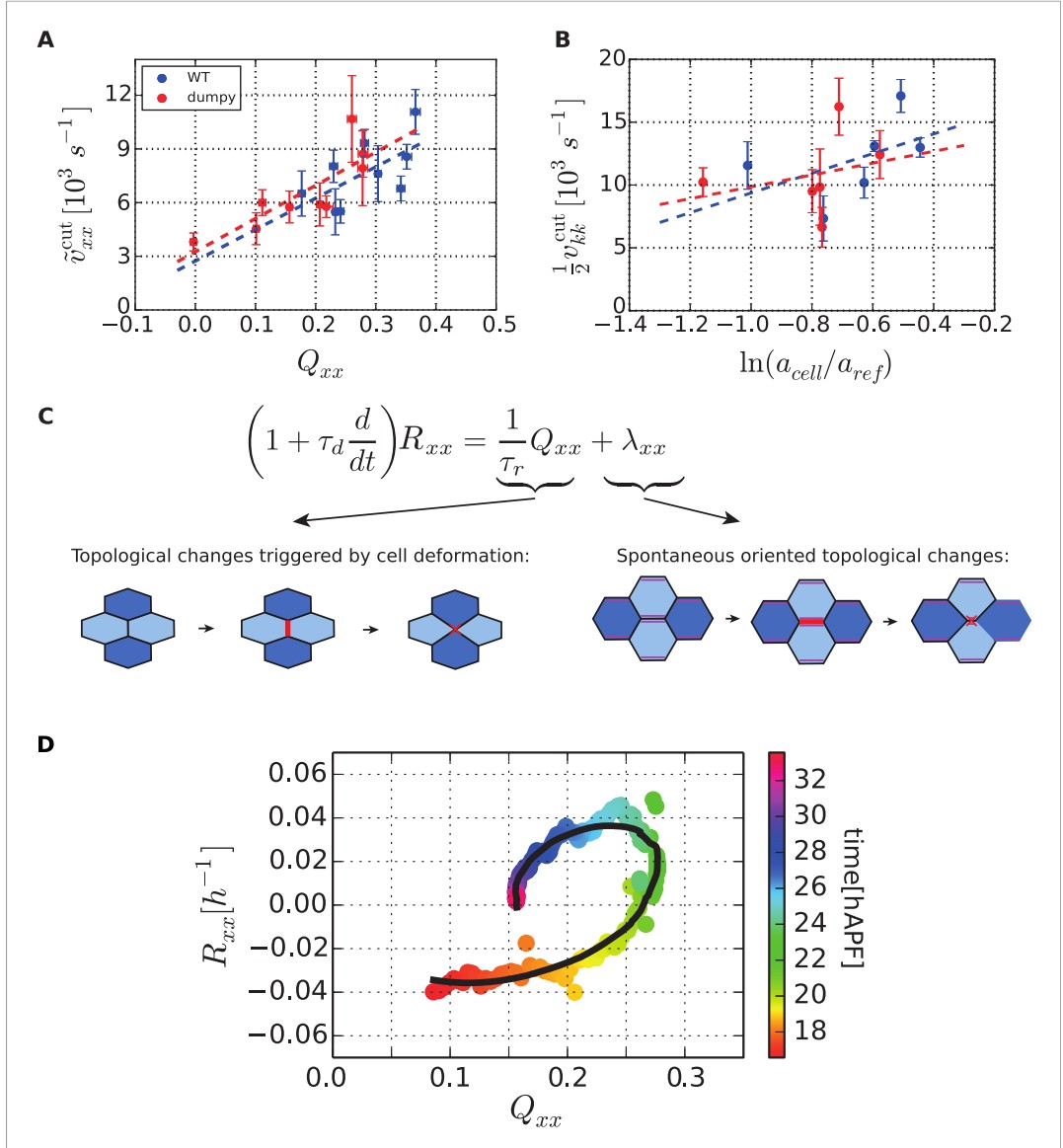

**Figure 9**. Dependency of stresses and topological changes on cell deformation. (**A**) Anisotropic recoil of circular cut boundaries after laser ablation of the blade at 22 hAPF, as a function of the projected average cell elongation in the region of the cut, for WT and *dumpy*$^{ov1}$ wings. (**B**) Isotropic recoil of circular cut boundaries after laser ablation of the blade at 22 hAPF, as a function of the average cell area in the region of the cut, for WT and *dumpy*$^{ov1}$ wings. (**C**) Topological changes are driven either by cell elongation or by polarity-dependent processes, with a delay $\tau_d$. (**D**) Shear due to topological changes as a function of cell elongation in the blade. Experimental points are color-coded according to time. The black line is a fit of *Equation 5* shown in (**C**) to the experimental data. Due to a delay $\tau_d$ in the response of topological changes to cell elongation, data points follow a spiraling curve during wing morphogenesis.

The following figure supplements are available for figure 9:

**Figure supplement 1**. Cell elongation state.

**Figure supplement 2**. Shear due to topological changes as a function of cell elongation in the blade for 6 analyzed wings.

## Model for the dynamics and the orientation of topological changes

So far, we have characterized the elastic properties of the tissue. On long time scales, elastic stresses can be relaxed by topological changes of the cellular network generating tissue shear viscosity. We therefore now develop a model describing the average rate and orientation of tissue shear $R$ due to such topological changes. Although $R$ includes the shear caused by T1 and T2 transitions, cell divisions and correlation effects, it is dominated by T1 transitions (*Figure 6A,B*). We asked how the shear rate $R$ due to topological changes is regulated in the wing blade. If cells in the tissue are elongated, this elongation could drive topological changes that give rise to the shear rate $R = Q/\tau_r$. Here, $\tau_r$ can be interpreted as the time during which cell elongation relaxes. Indeed, $\tau_r$ is also the time beyond which the tissue behaves as a viscous fluid with viscosity $K\tau_r$, due to elastic stress relaxation by topological changes including cell rearrangements and cell divisions. In addition, our observations of laser-severed wings suggest that the response of $R$ to cell elongation is not instantaneous, but follows with a time delay $\tau_d$ of a few hours (see previous discussion and *Figure 8—figure supplement 3*), which also must be incorporated in the theory.

In addition to elongation-induced rearrangements, planar polarized tissues may undergo oriented topological changes even in the absence of cell elongation. For instance, topological changes could be driven autonomously by increased contractility of certain bonds. Taking into account all these contributions, we postulate the following relation between the shear rate $R$ and cell elongation (*Figure 9C*):

$$\left(1 + \tau_d \frac{d}{dt}\right) R = \frac{1}{\tau_r} Q + \lambda.$$

(5)

Here, the delayed response to cell elongation is introduced via the time derivative of the shear $R$. This way of implementing a delay implies that the current value of cell elongation has the strongest impact on shear due to topological changes and the effect of recent values of cell elongation fades exponentially over time, disappearing beyond the time $\tau_d$. Shear driven by polarity dependent processes is characterized by the tensor $\lambda$. Thus, even if cells are not elongated ($Q = 0$), $\lambda$ drives shear due to topological changes. For shear along the AP axis $\lambda_{xx} < 0$, while for shear along the PD axis $\lambda_{xx} > 0$.

In order to verify whether *Equation 5* captures the dynamics of the shear created by topological changes during pupal wing morphogenesis, we plotted $R_{xx}$ vs the cell deformation component $Q_{xx}$ for different times (*Figure 9D*). For an instantaneous response of $R$ (no delay, $\tau_d = 0$) these points would fall on a straight line. For non-zero delay, the history of the process matters and these points follow a curve spiraling towards a fixed point (see Appendix 2, 'Effect of delay in topological changes'). We find indeed that experimental data for each WT wing follow a spiral, confirming the existence of a delay. A fit of the theory to the data allows us to estimate the coefficients in *Equation 5* ($\tau_d = (3.7 \pm 0.9)$ h, $\tau_r = (1.8 \pm 0.6)$ h and $\lambda_{xx} = (-0.10 \pm 0.04)$ h$^{-1}$). Interestingly, $\lambda_{xx}$ is negative, indicating that polarity driven topological changes create AP shear—consistent with conclusions from laser ablation experiments. Polarity driven AP-oriented topological changes may be related to increased tension of PD-oriented as compared to AP-oriented cell bonds. This anisotropy of cell bond tension could account for the positive value of $\zeta$ in *Equation 4* because it would tend to elongate cells in the AP-axis.

The fact that a constant negative value of $\lambda_{xx}$ accounts for the experimental data suggests that the tendency to undergo polarity-driven topological changes exists during both phase I and II. The transition from phase I to phase II occurs when cell elongation-driven topological changes begin to exceed the effect of polarity-driven topological changes. During phase I topological changes are largely polarity driven along the AP axis. *Equation 2* then implies that these topological changes, together with hinge contraction, contribute to the buildup of cell elongation in the PD axis.

Once flows have stopped, the value of $\lambda_{xx}$ determines the final value of cell elongation. Indeed, *Equations 2, 5* predict that in a steady state without shear flows ($R = 0$, $dQ/dt = 0$), the cell elongation is given by $Q = -\tau_r \lambda$. We can therefore use the cell elongation at the end of the videos, where the tissue is almost stationary, to estimate $\tau_r \lambda_{xx}$. In other words, the final cell elongation should be independent of external stresses. Instead final cell elongation is governed only by the internal dynamics of the tissue, that is, the tendency to undergo AP-oriented topological changes $\lambda_{xx}$ as well as the stress relaxation time-scale $\tau_r$. Interestingly, examining different perturbed conditions, we find that the final cell elongation and hence $\lambda_{xx}$ depend on cell area (*Figure 9—figure supplement 1B* and Appendix 4, 'Area dependence of $\lambda_{xx}$').

Overall, we find that the shear created by topological changes is driven in part by cell elongation and in part by cell polarity-dependent processes. Furthermore, topological changes respond to cell elongation with a delay of several hours.

*Equations 1–5* constitute a full theory for tissue mechanics taking into account cell shape changes. We now want to test whether this tissue description quantitatively accounts for cell and tissue shape changes during wing morphogenesis.

## Cell shape changes and tissue flows during pupal development can be understood by a continuum mechanical model

We ask here whether the tissue and cell properties described by our equations can quantitatively account for the observed changes in the shapes of hinge and blade as well as the elongation and topological changes of their constituent cells. We use a simplified description of the wing where the hinge and blade are represented by two rectangles that are attached to each other. These rectangles have the same areas as the hinge and blade and they undergo pure shear that is the average shear in the respective parts of the wing. To describe tissue flow, *Equations 1–5*, which characterize local tissue properties, have to be complemented by the condition of force balance. At the boundary of the tissue, elastic linkers connected to the cuticle impose external forces that have to balance tissue stresses. In our model the rectangles are therefore connected to an external frame by elastic elements (*Figure 10A* and *Figure 10—figure supplement 1*). These external elements correspond to the extracellular matrix and the frame corresponds to the cuticle. The elastic elements provide resistance to extension of blade and hinge along the PD and AP axes (see Appendix 3, 'Boundary stresses').

In each rectangle, anisotropic stresses and shear are described by *Equations 4, 5*, with different tissue parameters in hinge and blade. Isotropic internal stresses in the blade are described by *Equation 3*. Preferred cell area $a_0$ changes rapidly when cells divide and is modulated by the cell contractility $\bar{\zeta}$ (see Appendix 2, 'Constitutive equation for the tissue stress'). In the hinge, we do not solve the full mechanical problem but rather impose the observed hinge area contraction. We do so by adjusting the pressure in the hinge to match the observed hinge area.

We start with rectangles whose areas and aspect ratio are consistent with those of hinge and blade at 16 hAPF (see Appendix 3). The initial conditions for $\boldsymbol{Q}$ and $\boldsymbol{R}$ are the observed average cell elongation and the initial shear rate due to topological changes. To trigger cell flows in the model, we turn on active stresses $\zeta$, $\bar{\zeta}$ and polarity dependent topological changes $\lambda$. We solved *Equations 1–5* in two rectangles that correspond to hinge and blade. Note that the pressure in the hinge is not determined by *Equation 3*, but by imposing hinge area. In the blade, we used the measured rates of cell division and T2 transitions to calculate the blade area changes (*Equation 1*).

We normalized all elastic moduli and friction coefficients to the blade shear modulus $K$. The solutions $\boldsymbol{Q}(t)$, $a(t)$, $v(t)$, and $\tilde{\boldsymbol{v}}(t)$ of *Equations 1–5* depend on a set of parameters characterizing the tissue both in hinge and blade (see *Tables 1, 2*). In addition, elastic coefficients describe constraints imposed by linkers at the boundary. Friction due to motion with respect to the cuticle is captured by a friction coefficient $\gamma$. Furthermore, to fully account for the observed flows, we found that we needed to add a contribution of tissue area viscosity $\bar{\eta}$ to *Equation 3* (see Appendix 2, 'Constitutive equation for the tissue stress', *Equation 52*). We include this term in all subsequent calculations. Laser ablation of the extracellular matrix is introduced in our model by removing the elastic linkers at the tissue boundary at the side of ablation.

We first analyzed WT wings in both unperturbed and mechanically perturbed conditions where only the extracellular matrix was ablated. Because blade tissue was not damaged by matrix ablation we expected that most parameters characterizing the hinge and blade would be identical to those in unperturbed wings. However, $\lambda_{xx}$ can differ in wings characterized by different cell areas, as noted above. We therefore used the values of $\tau_r$, $\tau_d$ and $\lambda_{xx}$ determined by jointly fitting *Equation 5* to the data (see *Table 1* and *Figure 9—figure supplement 2*). In addition, we used experimentally determined values of $\zeta/K$ estimated from circular laser ablation experiments in WT wing blades (*Figure 9A*). A similar fit is performed for the hinge and corresponding parameters are superscripted with an index H.

This leaves 10 parameters unspecified. These characterize the isotropic stresses in the blade, the anisotropic stresses in the hinge, the stiffness of the elastic linkers at the boundary, and the coefficient describing external friction (*Table 2*). We estimated these parameters by performing joint fits of *Equations 1–5* to the quantified time dependence of tissue area changes and shear in unperturbed

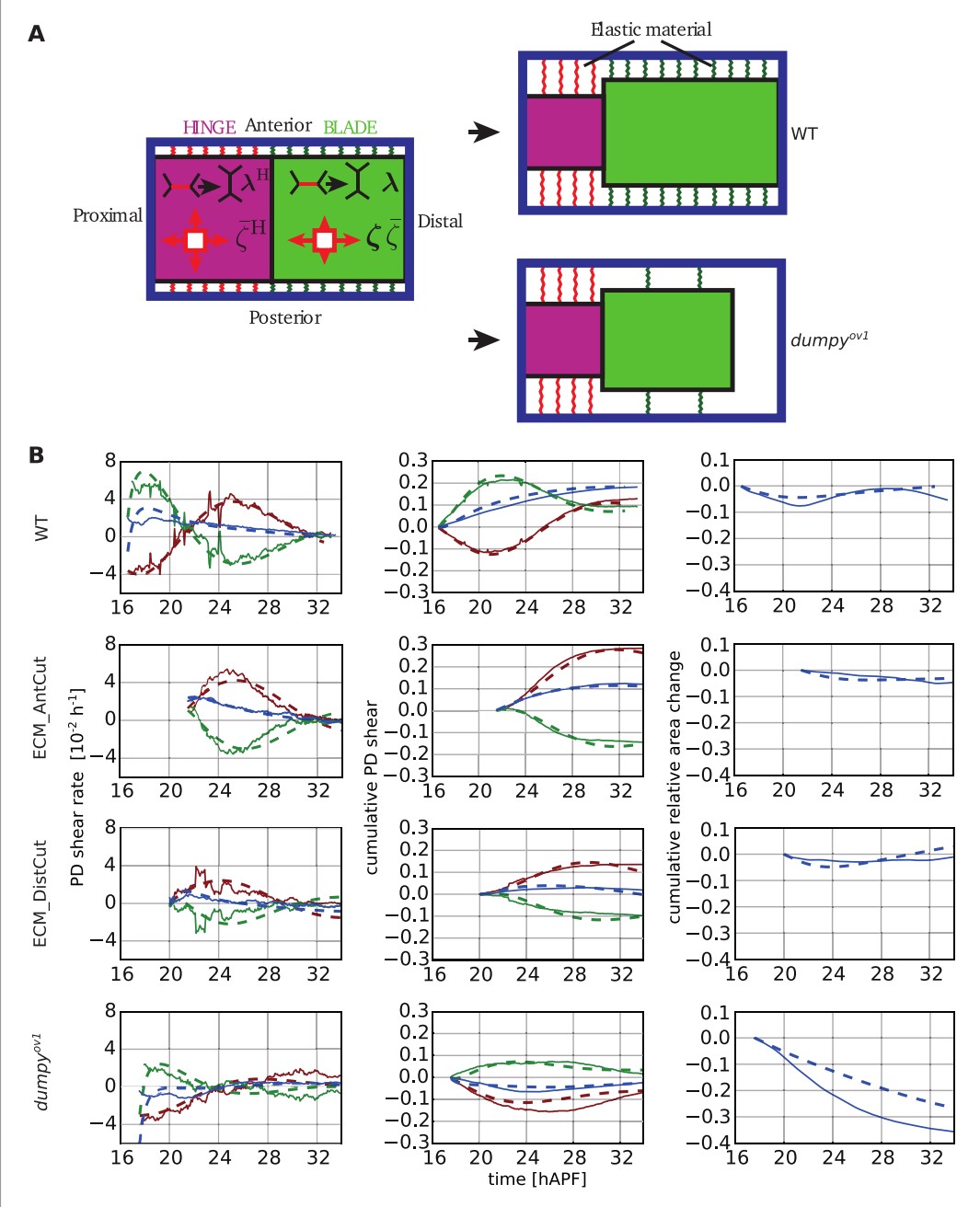

**Figure 10**. Continuum mechanical model of wing morphogenesis. (**A**) Schematics of the wing model: the hinge and blade are represented by rectangles. Within each rectangle, the tissue is subjected to cell-autonomous anisotropic and isotropic stresses $\zeta$ and $\bar{\zeta}$, and to topological changes driven by cell polarity-dependent processes $\lambda$. The complex elastic material connecting the wing to the cuticle is represented by AP-oriented elastic links (green and red springs on the cartoon) and PD-oriented springs (see *Figure 10—figure supplement 1*). In WT wings, the blade distal end is fixed, while it is free to move in the *dumpy^{ov1}* mutant. (**B**) (Left) Experimental (solid line) and theoretical (dashed line) time courses of tissue shear rate (blue curves), cell elongation change (green curves) and shear due to topological changes (red curves), in the blade and along the PD axis. (Middle) Experimental and theoretical time courses of cumulative tissue shear (blue curve), cell elongation (green curve) and cumulative shear due to topological changes (red curves), in the blade and along the PD axis. (Right) Experimental and theoretical cumulative relative blade area change. Model parameters were obtained by a fitting procedure to experimental data (*Tables 1, 2*). The continuum mechanical model recapitulates cell shape changes and tissue flow during wing morphogenesis.

*Figure 10. continued on next page*

*Figure 10. Continued*

The following figure supplements are available for figure 10:

**Figure supplement 1**. Schematics of the rectangle model.
**Figure supplement 2**. Continuum mechanical model in the hinge.

and mechanically perturbed WT wings. We found that anisotropic stresses in the hinge are small compared to isotropic stresses and therefore we could set $\zeta^H$ and $K^H$ to zero. A single set of the remaining 8 parameters accounted well for all time dependent data (see *Figure 10B* and *Figure 10—figure supplement 2* and *Table 2*). It is remarkable that this very simplified model can account for the main features of the complex shape changes that occur during pupal wing morphogenesis. The choice of rectangles indeed captures the average shear and area change that determine tissue shape change. Displaying the time evolution of the rectangles within a time-lapse video of the pupal wing reveals the close agreement between the calculated and observed shape changes of hinge and blade (see *Videos 10–13*).

Our experimental data suggest that distal elastic connections to the cuticle are weakened in the *dumpy^{ov1}* mutant wing. We therefore asked whether we could fit the rectangle model described above using WT tissue parameters but allowing parameters describing connections to the cuticle to change. We removed distal boundary connections to the blade (*Figure 10—figure supplement 1*) as suggested by laser ablation experiments and allowed other parameters characterizing external linkers to change. We used these values and the measured value of $\zeta/K$ (which is almost the same as in WT, see *Figure 9A*) in fits of the full dynamics of the rectangle model to the *dumpy^{ov1}* mutant data (see *Figure 10B* and *Figure 10—figure supplement 2*). These calculations show that distal and lateral connections to the wing blade margin are weakened, as expected (see *Table 2*).

Overall, the continuum theory of epithelia outlined in *Equations 1–5* together with appropriate boundary conditions recapitulates the dynamics of cell and tissue shape over 16 hr of morphogenesis. By fitting the theory to experimental measurements, we determine the values of tissue parameters characterizing intrinsic tissue time-scales, cell autonomous stresses, tissue elastic moduli and elastic moduli of external linkers. Using this theory to investigate mechanical parameters in a *dumpy^{ov1}* mutant wing, we confirm the existence of Dumpy-dependent elastic connections between the wing and cuticle. Thus, this theoretical approach is a powerful tool for studying how mutations influence specific aspects of cell and tissue mechanics.

## Discussion

Developing tissues are active viscoelastic materials that generate and respond to mechanical stresses to change their shapes. How tissue material properties emerge from the properties and behavior of

**Table 1**. Coefficients describing tissue properties in **Equation 5** are fitted with single values of time-scales $\tau_r$ and $\tau_d$ in hinge and blade while $\lambda_{xx}$ was allowed to vary among different wings (see **Figure 9—figure supplement 2**)

| | | WT #1 | WT #2 | WT #3 | ECMAntCut | ECMDistCut | DP |
|---|---|---|---|---|---|---|---|
| Blade | $\tau_r$ [h] | $1.7 \pm 0.1$ | | | | | |
| | $\tau_d$ [h] | $4.2 \pm 0.3$ | | | | | |
| | $\lambda_{xx}$ [h$^{-1}$] | $-0.11 \pm 0.01$ | $-0.11 \pm 0.01$ | $-0.10 \pm 0.01$ | $-0.10 \pm 0.01$ | $-0.068 \pm 0.007$ | $-0.094 \pm 0.008$ |
| Hinge | $\tau_r^H$ [h] | $4.6 \pm 2$ | | | | | |
| | $\tau_d^H$ [h] | $2.4 \pm 1$ | | | | | |
| | $\lambda_{xx}^H$ [h$^{-1}$] | $-0.05 \pm 0.01$ | $-0.05 \pm 0.01$ | $-0.04 \pm 0.01$ | $-0.03 \pm 0.01$ | $-0.01 \pm 0.01$ | $-0.04 \pm 0.01$ |

WT; wild type.

**Table 2.** Parameters of the rectangle model are divided in three groups describing blade tissue properties, hinge tissue properties and external links

| | | | | WT | ECMDistCut | ECMAntCut | Dp |
|---|---|---|---|---|---|---|---|
| Tissue | blade | cell autonomous shear stress | $\zeta_{xx}/K$ | 0.333 ± 0.003 | | | 0.316 ± 0.004 |
| | | shear elastic modulus | $K/K$ | 1 | | | |
| | | cell area contractility | $\bar{\zeta}/K$ | 0.05 ± 0.03 | | | |
| | | area elastic modulus | $\bar{K}/K$ | 2.07 ± 0.09 | | | |
| | | area viscosity coefficient | $\bar{\eta}/K[h]$ | 49 ± 2 | | | |
| | hinge | cell autonomous shear stress | $\zeta_{xx}^H/K$ | 0 | | | |
| | | shear elastic modulus | $K^H/K$ | 0 | | | |
| External links | blade | effective AP elastic constant | $kL_0/K$ | 0.5 ± 0.1 | | 0 | 0.005 ± 0.007 |
| | | effective PD elastic constant | $k_{PD}/K$ | 4.91 ± 0.04 | | | 5.3 ± 0.2 |
| | | friction coefficient | $\gamma/K[h]$ | 21.3 ± 0.8 | | | 22.1 ± 0.6 |
| | | distal connections | – | Yes | No | Yes | No |
| | hinge | effective AP elastic constant | $k_{PD}^H/K$ | 67.8 ± 0.4 | | | 78 ± 2 |
| | | effective PD elastic constant | $k_{PD}^H/K$ | 9.50 ± 0.07 | | | 16.8 ± 0.6 |
| | | friction coefficient | $\gamma/K[h]$ | 21.3 ± 0.8 | | | 22.1 ± 0.6 |

Cell autonomous shear stress in wing blade of WT and *dumpy^{ov1}* are determined from circular laser cut experiments. Unperturbed and mechanically perturbed WT wings are first simultaneously fitted using results listed in **Table 1**. Then, the *dumpy^{ov1}* wing is fitted keeping the values of hinge and blade tissue parameters the same as in WT. The effective anterior-posterior (AP) and PD elastic constants describe effects of external elastic elements providing resistance to changes in size of blade and hinge along the AP and PD direction. All quantities are normalized by the elastic shear modulus of the blade tissue $K$. Quantities containing spatial dimensions are also normalized by the initial length $L_0$ of the WT wing. Uncertainties reported for the parameters in this table (expect for the cell autonomous shear stress $\zeta_{xx}$) were determined by the fit. Note that they do not reflect uncertainties arising from approximations made in the rectangle model (supplement section 4) and from pre-processing of experimental data (supplement section 1.6).

their many constituent cells, and how these properties quantitatively account for tissue shape change is a major question in developmental biology. These questions have been difficult to answer because of both experimental and theoretical limitations. Methodologies for large-scale imaging and image analysis of entire tissues at the cellular level were insufficient, and direct measurement of forces and stresses in vivo have been rare (*Hutson et al., 2003*; *Farhadifar et al., 2007*; *Rauzi et al., 2008*; *Bosveld et al., 2012*; *Bambardekar et al., 2015*). Furthermore, theoretical approaches are required to understand how tissue shear emerges from different kinds of cellular processes. By developing new experimental and theoretical approaches, we have been able for the first time to account for the shape change of a developing tissue on the basis of its active and its viscoelastic properties. Our approach allows us to quantitatively understand how such tissue properties emerge from the interactions of its constituent cells.

Our work is stimulated by previous approaches to tissue dynamics. For example, previous work on tissue tectonics highlighted the decomposition of tissue shear in cell shape changes and cell intercalation (*Brodland et al., 2006*; *Graner et al., 2008*; *Marmottant et al., 2008*; *Blanchard et al., 2009*; *Butler et al., 2009*). Our triangle method refines these ideas using a geometrical scheme that defines and distinguishes exact cellular contributions to tissue shear. These are contributions due to cell deformation, cell division, T1 and T2 transitions.

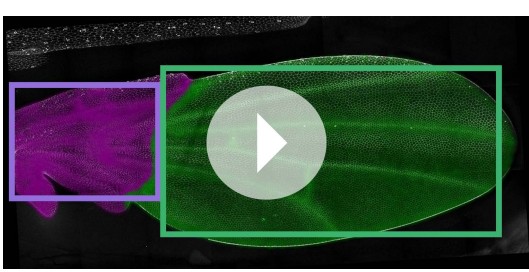

**Video 10.** Time evolution of the rectangles obtained from the rectangle model in a WT wing.

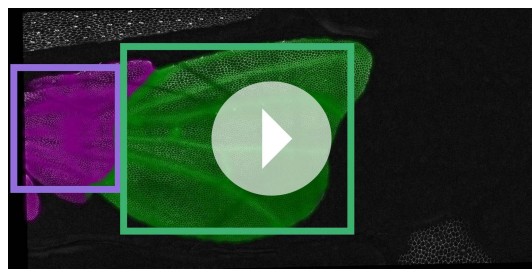

**Video 11.** Time evolution of the rectangles obtained from the rectangle model in a *dumpy^{ov1}* wing.

This approach has also led to the discovery of a new correlation effect that contributes significantly to tissue shear. This collective effect can result from local correlations between cell elongation and tissue rotation, or between cell elongation and cell area changes. For instance, this correlation effect can be produced by relative sliding of neighboring cell rows (*Figure 4H*).

In our work we combine this approach with a novel continuum mechanical model based on patterns of cell elongation. Our model combines visco-elastic material properties with additional cell autonomous so-called active stresses. These active stresses contribute to force balances and also can drive topological rearrangements. This model allows us to quantitatively discuss tissue deformations that emerge from the dynamic interplay of externally applied forces, internally generated stresses and the resulting collective cell rearrangements. Our model is related to continuum models of active gels and tissues (*Kruse et al., 2005*; *Brodland et al., 2006*; *Ranft et al., 2010*) that can be obtained by coarse-graining cell-based models (*Honda et al., 1984*; *Graner and Glazier, 1992*; *Drasdo et al., 1995*; *Chen and Brodland, 2000*; *Brodland et al., 2007*; *Farhadifar et al., 2007*; *Basan et al., 2011*). However, our model differs from existing continuum models in several ways: it is directly based on measurable cellular contributions to tissue shear, it includes a new time scale corresponding to a delay for the generation of T1 transitions, and it introduces a cell-autonomous contribution to T1 transitions. Such cell-autonomous T1 transitions (*Irvine and Wieschaus, 1994*) may for example be generated by orientation-dependent contractility of cell boundaries (*Bertet et al., 2004*; *Skoglund et al., 2008*; *Rauzi et al., 2010*; *Sawyer et al., 2011*). Overall, our approach can bridge scales from individual cellular events to cell flows and large-scale tissue shape changes.

We find that the wing tissue shapes itself through patterned contractions and shear that occur in the context of patterned extra-cellular matrix (ECM) connections to a surrounding cuticular scaffold. Thus, by pulling on these connections, the tissue forces itself into the right shape. In *dumpy* mutants, these connections are weakened and the final tissue shape is dramatically altered as compared to WT (*Waddington, 1940*; *Carmon et al., 2010*). Dumpy plays a similar role in shaping the *Drosophila* trachea in embryonic development (*Dong et al., 2014*). The mechanical function of such ECM connections may be generally important during epithelial morphogenesis in vivo.

Our quantitative analysis has provided new insights into the biology of pupal wing morphogenesis. We have discovered that cell-autonomous planar polarized T1 transitions occur early, during the first phase of the process. Surprisingly, they occur along the AP axis and actively increase cell elongation in the PD axis. Why were these AP-oriented T1 transitions not detected in our previous analysis (*Aigouy et al., 2010*)? Comparing the rate of T1 transitions with the amount of shear they generate shows that the orientation of T1 transitions at this time is less focused than it is later when T1 transitions cause shear in the PD axis. However, because their overall rate is higher in phase I, a slight bias in their orientation causes significant AP shear. The fact that there is only a slight AP bias in these T1 transitions likely explains why they have not been detected before, and highlights the importance of the large-scale quantitative analysis that we employ here. What mechanisms might underlie AP-oriented T1 transitions that occur in phase I? Quantitative analysis of laser ablation data indicates that the preferred shape of wing epithelial cells in the absence of external forces is not isotropic, but rather elongated in the AP axis. Increased contractility of PD-oriented cell boundaries could explain this preference, and would also

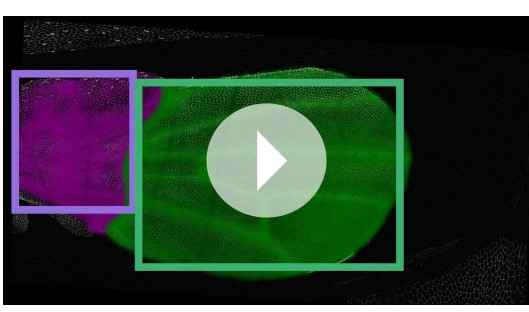

**Video 12.** Time evolution of the rectangles obtained from the rectangle model in a WT wing where the extracellular matrix was ablated distally at the onset of phase II.

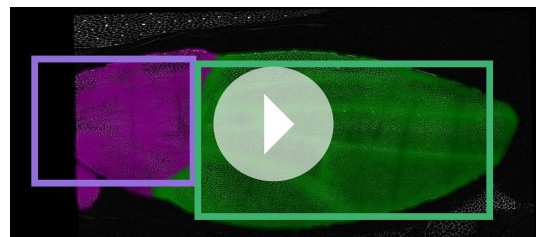

**Video 13.** Time evolution of the rectangles obtained from the rectangle model in a WT wing where the extracellular matrix was ablated anteriorly at the onset of phase II. Note that length of the hinge is increased by a correction term described in Appendix 4, 'Fitting of the rectangle model to cell and tissue shape in the hinge and blade'.

tend to favor AP-oriented T1 transitions. We note that components of both the Fat and Core planar cell polarity pathways are enriched on PD-oriented cell boundaries during phase I (*Merkel et al., 2014*). These systems are known to regulate both cell boundary tension and Cadherin turnover (*Classen et al., 2005*; *Rogulja et al., 2008*; *Mao et al., 2011*; *Bosveld et al., 2012*; *Warrington et al., 2013*; *Nagaoka et al., 2014*), and it will be interesting to investigate their involvement in orienting T1 transitions at this stage.

Our theoretical analysis suggests that AP-oriented T1 transitions are unlikely to strongly influence the final shape of the wing. However, they have a strong influence on peak cell elongation and final cell shape. What function could this serve during morphogenesis? One possibility is that AP-oriented T1 transitions influence the morphology of cuticular ridges on the adult wing surface. The cuticle of the adult wing is shaped in a reproducible pattern of ridges that form between cell rows (*Doyle et al., 2008*; *Merkel et al., 2014*). These ridges likely influence anisotropic mechanical properties of the wing during flight. Orderly packing of the wing epithelium into hexagons is necessary for the long-range order of these ridges (*Merkel et al., 2014*), and their spacing presumably depends on the spacing of cell rows at the time the adult cuticle is secreted. AP-oriented T1 transitions could influence both these features. First, they are predicted to increase the final magnitude of PD cell elongation and may therefore influence ridge spacing. Second, increased cell elongation contributed by AP-oriented T1 transitions may help focus the direction of the subsequent PD-oriented T1 transitions and increase hexagonal order. Interestingly, PCP mutant wings are characterized by irregular cell packing geometry (*Classen et al., 2005*), and less long-range order in the pattern of cuticular ridges (*Merkel et al., 2014*).

Our analysis has revealed unexpected properties of the cell elongation-dependent shift of T1 transitions towards the PD axis. Separately quantifying the orientations of cell boundary loss and cell boundary expansions during this shift shows that these two components of a T1 transition respond differently to cell elongation. The average orientation of cell boundary expansion shifts from the AP to the PD axis at 21 hAPF before the complementary change in orientation of cell boundary loss (*Figure 7*). This suggests that PD-oriented epithelial stresses promote expansion of cell boundaries along the PD axis at a lower threshold than is required to block shortening of PD-oriented boundaries. Thus, the mechanisms that operate to contract cell boundaries and to expand new ones respond differently to epithelial stresses. This difference means that cells tend to both gain and lose boundaries along the PD axis during the shift from AP to PD-oriented T1 transitions. Interestingly, this does not always simply represent the contraction and re-expansion of the same cell boundaries. Rather, contractions and expansions of different cell boundaries in the same direction occur as rows of cells move with respect to each other—a process that is associated with the maximal contribution of measured correlation effects (compare *Figure 6A* magenta line and *Figure 7*).

PD-oriented cell divisions also contribute to the shape change of the wing blade during pupal morphogenesis. Interestingly, while epithelial stresses promote cell division in cultured epithelial cells (*Puliafito et al., 2012*; *Streichan et al., 2014*), this does not appear to be the case in the wing. In fact cell divisions occur at a similar rate and for longer times in laser ablated and *dumpy*$^{ov1}$ mutant wings. Thus, there are more cell divisions overall in situations where stresses are reduced. This suggests that the cell divisions that occur normally are not initiated by the stresses associated with hinge contraction but are controlled by other factors. Examining their pattern suggests that signals from veins may play a role (data not shown, [*Garcia-Bellido et al., 1994*]). It is also possible that hormonal signals such as ecdysone could control their timing. We do not yet understand why perturbations that reduce stress result in slightly more cell divisions. These extra divisions may occur by different mechanisms that respond to wounding.

Furthermore, the magnitude of PD shear caused by cell divisions is proportional to the rate of cell divisions, suggesting that cell division orientation is also unaffected by reducing PD stresses.

Thus, it may be that cell divisions in the pupal wing are autonomously controlled by planar polarized cortical cues. Alternatively, because the AP-oriented T1 transitions still cause some PD cell elongation in laser severed and dumpy[ov1] mutant wings, it may be that residual cell elongation is still sufficient to orient cell divisions.

During development, tissues are shaped with extreme reproducibility. It has been estimated that the shape of the *Drosophila* wing is precise to within 1 cell diameter (*Abouchar et al., 2014*). How is such reproducibility achieved? The cell lineage in the wing is indeterminate, and both wing shape and wing area can be preserved in the face of a variety of developmental insults—including cell death and differential growth rates. This requires that all cells within the wing be able to sense its overall size and shape. How do cells acquire this information? Based on our findings, we would propose that stresses in the wing epithelium could be one cue.

Interestingly, the total area and shear of WT wings is more reproducible than would be expected from the variation in individual cellular contributions (*Figures 3, 6*). Thus, these cellular events influence and can compensate for each other. The fact that epithelial tension is required to maintain area homeostasis and to control tissue shear suggests a mechanism for this compensation. Variations in cell division, cell shape change, cell rearrangements and cell extrusions could be detected by cells as changes in epithelial tension. The ability of cells to sense and respond to tension could underlie this compensation and produce wings of reproducible sizes and shapes.

# Materials and methods

## Fly strains and crosses

Flies were raised at 25°C under standard conditions unless stated otherwise. Pupae were collected for imaging as described previously (*Classen et al., 2008*). Ecad::GFP flies (*Huang et al., 2009*) were used as control for all live imaging experiments. The dumpy[ov1] mutation (*Sturtevant et al., 1929*) was recombined with Ecad::GFP for imaging, and for quantifying tissue flows and cell behaviors in a dumpy[ov1] mutant (Bloomington, reference number 276). The Dumpy::YFP protein trap line (DGRC) was used to describe Dumpy distribution in the pupal wing. To inhibit cell divisions in the pupal wing while imaging it, we used the thermosensitive allele $cdc2^{E1-E24}$ of $cdk1$ that we recombined with Ecad:: GFP (*Stern et al., 1993*). Cells expressing two copies of $cdc2^{E1-E24}$ and shifted to 30°C arrest in G2 phase just prior to entering mitosis (*Weigmann et al., 1997*).

## Long-terms time-lapse imaging

Pupae were prepared for live imaging as previously described (*Classen et al., 2008*). Long-term time-lapses were acquired with a Zeiss spinning disk microscope driven by the Axiovision software and equipped with an inverted Axio Observer stand, a motorized xyz stage, and a Zeiss LCI Plan-Neofluar 63× 1.3 NA Imm Corr lens associated to an objective heater set to 25°C. The fly pupa was placed in a temperature-controlled chamber set to 25°C and equipped with a humidifier to prevent desiccation. Images were recorded with an AxioCamMR3 camera (2 × 2 binning). Laser power was measured through a 10×/0.45 NA lens using a power-meter (PT9610, Gigahertz-Optik). A power of 0.980 mW was found to be optimal to prevent noticeable bleaching during 24 hr of continuous acquisition with an exposure time of 265 ms. Briefly, the dorsal cell layer of the pupal wing was scanned within 5 min in the (x, y, z) dimensions, over 24 overlapping positions of about 30 z-sections each. This scan procedure was continuously repeated about 260 times to cover more than 20 hr of development. Custom Fiji macros helped keeping the tissue in focus in a semi-automated way.

## Data handling and image processing

We benefited from the flexible architecture provided by our computer department to handle several TB of data using custom unix bash scripts to archive, compress and store data on tape and retrieve them easily. Image pre-processing steps were mostly performed using custom Fiji macros called from a master bash script to enable parallelization using GNU parallel (*Tange, 2011*). A low-pass filter was first applied to images to remove high frequency noise. For each z-stack, the signal of highest contrast was projected using a C++ algorithm as previously described (*Merkel et al., 2014*). Tiles were stitched

using Fiji (*Preibisch, 2009*; *Schindelin et al., 2012*). Stitched images were then loaded in Packing Analyzer v8.5 (PA8.5) for cell edge detection by using a seeded-watershed algorithm. To facilitate cell tracking, cross-correlation of subsequent images was performed in PA8.5 to calculate local tissue displacement beforehand. Resulting cell-tracking masks were parsed using a custom C++ parser that extracted cell shape properties, cell lineages and cell neighbor relationships for further storage in a SQLite relational DB. Topology was kept in the DB by ordering cell neighbors counter-clockwisely around each cell, namely by following a directed path of cell–cell junctions around each cell. One DB file was created per video. All DB queries were carried out using R (*R Development Core Team, 2014*) or Python (www.python.org; *Hunter, 2007*; *Pérez and Granger, 2007*; *van der Walt et al., 2011*).

We restricted our analyses to a subset of cells that were trackable throughout the entire course of the video, excluding cells that became visible only at later stages due to the apposition of the two cell layers. At 32 hAPF, most cells of the dorsal cell layer are visible. Therefore, we manually drew regions of interest (ROIs) on 32 hAPF wings using the veins and hinge–blade interface as landmarks, where the Ecad::GFP signal is more intense. These ROIs include the blade region delimited by the hinge–blade interface and the wing margin, but also the hinge–blade interface itself. In addition, we defined a triangular portion of the hinge delimited by the hinge–blade interface and the most anterior sensory organ located in the bulk of the hinge. In other hinge regions, cells were too small and elongated to be consistently tracked. Starting from these ROIs at 32 hAPF, we developed a backward-tracking algorithm that uses information about cell divisions and cell extrusions to reconstitute the corresponding group of cells at start of recording (usually ~16 hAPF), discarding margin cells that were not present through the course of the video. In the resulting set of cells, cells in contact with the wing margin at start of recording (about 16 hAPF) were not perfectly segmented and therefore further discarded. This gave rise to a group of cells that is fully consistent in time.

Since part of the hinge could not be analyzed at cellular resolution, we complemented our analysis of the small portion of the hinge by using particle image velocimetry (PIV) (*Adrian and Westerweel, 2011*) to extract information about the entire hinge deformation as described thereafter. PIV was implemented by Benoit Aigouy and described elsewhere (*Merkel et al., 2014*).

## Laser ablations

Laser ablations were performed using an ultraviolet laser microdissection apparatus as described elsewhere (*Grill et al., 2001*). For tissue severing, a C-apochromat-40×/1.2 water immersion lens was used to focus the beam along AP-oriented line segments to ablate both dorsal and ventral cell layers. Subsequent long-term imaging of the whole wing was then carried out on our dedicated Zeiss spinning disk. Extracellular-matrix ablations were done using a 63×/1.4 oil lens to focus the beam between the tissue and the cuticle and to cut over 10 μm in depth. Circular cuts were also conducted with the 63×/1.4 oil lens but over a depth of 5 μm. Each circular cut experiment was repeated on five distinct pupae.

## Analysis of circular laser ablations

Fly wings expressing Ecad::GFP were used in all circular cut experiments. Circular cuts were performed to disconnect a small subset of cells present inside the circle from the rest of the tissue. The interface of the tissue with the ablated circular region was always visible and well spatially defined after laser ablation. Therefore we quantified its displacement to calculate the initial velocity gradient that describes the immediate tissue deformation in response to the ablation. To do so, we first fitted an ellipse to this manually segmented interface, at 50 s after ablation. The major and minor axes of the ellipse define the orientation of two orthogonal kymographs, each intersecting the ellipse in two points corresponding to two opposite sides of the interface between the tissue and the ablated region. Each kymograph depicts the displacement as a function of time of the two opposite sides of this interface, which were manually segmented using sub-pixel resolution ROI. Thus, we quantified the displacement of four points corresponding to the intersections of the minor and major axes with the ellipse (see *Figure 2—figure supplement 3*). This procedure was semi-automated in a custom Fiji macro. The initial response of the tissue reflects the orientation and amplitude of tissue stresses. This can be captured by the initial velocity gradient within a few seconds after the 4 s ablation. Indeed, the initial displacement normal to the cut boundary was approximately linear as a function of time within the first 5 s after ablation. A linear fit to the data

provided the initial normal velocities $V_\parallel$ and $V_\perp$ along the major and minor axis of the ellipse, respectively. We then define the velocity gradient tensor in the coordinate system of the ellipse and rotate it into the image coordinate system:

$$v_{ij}^{cut} = \begin{pmatrix} \cos\theta & -\sin\theta \\ \sin\theta & \cos\theta \end{pmatrix} \begin{pmatrix} V_\parallel/r_\parallel & 0 \\ 0 & V_\perp/r_\perp \end{pmatrix} \begin{pmatrix} \cos\theta & \sin\theta \\ -\sin\theta & \cos\theta \end{pmatrix}, \tag{6}$$

where $\theta$ is the angle between the major axis of the ellipse and the PD axis (x axis). The radii $r_\parallel$ and $r_\perp$ are the half lengths of major and minor axes of the ellipsoidal shape at the time the velocities are determined. This velocity gradient can be decomposed into trace and a traceless part as:

$$v_{ij}^{cut} = \begin{pmatrix} C & 0 \\ 0 & C \end{pmatrix} + \begin{pmatrix} \tilde{v}_{xx}^{cut} & \tilde{v}_{xy}^{cut} \\ \tilde{v}_{xy}^{cut} & -\tilde{v}_{xx}^{cut} \end{pmatrix}, \tag{7}$$

where $v_{kk}^{cut} = 2C$ is the isotropic expansion rate, $\tilde{v}_{xx}^{cut}$ is the shear rate projected onto the PD-axis. While in the main text, tensors are denoted as bold face symbols, in the supplement we often use an explicit index or matrix notation for tensors. Latin indices $i$, $j$ denote the x and y coordinates of a cartesian coordinate system of the tissue.

Circular cuts were performed in nine different regions of the wing where cell elongation differed (*Figure 2I,I'*). Due to the fast imaging settings optimized to catch the initial velocity gradient, the image quality was not sufficient for cell segmentation. Therefore we used wings of different animals at the same stage (22.5 hAPF) to estimate cell elongation $Q_{xx}$ for each of the nine ablation regions that were easily located using morphological landmarks such sensory organs and veins (see *Figure 9A*). Average cell area $a_{cell}$ was estimated using a similar approach (see *Figure 9B*).

We find that the shear rate projected onto the PD axis $\tilde{v}_{xx}^{cut}$ increases with cell elongation. We find that data of $\tilde{v}_{xx}^{cut}$ as a function of $\tilde{v}_{xx}^{cut}$ can be accurately fitted by a linear function both in WT and *dumpy$^{ov1}$* wing. Fit parameters are given by

- WT: $\tilde{v}_{xx}[h^{-1}] = (0.018 \pm 0.008)Q_{xx} + (0.003 \pm 0.001)$
- Dp: $\tilde{v}_{xx}[h^{-1}] = (0.019 \pm 0.005)Q_{xx} + (0.003 \pm 0.002)$

The correlation between the isotropic expansion rate $v_{kk}^{cut}$ and the logarithm of cell area is less apparent. However, following *Equation 8*, we performed a linear fit to the data and found the following best fit parameters:

- WT: $\frac{1}{2}v_{kk}[h^{-1}] = (0.008 \pm 0.007)\ln(a_{cell}/a_{ref}) + (0.017 \pm 0.002)$
- Dp: $\frac{1}{2}v_{kk}[h^{-1}] = (0.005 \pm 0.008)\ln(a_{cell}/a_{ref}) + (0.015 \pm 0.002)$

Parameters obtained in isotropic expansion rate fits have high uncertainties and we do not use them in the rest of the paper.

## Measurements of wing dimensions

In order to compare theory to the experimental data we need to approximate the observed hinge and blade shapes by rectangles. We first show how a characteristic height and length can be associated to an arbitrary two-dimensional shape. We then specify how hinge and blade regions were selected to obtain the associated height and length. Note that for the purpose of estimating the wing dimensions, we do not used tracked regions of a subpart of the wing as in 'Data handling and image processing', but we use all available segmented data. Finally, we introduce corrections that account for changes of the visible part of the tissue in the field of view.

### Rectangle approximation to an arbitrary shape

To approximate an arbitrary tissue shape with a rectangle, we have to define a height $h$ and a length $L$ along the x direction that properly represent that region. To do this, we take into account the region area and elongation. First, we impose the condition that the rectangle representing the shape has the same area:

$$hL = A. \tag{8}$$

Additionally, we can use the elongation tensor $\mathbf{Q}^t$ defined in Appendix 1, 'Characterization of wing blade anisotropy' to obtain the rectangle aspect ratio. We then define the aspect ratio of the rectangle to be:

$$\frac{h}{L} = e^{2Q_{xx}^t}. \tag{9}$$

These two conditions finally give us:

$$L = \sqrt{A}e^{Q_{xx}^t}, \tag{10}$$

$$h = \sqrt{A}e^{-Q_{xx}^t}. \tag{11}$$

## Boundary between hinge and blade

To separate the wing into hinge and blade, we used a tracked region of cells on the boundary between the hinge and the blade (Data handling and image processing). At each time point we fit a fourth order polynomial through the coordinates of cells in this boundary region. This curve is used as a boundary separating hinge and the blade.

## Extrapolation of region dimensions

During the early time of recording, the tissue flows in and out of view. In order to correct for this, we performed the following measurements:

- We first obtain $h^S$ and $L^S$ from the shape of the segmented hinge and blade as described above.
- We then calculate the average shear in the hinge and blade from PIV measurements (see Data handling and image processing). From the average shear, an other estimate of the hinge and length can be obtained, up to unknown factors $\beta_L$ and $\beta_h$:

$$L^{\mathrm{PIV}}(t) = \beta_L \exp\left( \int_{t_0}^{t} v_{xx} \, dt \right), \tag{12}$$

$$h^{\mathrm{PIV}}(t) = \beta_h \exp\left( \int_{t_0}^{t} v_{yy} \, dt \right). \tag{13}$$

- After a time $t_f$ = 22.7 hAPF for WT and $t_f$ = 24.5 hAPF for *dumpy*$^{ov1}$, no flow is visible at the boundary of the tissue, and we therefore expect the two measurements of height and length to coincide. We therefore determine the coefficients $\beta_L$ and $\beta_H$ by minimizing the difference between $L^{\mathrm{PIV}}$ and $L^S$ and $h^{\mathrm{PIV}}$ and $h^S$ for $t > t_f$. A good agreement was indeed found for the two measurements for $t > t_f$.

Wings where the extracellular matrix was ablated distaly and anteriorly start much later than unperturbed WT and *dumpy*$^{ov1}$ and we do not observe significant inflow of the tissue. The same procedure was applied for these wings, with $t_f$ the first time where measurements were started.

## Acknowledgements

We are grateful to Caren Norden, Christian Dahmann, Jan Bruges and Stephan Grill for critical reading of the manuscript, the Light Microscopy Facility for providing support in elaborating the image acquisition pipeline, Scionics for providing optimal strategies for the storage of large data sets and excellent support on the cluster. We thank Stephan Grill for giving us access to his laser cutter device. We thank Tony Ashford, Kevin Brychcy and Clara Munz for their help in segmenting cells. RE acknowledges a Marie Curie fellowship from the EU 7th Framework Programme (FP7). This work was supported by the Max Planck Gesellschaft, and by the BMBF. SE acknowledges funding from the ERC.

# Additional information

## Competing interests

FJ: Reviewing editor, *eLife*. The other authors declare that no competing interests exist.

## Funding

| Funder | Author |
| --- | --- |
| Max-Planck-Gesellschaft | Marko Popović, Matthias Merkel, Amitabha Nandi, Guillaume Salbreux, Frank Jülicher |
| European Research Council (ERC) | Suzanne Eaton |
| Seventh Framework Programme | Raphaël Etournay |
| Bundesministerium für Bildung und Forschung | Corinna Blasse, Gene Myers, Frank Jülicher, Suzanne Eaton, Marko Popović, Matthias Merkel |

The funders had no role in study design, data collection and interpretation, or the decision to submit the work for publication.

## Author contributions

CB, BA, HB, GM, developed key image processing and image analysis methods; RE, MP, MM, AN, GS, FJ, SE, participated in regular group discussions to develop the ideas presented here, which are the result of multiple cycles of experimental and theoretical analysis

## Author ORCIDs

Frank Jülicher, http://orcid.org/0000-0003-4731-9185

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

## Appendix 1

# Shear decomposition using triangles.

Here we define the average pure shear rate of a tissue region and its decomposition into cellular contributions (**Merkel, 2014**). The cellular network of the tissue region is triangulated ('Triangulated cellular networks'). Averaging triangle deformations over all triangles within the region yields expressions for the large-scale shear rate and the shear rate contributions by different cellular processes ('Decomposition of the large-scale tissue shear rate'). The determination of spatial patterns of shear (**Figure 5**, **Figure 5—figure supplement 1**) and rotation (**Figure 6C,D,F,G**) are discussed in 'Spatially resolved shear patterns' and 'Power spectrum of local tissue rotation'.

## Triangulated cellular networks

Here, we first describe the tiling of the cellular network in detail before defining deformation and elongation of a single triangle.

### Triangulation procedure

To define a triangulation of a given cellular network, we consider all vertices of the network that are not located at the margin. Each of these inner vertices abuts at least three cells. If a given inner vertex $m$ abuts exactly three cells $\alpha$, $\beta$, and $\gamma$, we define a single triangle $m$ that has its corners at the centers $\boldsymbol{R}^{\alpha}$, $\boldsymbol{R}^{\beta}$, and $\boldsymbol{R}^{\gamma}$ of the abutting cells. Here, $\boldsymbol{R}^{\alpha}$ denotes the area center of cell $\alpha$: $\boldsymbol{R}^{\alpha} = (\int \boldsymbol{r} \mathrm{d}^2 r)/A^{\alpha}$, where the integral runs over the projected area $A^{\alpha}$ of cell $\alpha$.

For an inner vertex $m$ that abuts $N > 3$ cells $\alpha_1,...,\alpha_N$, we define $N$ triangles $m_1,...,m_N$. The corners of each of these triangles are defined as follows. One corner of each triangle is defined by the average position $\boldsymbol{C} = (\boldsymbol{R}^{\alpha_1} + ... + \boldsymbol{R}^{\alpha_N})/N$ of all $N$ cell centers. The other two corners of a given triangle $m_i$ with $i = 1,...,N$ are defined by the cell centers $\boldsymbol{R}^{\alpha_i}$ and $\boldsymbol{R}^{\alpha_{(i+1)}}$ ($\boldsymbol{R}^{\alpha_{(N+1)}}$ corresponds to $\boldsymbol{R}^{\alpha_1}$). Applying these two rules to all inner vertices, we obtain a tiling of the whole network without gaps or overlaps, leaving out only a small stripe at the margin of the network, which is ca. half a cell diameter wide.

### Triangle deformation

We characterize the deformation of a given triangle $m$ occurring during a time interval $\Delta t$ by a tensor $M_{ij}^m$. The Latin indices $i, j,...$ denote dimension indices of vectors, tensors, and matrices. For matrices, the first index is the row index and the second index is the column index. To define $M_{ij}^m$, we denote the initial and the final positions of the triangle corners by $\boldsymbol{R}^1$, $\boldsymbol{R}^2$, $\boldsymbol{R}^3$ and $\boldsymbol{R}'^1$, $\boldsymbol{R}'^2$, $\boldsymbol{R}'^3$, respectively. Then, the triangle side vectors before and after the deformation read $\Delta \boldsymbol{R}^{\beta\alpha} = \boldsymbol{R}^{\beta} - \boldsymbol{R}^{\alpha}$ and $\Delta \boldsymbol{R}'^{\beta\alpha} = \boldsymbol{R}'^{\beta} - \boldsymbol{R}'^{\alpha}$ for $\alpha, \beta \in \{1, 2, 3\}$. Now, there is a unique affine transformation defined by the tensor $M_{ij}^m$ that maps the initial side vectors to the final side vectors:

$$\Delta R_i'^{\beta\alpha} = M_{ij}^m \Delta R_j^{\beta\alpha}, \tag{14}$$

for any combination $\alpha, \beta \in \{1, 2, 3\}$. Equal indices on one side of the equation imply a summation over the two spatial dimensions. From standard linear algebra follows that the tensor $\boldsymbol{M}^m = (M_{ij}^m)$ has the following unique matrix representation:

$$\boldsymbol{M}^m = \begin{pmatrix} \Delta R_x'^{21} & \Delta R_x'^{31} \\ \Delta R_y'^{21} & \Delta R_y'^{31} \end{pmatrix} \cdot \begin{pmatrix} \Delta R_x^{21} & \Delta R_x^{31} \\ \Delta R_y^{21} & \Delta R_y^{31} \end{pmatrix}^{-1}. \tag{15}$$

In this equation the dot denotes the matrix product.

Based on the affine transformation tensor $M_{ij}^m$, we compute a discrete velocity gradient tensor $v_{ij}^m$ for the triangle $m$:

$$v_{ij}^m = \frac{1}{\Delta t}\left(M_{ji}^m - \delta_{ij}\right).\tag{16}$$

Here, the $\delta_{ij}$ symbol denotes the Kronecker symbol. Note that the tensor $M_{ij}^m$ enters transposed into this equation. This leads to a discrete velocity gradient $v_{ij}^m$ that approximates the local velocity gradient $v_{ij} = \partial_i v_j$ in the continuum limit, where $\mathbf{v}(\mathbf{r})$ denotes the velocity field.

Finally, we decompose the discrete triangle velocity gradient into trace $v_{kk}^m$, traceless, symmetric part $\tilde{v}_{ij}^m$, and antisymmetric part $\omega^m$:

$$v_{ij}^m = \frac{1}{2}v_{kk}^m\delta_{ij} + \tilde{v}_{ij}^m - \omega^m\epsilon_{ij},\tag{17}$$

with the generator of rotations defined by

$$\epsilon = \begin{pmatrix} 0 & -1 \\ 1 & 0 \end{pmatrix}.\tag{18}$$

The trace $v_{kk}^m$ represents the relative area expansion rate of the triangle, the traceless, symmetric part $\tilde{v}_{ij}^m$ represents a discrete triangle shear rate, and the antisymmetric part $\omega^m$ represents a triangle rotation rate.

## Triangle elongation

We characterize triangle shape by an affine transformation tensor $S_{ij}^m$ with respect to a universal reference triangle. This reference triangle is equilateral, and it has a fixed orientation and a fixed area $A_0$. The corners of the reference triangle are denoted by $\mathbf{C}^1$, $\mathbf{C}^2$, and $\mathbf{C}^3$. At some fixed time point, we denote the corners of a given triangle $m$ by $\mathbf{R}^1$, $\mathbf{R}^2$, and $\mathbf{R}^3$. Like above, we denote the respective triangle side vectors by $\Delta\mathbf{C}^{\beta\alpha} = \mathbf{C}^\beta - \mathbf{C}^\alpha$ and $\Delta\mathbf{R}^{\beta\alpha} = \mathbf{R}^\beta - \mathbf{R}^\alpha$ for $\alpha, \beta \in \{1, 2, 3\}$. Then, we define the tensor $\mathbf{S}^m = (S_{ij}^m)$ via the following matrix representation:

$$\mathbf{S}^m = \begin{pmatrix} \Delta R_x^{21} & \Delta R_x^{31} \\ \Delta R_y^{21} & \Delta R_y^{31} \end{pmatrix} \cdot \begin{pmatrix} \Delta C_x^{21} & \Delta C_x^{31} \\ \Delta C_y^{21} & \Delta C_y^{31} \end{pmatrix}^{-1}.\tag{19}$$

The so-defined tensor $S_{ij}^m$ maps the side vectors of the reference triangle to the side vectors of the triangle $m$, analogously to **Equation 14**.

To extract shape properties from the tensor $\mathbf{S}^m = (S_{ij}^m)$, we use the following polar decomposition (**Kaballo, 2013**):

$$\mathbf{S}^m = (A^m/A_0)^{1/2}\exp(\mathbf{Q}^m)\cdot\mathbf{Rot}(\Theta^m).\tag{20}$$

Here, $A^m$ denotes the area of triangle $m$. This equation uniquely defines the symmetric, traceless $2 \times 2$ tensor $\mathbf{Q}^m$, where the exponential of a tensor is defined by the Taylor series of the exponential function, and $\mathbf{Rot}(\Theta^m)$ denotes the matrix describing a counter-clockwise rotation by the angle $\Theta^m$. The symmetric, traceless tensor $\mathbf{Q}^m$ denotes the elongation of triangle $m$ and the angle $\Theta^m$ denotes a triangle orientation angle.

## Decomposition of the large-scale tissue shear rate

In this subsection, we consider the deformation of the cellular network during a finite time interval $\Delta t$ between two consecutive video frames. We denote the observed states of the cellular network corresponding to these two video frames by $O_n$ and $O_{n+1}$, where $n$ is an integer. We explain how we quantify the overall tissue shear rate between these two video frames and how we decompose it into cellular contributions. To separate the shear by topological changes from the shear by pure cellular deformation, we first introduce virtual intermediate network states between the two observed states $O_n$ and $O_{n+1}$.

## Intermediate network states

Intermediate network states are introduced for the following reasons. First, topological changes occur instantaneously, and as a consequence of this, no tissue shear can occur *during* a topological transition. Second, topological transitions involve the addition/removal of triangles to/from the network (**Figure 4L-O**). Third, we cannot determine the exact time at which a topological transition occurred during the interval $\Delta t$ between two observed network states $O_n$ and $O_{n+1}$.

To separate different types of changes between $O_n$ and $O_{n+1}$, we introduce three virtual intermediate network states denoted $I_1$, $I_2$, and $I_3$. This is illustrated by the following scheme:

$$O_n \xrightarrow{\text{T2 transitions}} I_1 \xrightarrow{\text{pure deformation}} I_2 \xrightarrow{\text{T1 transitions}} I_3 \xrightarrow{\text{cell divisions}} O_{n+1}. \tag{21}$$

We first define the intermediate state $I_3$ by starting from the observed state $O_{n+1}$. To this end, we fuse in $I_3$ all pairs of daughter cells in $O_{n+1}$. This corresponds to undoing the cell divisions occurring during the time interval $\Delta t$. The position of the mother cell center in $I_3$ is defined as the midpoint between the daughter cell centers in $O_{n+1}$. We then define the intermediate state $I_1$ from the state $O_n$ by removing all cells that will undergo a T2 transition during the time interval $\Delta t$. As a consequence, $I_1$ and $I_3$ contain the same set of cells. However, the topology and the cell center positions are different between these two states. Finally, we construct the state $I_2$ from the state $I_1$ by displacing each cell center to the position it attains in state $I_3$. Thus, the intermediate state $I_2$ has the same topology as the state $I_1$, but cell center positions correspond to those in $I_3$. As a consequence of all these definitions, all T2 transitions occur between states $O_n$ and $I_1$, any deformation (without topological changes) occurs between $I_1$ and $I_2$, all T1 transitions occur between $I_2$ and $I_3$, and all cell divisions occur between $I_3$ and $O_{n+1}$ (compare **scheme (21)**). Furthermore, since topological transitions occur instantaneously, the pure deformation between states $I_1$ and $I_2$ coorresponds to the time interval $\Delta t$.

The scheme in (**21**) is the basis for the computation of the shear rate and all cellular contributions to it. In the following, we will separately discuss shear in the absence of topological transitions (i.e., between the states $I_1$ and $I_2$) and the shear by topological transitions.

## Large-scale shear in the absence of topological transitions

The overall tissue shear rate $\tilde{v}_{ij}$ between the states $I_1$ and $I_2$ is computed as the average triangle shear rate:

$$\tilde{v}_{ij} = \langle \tilde{v}_{ij}^m \rangle_{I_1}. \tag{22}$$

In this equation, the average corresponds to an area-weighted average over all triangles, where the area weights are taken from state $I_1$. More generally, for any triangle-based quantity $q^m$, we define such an average by

$$\langle q^m \rangle_X = \frac{1}{A_X} \sum_m A_X^m q^m. \tag{23}$$

Here, the sum runs over all triangles $m$ of the network, $A_X^m$ denotes the area of triangle $m$ in state $X \in \{O_n, I_1, I_2, I_3, O_{n+1}\}$, and $A_X$ is the total area of the triangulation in state $X$: $A_X = \sum_m A_X^m$. The index $X$ is necessary, because the triangle areas generally change between two states. The average quantity $q^m$ may be evaluated for a network state $Y$ different from $X$ if both contain the same set of triangles. Interestingly, the large-scale shear rate $\tilde{v}_{ij}$ can equally be computed from margin displacements alone (**Merkel, 2014**; Merkel et al., in preparation). Furthermore, since there is no shear occurring during topological transitions, the average shear rate $\tilde{v}_{ij}$ between the intermediate states $I_1$ and $I_2$ also corresponds to the average shear rate between the observed states $O_n$ and $O_{n+1}$.

The overall tissue shear rate between the intermediate states $I_1$ and $I_2$ can be expressed in terms of the average triangle elongation change during that time:

$$\tilde{v}_{ij} = \frac{\langle Q_{ij}^m(I_2)\rangle_{I_2} - \langle Q_{ij}^m(I_1)\rangle_{I_1}}{\Delta t} + J_{ij} + D_{ij}. \tag{24}$$

Here,

$$J_{ij} = -\epsilon_{ik}\langle\omega^m\rangle_{I_1}\langle Q_{kj}(I_1) + Q_{kj}(I_2)\rangle_{I_1}, \tag{25}$$

is a corotational term and we have introduced the correlation contribution to shear $D_{ij}$, which can be approximated as

$$D_{ij} \approx \frac{\langle Q_{ij}^m(I_2)\rangle_{I_1} - \langle Q_{ij}^m(I_2)\rangle_{I_2}}{\Delta t}$$
$$- \epsilon_{ik}\left(\langle\omega^m[Q_{kj}(I_1) + Q_{kj}(I_2)]\rangle_{I_1} - \langle\omega^m\rangle_{I_1}\langle Q_{kj}(I_1) + Q_{kj}(I_2)\rangle_{I_1}\right). \tag{26}$$

In **Equation 26**, the first term captures effects of area expansion rate correlation with cell elongation, while the second term corresponds to correlations of rotation rate with cell elongation (**Figure 6E**). Here, we neglected higher order terms in the deformation and in the cell elongation. An exact decomposition of $\tilde{v}_{ij}$ in the limit of small time intervals $\Delta t$ can also be obtained (**Merkel, 2014**; Merkel et al., in preparation).

## Contributions to shear by topological changes

Between the two states $O_n$ and $O_{n+1}$, we define the contributions of topological changes to the shear rate as follows:

$$T_{ij} = \frac{1}{\Delta t}\left[\langle Q_{ij}(I_2)\rangle_{I_2} - \langle Q_{ij}(I_3)\rangle_{I_3}\right], \tag{27}$$

$$C_{ij} = \frac{1}{\Delta t}\left[\langle Q_{ij}(I_3)\rangle_{I_3} - \langle Q_{ij}(O_{n+1})\rangle_{O_{n+1}}\right], \tag{28}$$

$$E_{ij} = \frac{1}{\Delta t}\left[\langle Q_{ij}(O_n)\rangle_{O_n} - \langle Q_{ij}(I_1)\rangle_{I_1}\right]. \tag{29}$$

The total change of the average triangle elongation between the observed video states is $\Delta Q_{ij}(O_n, O_{n+1}) = \langle Q_{ij}(O_{n+1})\rangle_{O_{n+1}} - \langle Q_{ij}(O_n)\rangle_{O_n}$. Putting everything together, we obtain **Equation 2** from the main text:

$$\tilde{v}_{ij} = \frac{DQ_{ij}}{Dt} + R_{ij}, \tag{30}$$

with $R_{ij} = T_{ij} + C_{ij} + E_{ij} + D_{ij}$. The corotational derivative $DQ_{ij}/Dt$ is defined by

$$\frac{DQ_{ij}}{Dt} = \frac{\Delta Q_{ij}(O_n, O_{n+1})}{\Delta t} + J_{ij}, \tag{31}$$

and the quantities $\tilde{v}_{ij}$, $J_{ij}$, and $D_{ij}$ are defined in the previous subsection.

## Spatially resolved shear patterns

To visualize the spatial patterns of shear contributions, we divided the entire wing into a fixed grid of non-overlapping boxes, sized 26 μm × 26 μm. Then, each triangle is assigned to the box that contains the triangle center. This allows for the computation of average triangle quantities within each of these boxes. As in the previous section, for a given time interval $\Delta t$ between two consecutive video frames $O_n$ and $O_{n+1}$, we created intermediate states $I_1$, $I_2$, and $I_3$ (compare **scheme (21)**). Note that the set of triangles associated to a given box depends not only on the time interval $n$, but also on the state considered ($O_n$, $I_1$, $I_2$, $I_3$, or $O_{n+1}$).

The five shear patterns shown in **Figure 5** and **Figure 5—figure supplement 1** were computed as follows. The total shear rate $\tilde{v}_{ij}$, the corotational term $J_{ij}$, and the correlations $D_{ij}$ in a given

box were computed using **Equations 22, 25, 26**, respectively, where for each average appearing in these expressions, only those triangles associated to the given box in state $I_1$ were included. The remaining quantities were computed as follows:

$$\left[\frac{DQ_{ij}}{Dt}\right]^b = \frac{1}{\Delta t}\left[\langle Q_{ij}^m(O_{n+1})\rangle_b - \langle Q_{ij}^m(O_n)\rangle_b\right] + J_{ij}^b \text{ (cell elong. shear),} \tag{32}$$

$$T_{ij}^b = \frac{1}{\Delta t}\left[\langle Q_{ij}^m(I_2)\rangle_b - \langle Q_{ij}^m(I_3)\rangle_b\right] \text{ (T1 shear),} \tag{33}$$

$$C_{ij}^b = \frac{1}{\Delta t}\left[\langle Q_{ij}^m(I_3)\rangle_b - \langle Q_{ij}^m(O_{n+1})\rangle_b\right] \text{ (CD shear).} \tag{34}$$

Here, $b$ denotes the box index. The averaging bracket $\langle Q_{ij}^m(X)\rangle_b$ denotes an average of the triangle elongation in state $X$ over all triangles $m$ that are assigned to box $b$ in state $X$ with area weights from state $X$, where $X \in \{O_n, I_1, I_2, I_3, O_{n+1}\}$:

$$\langle Q_{ij}^m(X)\rangle_b = \frac{1}{A_X}\sum_m A_X^m Q_{ij}^m(X). \tag{35}$$

Here, the sums runs over all triangles $m$ that are associated to box $b$ in state $X$. The contributions by T2 transitions were negligible.

For the pattern of corotational cell elongation change, we neglected convective contributions. To evaluate, how much this affected the shown patterns, we compared our results to an alternative implementation that tracks patches of tissue over time. Here, the fixed grid was only used for the initialization of the patches at the beginning of the video. For this alternative implementation, including a convective contribution is not necessary since each patch of tissue is followed over time. However after all, visual comparison of both implementations yielded a good agreement (not shown). Thus, the patterns in **Figure 5** and **Figure 5—figure supplement 1** were not essentially affected by neglecting the convective term.

## Power spectrum of local tissue rotation

Here, we explain how we computed the spatial power spectrum of the local tissue rotation rate shown in **Figure 6D**. For a given time interval between two consecutive video frames, the triangle rotation rates $\omega^m$ are computed as described by **Equation 17** and 'Decomposition of the large-scale tissue shear rate'. This yields a piecewisely constant field of local rotation rates $\omega(\mathbf{r})$: at a given position $\mathbf{r}$ that lies within a triangle $m$, the value of the field is defined by the triangle rotation rate $\omega(\mathbf{r}) = \omega^m$.

The power spectrum $|\mathcal{F}[\omega]|^2(\mathbf{q})$ of this field is then computed as the squared norm of following Fourier transform:

$$\mathcal{F}[\omega](\mathbf{q}) = \frac{1}{2\pi(L_xL_yW)^{1/2}}\int w_{2D}(\mathbf{r})\omega(\mathbf{r})e^{i\mathbf{q}\cdot\mathbf{r}}d^2r. \tag{36}$$

The integration in this expression was carried out over a squared region with a side length of $L_x = L_y = 800$ pixels (164 μm) that is located in the center of the wing, covering parts of the longitudinal veins 2 to 4. The product in the exponential is a dot product. The components of the wave vector $\mathbf{q} = (q_x, q_y)$ could take values $q_x = 2\pi n_x/L_x$ and $q_y = 2\pi n_y/L_y$, where $n_x$ and $n_y$ are integers with $-N/2 \leq n_x, n_y < N/2$. We set $N = 100$. The function $w_{2D}(\mathbf{r})$ is a window function defined on the squared region. With the position $\mathbf{r} = (r_x, r_y)$ measured relative to the bottom-left corner of the squared region, it reads

$$w_{2D}(\mathbf{r}) = w_{BH}\left(\frac{r_x}{L_x}\right)w_{BH}\left(\frac{r_y}{L_y}\right). \tag{37}$$

Here, $w_{BH}(z)$ denotes the one-dimensional Blackman Harris Window, which is defined by

$$w_{BH}(z) = a_0 - a_1 \cos(2\pi z) + a_2 \cos(4\pi z) - a_3 \cos(6\pi z), \tag{38}$$

with $a_0 = 0.35875$, $a_1 = 0.48829$, $a_2 = 0.14128$, and $a_3 = 0.01168$. The scalar $W$ is defined by

$$W = \left( \int_0^1 w_{BH}^2(z) dz \right)^2 \approx 0.067. \tag{39}$$

It corresponds to the average value of the squared window function $W = \langle w_{2D}^2 \rangle$. $W$ is used in the normalization factor in **Equation 36** in order to ensure that the second moment of $\omega$ corresponds to the integrated power spectrum

$$\langle \omega^2 \rangle = \frac{2\pi}{L_x} \frac{2\pi}{L_y} \sum_{\mathbf{q}} |\mathcal{F}[\omega]|^2 (\mathbf{q}), \tag{40}$$

in the limit $N \to \infty$ and in the absence of any correlation between the averaging window $w_{2D}(\mathbf{r})$ and $\omega(\mathbf{r})$ such that $\langle w_{2D}^2 \omega^2 \rangle = \langle w_{2D}^2 \rangle \langle \omega^2 \rangle$. Here, the average of any spatially varying function $q(\mathbf{r})$ is defined by the area average over the squared region: $\langle q \rangle = (\int q(\mathbf{r}) d^2 r)/L_x L_y$.

We numerically evaluated the Fourier transform using the following expression:

$$\mathcal{F}[\omega](\mathbf{q}) \approx \frac{1}{2\pi (L_x L_y W)^{1/2}} \sum_m w_{2D}(\mathbf{r}_c^m) \omega^m \int_{A^m} e^{i\mathbf{q} \cdot \mathbf{r}} d^2 r, \tag{41}$$

where we used the fact that the window function $w_{2D}(\mathbf{r})$ varied only little over the size of a single triangle. Here, the vector $\mathbf{r}_c^m$ denotes the center position of triangle $m$ and the integral is carried out over the area of triangle $m$. For a triangle $m$ with corners $\mathbf{r}_0 = (x_0, y_0)$, $\mathbf{r}_1 = (x_1, y_1)$, and $\mathbf{r}_2 = (x_2, y_2)$ in counter-clockwise order, the integral over triangle area $A^m$ can be expressed as:

$$\int_{A^m} e^{i\mathbf{q} \cdot \mathbf{r}} d^2 r = \sum_{i=0}^2 e^{i\mathbf{q} \cdot \mathbf{r}_i} \frac{(x_i - x_{i-1})(y_{i+1} - y_i) - (x_{i+1} - x_i)(y_i - y_{i-1})}{(\mathbf{q} \cdot [\mathbf{r}_i - \mathbf{r}_{i-1}])(\mathbf{q} \cdot [\mathbf{r}_{i+1} - \mathbf{r}_i])}. \tag{42}$$

In this expression, indices are taken modulo three such that $i = 3$ corresponds to $i = 0$ and $i = -1$ corresponds to $i = 2$.

**Figure 6D** results from averaging the power spectra $|\mathcal{F}[\omega]|^2$ computed from the 11 time intervals in between 12 consecutive video frames. From **Figure 6D**, we can extract the width of the observed rotation bands. This width corresponds to the position of the dominant mode in the power spectrum, which we observe at $\mathbf{q}_{peak} \approx (0, \pm 0.3 d_0/2\pi)$, where $d_0 = 4$ μm denotes the typical cell diameter at 21 hAPF. This mode corresponds to a wave length of ca. $3d_0$, and thus to a width of the bands of ca. $1.5d_0$.

## Role of T1 transitions in the correlation-induced shear

We have divided all triangles in the blade region in a WT wing into two groups. The first group contains all triangles that will disappear due to a T1 transition during the next 9 frames of the video, corresponding to about 45 min. The second group contains all other triangles. The correlation contribution to the total tissue shear can be approximately written as

$$D_{ij} \approx \frac{A_{T1}}{A} D_{ij}^{T1} + \frac{A_{\bar{T}1}}{A} D_{ij}^{\bar{T}1}. \tag{43}$$

Here, $A_{T1}$ and $A_{\bar{T}1}$ are the areas and $D_{ij}^{T1}$ and $D_{ij}^{\bar{T}1}$ are the correlation contributions to shear of the first and the second group of triangles, respectively. The area of the blade region is denoted by $A$.

In the inset of **Figure 6I** we show that the area of the first group of cells $A_{T1}$ (green line) is only a small fraction of the total blade area $A$ (blue line). On average, the ratio is $A_{T1}/A = 15.3\%$. Despite this, we show in **Figure 6I** that the measured contribution of the first group projected

onto the PD axis $A_{T1}/AD_{xx}^{T1}$ (blue line) accounts for a significant amount of shear due to correlation effects $D_{xx}$ (magenta line).

## Characterization of wing blade anisotropy

Here, we define the wing blade anisotropy $Q^t$ plotted in *Figure 6—figure supplement 1B*. To this end, we first define a matrix

$$M = \begin{pmatrix} M_{xx} & M_{xy} \\ M_{yx} & M_{yy} \end{pmatrix}, \tag{44}$$

where

$$M_{ij} = \frac{1}{A} \int_A \left( r_i - r_i^{CM} \right) \left( r_j - r_j^{CM} \right) \mathrm{d}x\mathrm{d}y. \tag{45}$$

Here, $A$ denotes the total area of the wing blade and $r_i^{CM}$ denotes its area center:

$$r_i^{CM} = \frac{1}{A} \int_A r_i \mathrm{d}x\mathrm{d}y. \tag{46}$$

Note that the matrix $M$ is symmetric. Now, the wing blade anisotropy $Q^t$ is uniquely defined by the following relation:

$$e^{2Q^t} = (\det M)^{-1/2} M, \tag{47}$$

Here, $\det M$ denotes the determinant of $M$ and the exponential of a matrix is defined by the Taylor series of the exponential function. The unique tensor $Q^t$ that fulfills *Equation 47* has to be symmetric and traceless (compare *Equation 20* and [*Merkel, 2014*; Merkel et al., in preparation]). Put differently, it is a nematic tensor like the total shear rate $\tilde{v}$ or the average cell elongation $Q$.

Note that in general, the change of wing blade anisotropy $Q^t$ can not directly be computed from the total tissue shear rate $\tilde{v}$. However, for the special case where shear occurs homogeneously over the entire wing area and where the axes of the shear rate tensor $\tilde{v}$ and of wing blade anisotropy $Q^t$ are either parallel or perpendicular to each other, one obtains for the tensor component along the x axis:

$$Q_{xx}^t(t) = Q_{xx}^t(0) + \int_0^t \tilde{v}_{xx} \mathrm{d}t'. \tag{48}$$

**Appendix 2**

## Hydrodynamic theory of epithelia.

In this section, we discuss a coarse-grained hydrodynamic theory that describes the mechanics of a two dimensional epithelium, the cells of which can rearrange through topological changes. The hydrodynamic theory describes the evolution of two main physical quantities: the velocity field $v_i$ within the tissue, and the average cell elongation $Q_{ij}$, treated as an internal variable of the tissue. In addition, we take into account a polarity field $q_{ij}$, which gives a preferred axis in the tissue and allows for internal anisotropic processes to occur.

To obtain a full description of the epithelium mechanics, it is necessary to specify tissue processes that drive the shear created by topological changes. The shear rate $R_{ij}$ created by topological changes is a tensorial quantity. In the spirit of hydrodynamic theories, we propose a constitutive relation for this tensor by writing all possible terms compatible with its symmetry, at lowest order in the driving fields. In the following we describe the full hydrodynamic theory.

### Tissue strain rate decomposition

The velocity of the tissue is denoted by the vector $v_i$. The tissue strain rate $v_{ij} = \frac{1}{2}(\partial_i v_j + \partial_j v_i)$ is decomposed in an anisotropic traceless part, $\tilde{v}_{ij}$, and an isotropic part corresponding to the trace of the tissue strain rate:

$$v_{ij} = \tilde{v}_{ij} + \frac{1}{2} v_{kk} \delta_{ij}. \tag{49}$$

Here and in the following summation over identical indices is implied, $v_{kk} = \sum_k v_{kk}$. The trace of the strain rate $v = v^{kk}$ describes changes in the area of the tissue. The anisotropic traceless part $\tilde{v}_{ij}$, called shear rate, describes tissue shape changes which conserve the area.

Now we discuss the decomposition of isotropic and anisotropic components of strain rate in distinct cellular processes.

### Isotropic part

We denote the average cell area $a$. The isotropic part of the velocity gradient and the change of cell area are related by the equation

$$v = \frac{1}{a}\frac{da}{dt} + k_d - k_e, \tag{50}$$

with $k_d$ the rate of cell division and $k_e$ the rate of cell extrusion. The derivative $d/dt$ denotes a convected derivative, $df/dt = \partial_t f + v_i \partial_i f$. **Equation 79** is equivalent to the continuity equation for the cell density $\rho = 1/a$:

$$\partial_t \rho + \partial_i(\rho v_i) = \rho(k_d - k_e). \tag{51}$$

### Traceless anisotropic part

The difference between the traceless velocity gradient tensor $\tilde{v}_{ij}$ and the rate of cell deformation tensor $DQ_{ij}/Dt$ is accounted for by a tensor of shear due to topological changes $R_{ij}$:

$$\tilde{v}_{ij} = \frac{DQ_{ij}}{Dt} + R_{ij}, \tag{52}$$

where $D/Dt$ denotes a corotational convected derivative, defined for a tensor $A_{ij}$ as

$$\frac{DA_{ij}}{Dt} = \partial_t A_{ij} + v_k \partial_k A_{ij} + \omega_{ik} A_{kj} + \omega_{jk} A_{ik}, \tag{53}$$

with $\omega_{ij} = (\partial_i v_j - \partial_j v_i)/2$ the antisymmetric part of the velocity gradient. Following the triangle method described in 'Triangulated cellular networks', the tensor $R_{ij}$ can be decomposed into several contributions:

$$R_{ij} = T_{ij} + C_{ij} + E_{ij} + D_{ij}, \tag{54}$$

where $T_{ij}$ is the shear rate tensor created by $T_1$ transitions, $C_{ij}$ is the shear rate created by cell division, $E_{ij}$ is the shear rate created by cell extrusion, and $D_{ij}$ denotes a tensor accounting for correlation effects.

## Constitutive equation for the tissue stress

The tissue stress can be decomposed into an anisotropic traceless part, $\tilde{\sigma}_{ij}$, and an isotropic part corresponding to a two dimensional tissue pressure $P$:

$$\sigma_{ij} = \tilde{\sigma}_{ij} - P\delta_{ij}. \tag{55}$$

In an elastic tissue, the anisotropic part of the stress and the pressure depends on cell deformation. We therefore write the following constitutive equations for the stress:

$$\tilde{\sigma}_{ij} = 2KQ_{ij} + \zeta_{ij}, \tag{56}$$

$$P = -\bar{K}\ln\frac{a}{a_0} - \bar{\eta}\frac{a_0}{a}\frac{d}{dt}\left(\frac{a}{a_0}\right). \tag{57}$$

Here, $K$ and $\bar{K}$ are a shear and an area elastic modulus. The traceless symmetric tensor $\zeta_{ij} \sim q_{ij}$ describes a cell autonomous active stress which can also be interpreted as a spontaneous cell elongation. Here, $\bar{\eta}$ is a cell area viscosity, related to the dissipation associated with changes of cell area. Note that this viscosity was not included in **Equation 3** in the main text, but is used in the fitting procedure described in 'Fitting of the rectangle model to cell and tissue shape in the hinge and blade'.

Cell division is associated with a corresponding rapid reduction of cell area (**Figure 3C,D**) suggesting that the preferred cell area $a_0$ changes during cell division. We thus express the preferred cell area $a_0$ as:

$$a_0 = \bar{a}_0\exp\left(-\frac{\bar{\zeta}}{\bar{K}}\right), \tag{58}$$

where $\bar{a}_0$ is a cell intrinsic preferred cell area that changes rapidly due to cell division and $\bar{\zeta}$ corresponds to an isotropic contractile tension. The contribution $\bar{a}_0$ follows the equation

$$\frac{1}{\bar{a}_0}\frac{d\bar{a}_0}{dt} = -k_d, \tag{59}$$

which corresponds to halving of $\bar{a}_0$ for each round of division. In addition to **Equation 59** we can choose the value of $\bar{a}_0$ at an arbitrary time point. For simplicity we choose $\bar{a}_0(t_0) = a(t_0)$.

With the decomposition of the preferred cell area **Equation 58** and using the isotropic strain rate decomposition **Equation 50**, the isotropic pressure can be expressed as

$$P = -\bar{K}\left(\ln\frac{A}{A_0} + \int_{t_0}^{t} k_e dt\right) - \bar{\zeta} - \bar{\eta}(v + k_e), \tag{60}$$

where $A = A_0\exp(\int_{t_0}^{t} v dt)$ is the change of tissue area between $t_0$ and $t$, $A_0$ is the tissue area. **Equation 60** clarifies that the definition of the reference area **Equation 58** corresponds to a decomposition of the tissue pressure into an elastic part (first term in the right-hand side of **Equation 60**), a part corresponding to the stress generated by loss of cells by extrusion, isotropic contractility $\bar{\zeta}$ driving tissue contraction or expansion, and a viscous part associated with the rate of cell area change.

## Constitutive equations for the shear created by topological changes

The shear created by topological changes, $R_{ij}$, is a traceless symmetric tensor, whose value must depend on other traceless symmetric tensors in the tissue. Physically, topological changes occur preferentially along a direction set by an external cue. In the hydrodynamic description we propose, such a cue can arise from cell elongation $Q_{ij}$ or cell polarity $q_{ij}$. In the spirit of linear response theory, we write that $R_{ij}$ depends linearly on the cell deformation tensor

$$\left(1 + \tau_d \frac{D}{Dt}\right) R_{ij} = \frac{1}{\tau_r} Q_{ij} + \lambda_{ij}. \tag{61}$$

Here, D/Dt is the corotational convected derivative (see **Equation 53**) and $\tau_r$ is a timescale, characterising the rate of shear corresponding to topological changes induced by cell deformation. Note that for simplicty, we neglect nonlinearities arising from the corotational derivative in **Equation 61** in the main text and in these supplements. The traceless symmetric tensor $\lambda_{ij} \sim q_{ij}$ describes the effect of polarity-induced topological changes. Tissue stability requires $\tau_r > 0$. The time $\tau_d$ characterizes the delay in the response of topological changes to cell elongation and cell polarity.

## Effect of delay in topological changes

We discuss here the delay term for topological changes introduced in **Equation 4** of the main text. Neglecting non-linearities introduced by convection and rotations, the shear de-composition **Equation 52** and constitutive **Equation 61** can be written:

$$\partial_t Q_{ij} = -R_{ij} + \tilde{v}_{ij}, \tag{62}$$

$$\tau_d \partial_t R_{ij} = \frac{Q_{ij}}{\tau_r} - R_{ij} + \lambda_{ij}, \tag{63}$$

which appear as a dynamical system for the evolution of $Q_{ij}$ and $R_{ij}$. This dynamical system evolves towards the fixed point $R_{ij}^* = \tilde{v}_{ij}$, $Q_{ij}^* = -\tau_r[\lambda_{ij} - \tilde{v}_{ij}]$. Assuming a constant shear rate $\tilde{v}_{ij}$, deviations around the fixed point follow the equation

$$\partial_t \begin{pmatrix} Q_{ij} - Q_{ij}^* \\ R_{ij} - R_{ij}^* \end{pmatrix} = \begin{pmatrix} 0 & -1 \\ \dfrac{1}{\tau_r \tau_d} & -\dfrac{1}{\tau_d} \end{pmatrix} \begin{pmatrix} Q_{ij} - Q_{ij}^* \\ R_{ij} - R_{ij}^* \end{pmatrix}. \tag{64}$$

The dynamics of the system around the fixed point depends on the timescales $\tau_r$ and $\tau_d$. For a short enough delay $\tau_d < \tau_r/4$, the eigenvalues of the matrix in **Equation 64** are real negative, and the system relaxes exponentially to the stable fixed point. For a long enough delay $\tau_d > \tau_r/4$, the matrix has complex eigenvalues $-1/2\tau_d \pm i\sqrt{4\tau_d\tau_r - \tau_r^2}/2\tau_r\tau_d$. In that case, the system behaves as a damped oscillator and relaxes to the fixed point following a spiral in the phase space $R_{ij}$, $Q_{ij}$. The system relaxes towards the fixed point on a timescale $2\tau_d$, with a period of oscillations

$$T = 2\pi \frac{2\tau_r \tau_d}{\sqrt{4\tau_d\tau_r - \tau_r^2}}. \tag{65}$$

A relaxation resembling a damped oscillator in the phase space $R_{xx}$, $Q_{xx}$ is indeed observed experimentally (**Figure 9D** and **Figure 9—figure supplement 2**). Fitting the relaxation in the phase space $R_{xx}$, $Q_{xx}$ to **Equations 62, 63** yields a delay $\tau_d$ of the order of a few hours, thus playing a significant role during pupal wing development.

## Appendix 3

# Rectangle model for pupal wing morphogenesis.

In this section, we use the theoretical framework described in the previous section to obtain a simplified description of the morphogenesis of the pupal wing. We represent the hinge and blade by rectangles, connected by elastic springs to an external fixed frame, corresponding to the cuticle (**Figure 10A** and **Figure 10—figure supplement 1**).

## Geometry and boundary conditions
### Geometry
For simplicity, we choose to represent the hinge and the blade by two rectangles. The $x$ axis labels the proximal-distal direction, while the $y$ axis labels the anterior-posterior direction. The two rectangles only deform uniformly, and are therefore subjected to a homogeneous velocity gradient with components $v_{xx}$ and $v_{yy}$. The position of the interface between the hinge and the blade is denoted $x_{HB}$, the heights of the hinge and of the blade $h^H$ and $h$. In the WT situation, we consider the total length of the wing (including the hinge and the blade) to be constant and equal to $L$.

### Velocity
Because the proximal distal velocity field $v_x$ has to be continuous at the contact point $x_{HB}$ (**Figure 10—figure supplement 1B**), and cancels at the proximal and distal ends, the gradient of flow in the hinge and in the blade can be directly obtained as a function of the velocity of the interface $v_{HB} = dx_{HB}/dt$:

$$v_{xx}^H = \frac{v_{HB}}{x_{HB}},\tag{66}$$

$$v_{xx}^B = -\frac{v_{HB}}{L - x_{HB}}.\tag{67}$$

Similarly, the gradient of flow in the $y$ direction can be related to the change of heights of the hinge and of the blade:

$$v_{yy}^H = \frac{1}{h^H}\frac{dh^H}{dt},\tag{68}$$

$$v_{yy} = \frac{1}{h}\frac{dh}{dt}.\tag{69}$$

## Boundary stresses
### WT wing
From force balance, the stresses acting in the hinge $\sigma_{xx}^H$ and in the blade $\sigma_{xx}$, integrated along the height, have to be balanced at the interface:

$$\gamma\frac{dx_{HB}}{dt} = -h^H\sigma_{xx}^H + h\sigma_{xx} - k_{PD}^H(x_{HB} - x_0) + k_{PD}((L - x_{HB}) - (L_0 - x_0)).\tag{70}$$

Here, $\gamma$ is an effective friction coefficient, reflecting dissipative processes external to the tissue and limiting its velocity. The two elastic moduli constraining the length of the hinge and blade are denoted $k_{PD}^H$ and $k_{PD}$. They reflect external attachments of the tissue to the cuticle (see **Figure 10A** and **Figure 10—figure supplement 1A**). The initial PD lengths of the hinge and the entire wing are denoted $x_0$ and $L_0$, respectively.

In addition, we consider the effects of the elastic connections at the anterior and posterior boundaries. These connections resist changes of heights of the two rectangles, and can have

a different elastic modulus along the hinge and the blade, denoted $k^H$ and $k$, respectively. Thus, force balance at the anterior and posterior boundaries in the hinge and in the blade can be written:

$$\sigma_{yy}^H = -k^H \left( h^H - h_0^H \right),\qquad(71)$$

$$\sigma_{yy} = -k(h - h_0).\qquad(72)$$

Here, $h_0$ and $h_0^H$ are the heights where the springs are relaxed in the blade and in the hinge. We take these reference heights to be equal to the initial blade and hinge heights.

## Extracellular-matrix ablation at the anterior margin

Laser cut experiments severing the blade from the cuticle at the anterior margin modify the elastic modulus resisting the deformation of the blade along the $y$ direction (**Figure 8—figure supplement 3A** and **Figure 10—figure supplement 1A**). To reflect this change, we introduce a modified boundary condition for the stress along the anterior posterior direction, replacing **Equation 72**:

$$\sigma_{yy} = 0.\qquad(73)$$

Other boundary conditions are not modified and are taken as in the WT case.

## Extracellular-matrix ablation at the distal margin

The distally severed blade retracts distally away from the cuticle (**Figure 8—figure supplement 3A** and **Figure 10—figure supplement 1A**). As a result, the total length of hinge and blade $L$ can change. The dynamics for the evolution of the length of the blade is given by the dynamic equation:

$$\gamma \frac{dL}{dt} = -h\sigma_{xx} - k_{PD}(L - x_{HB}),\qquad(74)$$

where $\gamma$ is an effective friction coefficient characterising dissipative processes external to the tissue. We use the same coefficient $\gamma$ in **Equations 72, 74**, because we expect that both frictions arise from the same frictional forces acting between the wing and the surrounding cuticle.

## *dumpy*$^{ov1}$ mutants

In *dumpy*$^{ov1}$ mutants the distal margin of the wing detaches from the cuticle similar to laser ablation of ECM on the distal margin (**Figure 8A,B** and **Figure 10—figure supplement 1A**). We therefore use the boundary condition **Equation 74** for *dumpy*$^{ov1}$ mutants.

## Appendix 4

### Fitting procedure.

In this section, we describe the fitting procedure to experimental measurements. The fitting procedure was performed in two stages:

- First, the constitutive equation for topological changes, *Equation 5* in the main text, was fitted to experimental data.
- Using the parameters obtained for the fit to *Equation 18*, the dynamics of cell shape and tissue shape predicted by the rectangle model of section Appendix 3 was then fitted to experimental data by adjusting the remaining parameters.

We analysed six wings, for which the full blade and a fraction of the hinge were segmented. In this section, tissue flows were measured by PIV analysis rather than by cell tracking (see 'Materials and methods', Data handling and image processing). The six analysed wings were 3 WTs, one *dumpy*[ov1] mutant (Dp), one wing where the extracellular matrix was distally ablated (ECMDistCut) and one wing where the extracellular matrix was anteriorly ablated (ECMAntCut).

### Fitting of the constitutive equation for topological changes

We discuss here the fitting procedure to the constitutive *Equation 5* in the main text, for the shear rate created by topological changes along the AP axis, for the blade and for the hinge:

$$\left(1 + \tau_d \frac{d}{dt}\right) R_{xx} = \frac{1}{\tau_r} Q_{xx} + \lambda_{xx}. \tag{75}$$

From a given measured time evolution of $Q_{xx}$, *Equation 75* can be solved to yield the tensor of shear created by topological changes:

$$R_{xx}^{th}(t, Q_{xx}) = e^{-(t-t_0)/\tau_d} \left[ R_{xx}(t_0) + \frac{1}{\tau_d} \int_{t_0}^{t} dt' e^{(t'-t_0)/\tau_d} \left( \frac{1}{\tau_r} Q_{xx}(t') + \lambda_{xx} \right) \right]. \tag{76}$$

We then define the following objective functions:

$$S_B\left(\tau_r, \tau_d, \lambda_{xx}^k\right) = \sum_{k \in \text{wings}} \sum_n \left[ R_{xx}^{\exp,k}(t_n) - R_{xx}^{th,k}\left(t_n, Q_{xx}^{\exp,k}\right) \right]^2, \tag{77}$$

$$S_H\left(\tau_r^H, \tau_d^H, \lambda_{xx}^{H,k}\right) = \sum_{k \in \text{wings}} \sum_n \left[ R_{xx}^{H,\exp,k}(t_n) - R_{xx}^{H,th,k}\left(t_n, Q_{xx}^{H,\exp,k}\right) \right]^2, \tag{78}$$

where $k$ labels the six wings which were analysed for the fitting procedure and $t_n$ corresponds to time points of experimental measurements. Components of tensors $R_{ij}^{H,\exp,k}$, $R_{ij}^{\exp,k}$ of shear created by topological changes were quantified by subtracting the time derivative of cell elongation $dQ_{ij}/dt$ to the gradient of flow velocity $v_{ij}$, rather than by direct evaluation as described in 'Contributions to shear by topological changes'. $R_{xx}^{th,k}$ is obtained by using *Equation 76*, with $t_0$ taken as the first time point of the measurements. Different wings are fitted with a single set of values for the timescales $\tau_r$ and $\tau_d$ and different values of $\lambda_{xx}^k$. Initial values $R_{xx}(t_0)$ were treated as fit parameters.

To minimize the objective function, we use the Levenberg–Marquardt implementation (*Jones et al., 2001*). The resulting optimal fit parameters are reported in *Table 1*. A bootstrapping algorithm was used to estimate parameters uncertainties.

### Fitting of the rectangle model to cell and tissue shape in the hinge and blade

We now describe the fitting procedure of the rectangle model described in section Appendix 3 to experimental measurements. Specifically, we fit the hinge and blade shapes predicted by the rectangle model to measurements of the average strain rate in the hinge and blade. In the WT

situation, the dynamics of the rectangle model takes the form of a dynamical system for the cell deformation in the hinge $Q_{xx}^H$ and blade $Q_{xx}$, the topological changes tensors in the hinge $R_{xx}^H$ and blade $R_{xx}$, the blade height $h$ and the position of the hinge–blade interface $x_{HB}$. This dynamical system can be written as:

$$\frac{dQ_{xx}}{dt} = \frac{1}{2}\left(v_{xx} - v_{yy}\right) - R_{xx},$$

(79)

$$\frac{dQ_{xx}^H}{dt} = \frac{1}{2}\left(v_{xx}^H - v_{yy}^H\right) - R_{xx}^H,$$

(80)

$$\frac{d}{dt}R_{xx} = -\frac{1}{\tau_d}R_{xx} + \frac{1}{\tau_r\tau_d}Q_{xx} + \frac{1}{\tau_d}\lambda_{xx},$$

(81)

$$\frac{d}{dt}R_{xx}^H = -\frac{1}{\tau_d}R_{xx}^H + \frac{1}{\tau_r\tau_d}Q_{xx}^H + \frac{1}{\tau_d}\lambda_{xx}^H,$$

(82)

$$\gamma\frac{dx_{HB}}{dt} = -h^H\sigma_{xx}^H + h\sigma_{xx} - k_{PD}^H(x_{HB} - x_0),$$

$$+ k_{PD}((L - x_{HB}) - (L_0 - x_0)),$$

(83)

$$\frac{dh}{dt} = hv_{yy}.$$

(84)

**Equations 79, 80** correspond to the strain rate decomposition **Equation 52** in the hinge and in the blade, **Equations 81, 82** to the constitutive **Equation 61** for the tensor of topological changes in the hinge and in the blade, **Equation 83** to the force balance **Equation 70** at the hinge–blade interface, **Equation 84** to the height kinematic **Equation 69**.

Solving the dynamical system above requires to compute some components of the strain rate tensor $v_{ij}$ and of the stress tensor $\sigma_{ij}$. These components are obtained by solving the following system of equation for the components of the strain rate $v_{xx}$, $v_{xx}^H$, $v_{yy}$, $v_{yy}^H$, of the stress tensor $\sigma_{xx}$, $\sigma_{xx}^H$, $\sigma_{yy}$, $\sigma_{yy}^H$, and of the hinge pressure $P_H$:

$$v_{xx}^H = \frac{1}{x_{HB}}\frac{dx_{HB}}{dt},$$

(85)

$$v_{xx} = -\frac{1}{L - x_{HB}}\frac{dx_{HB}}{dt},$$

(86)

$$v_{xx}^H + v_{yy}^H = \frac{1}{A_H(t)}\frac{dA_H}{dt},$$

(87)

$$\sigma_{xx}^H = 2K^HQ_{xx}^H + \zeta_{xx}^H - P^H,$$

(88)

$$\sigma_{xx} = 2KQ_{xx} + \zeta_{xx} + \bar{\zeta} + \bar{K}\left(\ln\frac{A}{A_0} + \int_{t_0}^{t}k_e dt\right) + \bar{\eta}\left(v_{xx} + v_{yy} + k_e\right),$$

(89)

$$\sigma_{yy}^H = -2K^HQ_{xx}^H - \zeta_{xx}^H - P^H,$$

(90)

$$\sigma_{yy} = -2KQ_{xx} - \zeta_{xx} + \bar{\zeta} + \bar{K}\left(\ln\frac{A}{A_0} + \int_{t_0}^{t}k_e dt\right) + \bar{\eta}\left(v_{xx} + v_{yy} + k_e\right),$$

(91)

$$\sigma_{yy}^H = -k^H\left(h^H - h_0^H\right),$$

(92)

$$\sigma_{yy} = -k(h - h_0), \tag{93}$$

where $A = x_{HB}\,h$ is the blade area, the rate of cell extrusion $k_e$ in **Equation 89** is obtained from experimental measurements, $dx_{HB}/dt$ in **Equations 85, 86** can be obtained from **Equation 83**, and the hinge area $A_H$ in **Equation 87** is obtained from experimental measurements. In the system of equations above, **Equations 85, 86** correspond to **Equations 66, 67**, **Equation 87** arises from the definition of the hinge area in the rectangle model, $A_H = x_{HB}\,h^H$, **Equations 88–93** arise from the constitutive equations for the stress **Equations 55–57** in the hinge and blade, and **Equations 92, 93** correspond to the stress boundary conditions **Equations 71, 72**.

To solve the dynamic equations for the wing where the extracellular matrix was anteriorly ablated (ECM_AntCut), the same system of equations is solved, with the boundary condition **Equation 93** replaced by the boundary condition given by **Equation 73**. Additionally, we accounted for the fact that a smaller part of the hinge is visible in the ECM_AntCut wing compared to the WT wing. We added a correction term to the measured length of the ECM_AntCut wing such that its total length is the same as for the WT wing.

To solve the dynamic equations for the wing where the extracellular matrix was distally ablated (ECM_DistCut) and for *dumpy*$^{ov1}$ mutant wings, a new dynamical equation for the length of the wing $L$ is added, **Equation 74**

$$\gamma\frac{dL}{dt} = -h\sigma_{xx} - k_{PD}(L - x_{HB}). \tag{94}$$

For the distally ablated wing, all other equations are unchanged. For the *dumpy*$^{ov1}$ mutant wing, **Equation 93** is replaced by the boundary condition **Equation 73** and all other equations are unchanged.

From the strain rate in the blade $v_{ij}$ and in the hinge $v_{ij}^H$, we obtain the cumulative strain rate $u_{ij}$ and $u_{ij}^H$:

$$u_{ij} = \int_{t_0}^{t} dt'\, v_{ij}, \tag{95}$$

$$u_{ij}^H = \int_{t_0}^{t} dt'\, v_{ij}^H, \tag{96}$$

with $t_0$ taken as the first time point of experimental observation for each condition.

We then define the following objective function:

$$S\left(\frac{\bar{\zeta}}{K}, \frac{\bar{K}}{K}, \frac{\zeta_{xx}^H}{K}, \frac{K^H}{K}, \frac{k}{K}, \frac{k^H}{K}, \frac{k_{PD}^H}{K}, \frac{k_{PD}}{K}, \frac{\gamma}{K}, \frac{\bar{\eta}}{K}\right) =$$

$$\sum_{k\in\text{wings}}\sum_{n}\left[u_{xx}^{\exp,k}(t_n) - u_{xx}^{th,k}(t_n)\right]^2 + \left[u_{yy}^{\exp,k}(t_n) - u_{yy}^{th,k}(t_n)\right]^2 \tag{97}$$

$$+ \left[u_{xx}^{H,\exp,k}(t_n) - u_{xx}^{H,th,k}(t_n)\right]^2 + \left[u_{yy}^{H,\exp,k}(t_n) - u_{yy}^{H,th,k}(t_n)\right]^2,$$

where the fitted wings are WT #2, ECM_AntCut and ECM_DistCut, and $n$ labels successive time points. The parameter $\zeta_{xx}/K$ was obtained from fitting to circular laser cut experiments (see 'Materials and methods', Analysis of circular laser ablations). We found that the fitting procedure returned small values of the parameters $\zeta_{xx}^H/K$ and $K^H/K$ which were therefore set to zero. The fit function was minimised for the remaining 8 parameters, and the optimal parameters are reported in **Table 2**.

A separate fit was performed for Dumpy wings, with the following objective function:

$$S\left(\frac{k^{\mathrm{Dp}}}{K}, \frac{k^{\mathrm{H,Dp}}}{K}, \frac{k_{PD}^{\mathrm{H,Dp}}}{K}, \frac{k_{PD}^{\mathrm{Dp}}}{K}, \frac{\gamma^{\mathrm{Dp}}}{K}\right) =$$

$$\sum_n \left[u_{xx}^{\mathrm{exp,Dp}}(t_n) - u_{xx}^{\mathrm{th,Dp}}(t_n)\right]^2 + \left[u_{yy}^{\mathrm{exp,Dp}}(t_n) - u_{yy}^{\mathrm{th,Dp}}(t_n)\right]^2 \qquad (98)$$

$$+ \left[u_{xx}^{\mathrm{H,exp,Dp}}(t_n) - u_{xx}^{\mathrm{H,th,Dp}}(t_n)\right]^2 + \left[u_{yy}^{\mathrm{H,exp,Dp}}(t_n) - u_{yy}^{\mathrm{H,th,Dp}}(t_n)\right]^2,$$

where the tissue parameters $\bar{\zeta}/K$, $\bar{K}/K$ and $\bar{\eta}/K$ have the value obtained from WT fits, and $\zeta_{xx}^{\mathrm{H}}/K$ and $K^{\mathrm{H}}/K$ were set to zero as in WT fits. The parameter $\zeta_{xx}^{\mathrm{Dp}}/K$ was obtained from fitting to circular laser cut experiments (see 'Materials and methods', Analysis of circular laser ablations). The values of the remaining five fitted parameters are reported in **Table 2**.

Reported parameter uncertainties are obtained by evaluating the inverse Hessian associated to the objective functions defined in **Equations 97, 98** and estimating measurement errors from the value of the objective function at the minimum.

## Area dependence of $\lambda_{xx}$

We quantified the final cell elongation in unperturbed WT, *dumpy*ᵒᵛ¹ mutant and distally severed WT wings. In addition, we studied a temperature sensitive *cdc2* mutant in which cell divisions are inhibited. For all perturbations, the final cell elongation $Q_{xx}^f$ was positive, corresponding to elongation in the PD axis. The magnitude of $Q_{xx}^f$ varied suggesting that these perturbations alter the internal dynamics of the wing blade. Interestingly, we find that the final cell elongation $Q_{xx}^f$ is correlated with the final cell area for WT and all perturbed wings (**Figure 9—figure supplement 1**). According to **Equation 5** in the main text, the final cell elongation is given by $-\tau_r \lambda_{xx}$. This suggests that the magnitude of polarization-induced topological changes $\lambda_{xx}$ and/or the stress relaxation time $\tau_r$ increase with cell area. In the fit presented in the main text and in **Figure 10** and **Figure 9—figure supplement 2**, we accounted for this dependency by taking different values of $\lambda_{xx}$ for different conditions.

