## [Decision Letter]

Thank you for sending your work entitled “Interplay of cell dynamics and epithelial tension during morphogenesis of the *Drosophila* pupal wing” for consideration at *eLife*. Your article has been favorably evaluated by Naama Barkai (Senior editor), a Reviewing editor, and three reviewers.

The Reviewing editor and the reviewers discussed their comments before we reached this decision, and the Reviewing editor has assembled the following comments to help you prepare a revised submission.

In general all of the reviewers felt the body of work was very impressive, and brings important new information to the field. Using live imaging, genetic/physical perturbations, quantitative data analysis and modelling they explore morphogenesis of the *Drosophila* pupal wing and how tissue shape changes in the *Drosophila* wing are related to cell shapes and mechanics. It also reveals the importance of Dumpy-dependent attachment of the disk to the cuticle. The authors develop a method to extract cell-level contributions to tissue area changes and tissue shear, and demonstrate that the various drivers of tissue shape are changing with time in interesting and robust patterns. In addition, the authors develop a fairly simple continuum model for tissue shape changes that can fit all the data in WT, mutants, and ablated embryos. This work demonstrates that both external boundary conditions as well as internal force generation are necessary to generate the WT cell shape. The deformation of wing tissues in normal and dumpy mutants is decomposed into contributions from cell growth, rearrangement and mitosis. An effort is made to distinguish autonomous cell behaviours from responses to stresses that arise from hinge reshaping and pinning of the blade margins. The experiments have been done carefully and bring important new information to the field.

In their revised version of the manuscript the authors should address the following points raised by the reviewers.

1) Calling ṽ the change in the “shear” is not precise. Some people use “shear” to refer to “shear stress” instead of shear strain. Why not call it the shear strain?

2) Why is there no autonomous active pressure in [Disp-formula equ3], which is the analogue to the active shear stress in [Disp-formula equ4]?

3) Why does [Disp-formula equ52] in the supplemental include a viscous contribution to the pressure (and that's what the authors say they fit), but [Disp-formula equ3] includes no such term? It is not clear when the dissipative term is included and when it is not.

4) Given that several main results in this paper rely on the fact that there is a delay (*τ*_d_) between the cellular topological changes (*R*) and the cell elongation (*Q*), and that this delay seems to be about 4 hours, it would be extremely useful to have some discussion of what is generating that delay. Naively, I would expect that cell elongation would drive rearrangements instantaneously—changes to shape should immediately generate T1s in fairly generic geometries. I find the existence of *τ*_d_ to be incredibly surprising (although the data is convincing that it exists.) So this 4 hour timescale must be something having to do with a signaling cascade specific to biological systems? Are there any candidates for this?

5) Building upon this, it would be useful to provide a sentence or two (or a reference) for why the functional form of [Disp-formula equ5] is chosen. There are other ways to incorporate delays into partial differential equations—why is this the correct way to do it?

6) It was not clear why the authors choose to use shear to characterize deformation (subsection headed “A method to quantify cellular contributions to wing blade deformation”). The authors might consider that elongation in the proximal-distal (PD) direction might be a more natural and easily understood measure. Shear is notoriously difficult to interpret mechanically. What defines its positive sense? An examination of Mohr's circle or some other strain clarification tool highlights the challenges associated with its definition and interpretation, even for mechanicians, never mind biologists and others lacking that specific training. By describing the deformations in terms of elongations, these unnecessary complications might be avoided. For example, one of the reviewers struggles greatly with the authors defining shear in terms of “the negative change in average triangle elongation” (in the third paragraph of the aforementioned subsection and then seemingly contradicted in the fourth paragraph). If a triangle elongates in the PD direction, that motion could be just as easily assigned to positive or negative shear (a square whose top moves to the right or to the left, respectively) suggesting that the correspondence is arbitrary, and not meaningful. Modern continuum mechanics texts and a few biologically-motivated articles provide shear definitions that are mathematically-motivated and rigorous.

7) While using shear strain is okay (and is natural for a physics audience) it is not as natural for a solid mechanician. All the reviewers agree that the description of what is meant by “shear” is confusing at present in the manuscript, and could benefit from making more connections to the engineering literature. When using Hooke's law, it and other constitutive equations should be qualified with a statement that points out it is a simple and appropriate starting point, but that nonlinear elasticity may play a role.

8) The authors are missing references to other cell mechanics models, including those by Brodland et al. and comparison with previous cell-based computational models and constitutive equations that address how tissue deformation is related to mitosis, cell rearrangement and growth, and other factors. The paper has the potential to bring much understanding of tissue reshaping, and defining the deformations and their contributions in terms of elongation (normal strain) rather than shear (shear strain) would represent a tremendous improvement to its presentation, rigor and accessibility. It would also remove many of the hand-waving machinations that clutter an otherwise lovely story. It furthermore seems a shame that an article that could serve as a major reference to future researchers does not bring clarity and simplicity to the mechanical analysis and lead future investigators in that direction.

9) Wing area is determined by mass accumulation, tissue (cell) height and cell loss, not by other parameters. This should be made clear and discussed (in the subsection “Cellular contributions to wing area changes” and in the Discussion), i.e. there is probably no growth (Figure 3).

10) It is surprising to see how similar tissue tension is in the wild type and Dumpy mutant tissue, as measured by both the isotropic and anisotropic components of wound opening following a cut. This suggests that the tissue is able to re-establish the force balance when uncoupled from the cuticle.

This is surprising and should be discussed more fully in the text (subsection headed “Dumpy-dependent physical constraints at the margin maintain epithelial tension in EDH the wing”).

Is it correct that cell shape anisotropy is better aligned with tension anisotropy in the mutant as suggested by Figure 2 and Figure 2—figure supplement 3?

11) There are several striking findings that are not that fully explained in the text in the wild type, cell division is shut down by 22.5h APF. This is not the case in the mutant. Interestingly, this is exactly when the shift in the T1 cell contribution to shear occurs (Figure 8). Is it possible that this is related to DILP8 function? If not, what is it that happens at 22.5h?

12) The biggest behaviour change in the mutant is the increase in the rate of cell extrusion and in area compression. This implies that area compression and extrusion are mechanistically related (i.e. extrusion is driven by compression) and are subject to extrinsic regulation. If this is the case (i.e. if the rate of compression is proportional to rate of extrusion with a delay), this should be made clear. In Figure 3—figure supplement 1 and Figure 3—figure supplement 2 it appears that extrusion varies most between wildtype animals and perturbations.

Again, this suggests that extrusion is not subject to intrinsic control but is regulated by tissue mechanics, as previously suggested. Division appears largely insensitive to mechanics, despite what is stated in the text (in the last paragraph of the subsection headed “Cellular contributions to wing area changes”). Interestingly, extrusion does not contribute to shear. This may suggest that it is driven by isotropic forces, e.g. isotropic compression. Please discuss (e.g. in the Discussion: “This suggests that cells may measure epithelial tension to communicate with each other and reproducibly control tissue area despite variable contributions of cell division, cell extrusion and cell area).

13) The authors suggest that the precision of the wildtype tissue is greater than that of individual cellular processes. This is a very important point. While the data suggest that this may the case, it would be good to come up with a quantitative measure of precision and variation by which to test this idea with, i.e. compare cell number/cell area/division/extrusion precision/variation with that of tissue area and/or tissue shape precision/variation. Importantly, if there is no growth during the experiment, area precision may simply reflect the absence of cell mass accumulation. If this is the case, it is a trivial result.

---

## [Author Response]

*1) Calling ṽ the change in the “shear” is not precise. Some people use “shear” to refer to “shear stress” instead of shear strain. Why not call it the shear strain*?

Indeed we failed to precisely define what we mean by “shear” in the main text, and we thank the reviewers for pointing this out. We now clearly define “shear” in the revised manuscript as pure shear, corresponding to axial elongation at constant area, similar to the shape change associated with convergence-extension (please see the first paragraph of the subsection headed “A method to quantify cellular contributions to wing blade deformation”). All shear tensors in the manuscript describe this type of pure shear. We distinguish it from simple shear, which is a combination of pure shear and rotation. We have added new figure panels (Figure 4) to clarify the definition of shear and the decomposition of simple shear in shear and rotation.

*2) Why is there no autonomous active pressure in*
[Disp-formula equ3]*, which is the analogue to the active shear stress in*
[Disp-formula equ4]?

In our model, there is an autonomous active pressure, which is analogous to the active shear stress in [Disp-formula equ4]. In [Disp-formula equ3], this active pressure is hidden in the definition of the preferred cell area *a*_0_, which depends on active pressure. This is explained in Appendix 3.1.2 in [Disp-formula equ53 equ54 equ55]. We now also explain this in the main text (in the subsection headed “Relationship between tissue stress and cell elongation”).

*3) Why does*
[Disp-formula equ52]
*in the supplemental include a viscous contribution to the pressure (and that's what the authors say they fit), but*
[Disp-formula equ3]
*includes no such term? It is not clear when the dissipative term is included and when it is not*.

We first started from [Disp-formula equ3] given in the main text. By comparing our theory to the observed tissue flows, we found that we needed to add the contribution of tissue area viscosity η¯ to the tissue pressure (see Appendix 2.4, [Disp-formula equ52] and Appendix 3.1.2, [Disp-formula equ55]). We now explain this more clearly in the main text in the fourth paragraph of the subsection headed “Cell shape changes and tissue flows during pupal development can be understood by a continuum mechanical model”. The tissue area viscosity is included in the fits shown in Figure 10 and Figure 10—figure supplement 2 and the corresponding values are shown in Table 1.

For the sake of logical flow, we prefer to begin with the simple description of tissue pressure given in [Disp-formula equ3] and only introduce area viscosity after it is motivated by fits to the experimental data.

*4) Given that several main results in this paper rely on the fact that there is a delay (τ*_*d*_*) between the cellular topological changes (*R*) and the cell elongation (*Q*), and that this delay seems to be about 4 hours, it would be extremely useful to have some discussion of what is generating that delay. Naively, I would expect that cell elongation would drive rearrangements instantaneously—changes to shape should immediately generate T1s in fairly generic geometries. I find the existence of τ*_*d*_
*to be incredibly surprising (although the data is convincing that it exists.) So this 4 hour timescale must be something having to do with a signaling cascade specific to biological systems? Are there any candidates for this*?

We were as surprised as the reviewers to notice the existence of this delay. We are currently investigating this, both experimentally and theoretically. One possibility is based on the idea that T1 transitions are influenced by planar cell polarity (PCP) domains. As cells elongate, the Core PCP domains reorient along the PD axis on the time scale of a few hours (3; 42). One might imagine that these PCP domains contribute to the disassembly of cell boundaries required for PD-oriented T1 transitions. Thus, the delay could correspond to the time needed to reorient PCP domains. At the moment, we think that it is premature to include this speculation in the Discussion.

*5) Building upon this, it would be useful to provide a sentence or two (or a reference) for why the functional form of*
[Disp-formula equ5]
*is chosen. There are other ways to incorporate delays into partial differential equations*—*why is this the correct way to do it*?

As the reviewers noticed, there are different ways to incorporate delays in partial differential equations. [Disp-formula equ5] can be generalized in the form

R(t)= ∫−∞tG(t−t')Q(t')dt'+λ,

where *G*(*t*) is a linear response function. Different choices of *G*(*t*) correspond to different types of delays. The choice of [Disp-formula equ5] corresponds to the exponential linear response function

G(t)=1τrτde−tτd .

This exponential decay implies that the current value of cell elongation has the strongest impact on shear due to topological changes and the effect of recent values of cell elongation fades exponentially over time, disappearing beyond the time *τ*_*d*_. This choice corresponds to an internal process that relaxes during the time *τ*_*d*_ (such as PCP reorientation).

For comparison, we could also consider a different type of delay:

R(t)= 1τrQ(t− τd)+λ,

which is equivalent to the linear response function

G(t)= 1τrδ(t− τd),

where *δ* denotes the Dirac delta function. Such a description would be applicable if information about cell elongation is processed during the exact time *τ*_*d*_ before it can affect T1 transitions or shear. This choice of fixed delay is more complex and seems less realistic. For a complex biological process, the general linear response function is expected to consist of many exponential relaxation contributions. Our choice of [Disp-formula equ5] is equivalent to the simplification of only keeping the slowest relaxation process that dominates over longer times.

We now briefly describe why we chose this implementation of the time delay in the main text (in the third paragraph of the subsection headed “Model for the dynamics and the orientation of topological changes”).

*6) It was not clear why the authors choose to use shear to characterize deformation (subsection headed “A method to quantify cellular contributions to wing blade deformation”). The authors might consider that elongation in the proximal-distal (PD) direction might be a more natural and easily understood measure*.

This problem probably arises from the fact that we did not clearly define what we meant by shear as discussed in point 1 above. We do in fact use the elongation in the PD direction (which is pure shear) as a measure for shear. Note also that the definition of the deformation requires a reference state. In the dynamic tissue the reference state changes dynamically as cells rearrange. Therefore the relevant measure that we quantify is the rate of deformation, which is the rate of pure shear along the PD axis. This is now described more clearly in the first paragraph of the subsection headed “A method to quantify cellular contributions to wing blade deformation”.

*Shear is notoriously difficult to interpret mechanically. What defines its positive sense*?

This arises from the same misunderstanding. We define positive increase in pure shear as increase of elongation along the PD axis.

*An examination of Mohr's circle or some other strain clarification tool highlights the challenges associated with its definition and interpretation, even for mechanicians, never mind biologists and others lacking that specific training. By describing the deformations in terms of elongations, these unnecessary complications might be avoided. For example, one of the reviewers struggles greatly with the authors defining shear in terms of “the negative change in average triangle elongation” (in the third paragraph of the aforementioned subsection and then seemingly contradicted in the fourth paragraph). If a triangle elongates in the PD direction, that motion could be just as easily assigned to positive or negative shear (a square whose top moves to the right or to the left, respectively) suggesting that the correspondence is arbitrary, and not meaningful. Modern continuum mechanics texts and a few biologically-motivated articles provide shear definitions that are mathematically-motivated and rigorous*.

In our original manuscript we did not clearly state that the pure shear of a single triangle corresponds to the positive change in triangle elongation. In the revised manuscript we now say this explicitly (please see the subsection entitled “A method to quantify cellular contributions to wing blade deformation”).

The issue of the “negative change in average triangle elongation” arises because we discuss the contribution to tissue shear by topological changes. During a topological transition, the triangulation changes and thus the average triangle elongation changes by a finite amount *Δ****Q***. However, at the moment the topological change occurs there is no tissue shear. Therefore, tissue shear and triangle elongation are no longer the same. This can be compensated by introducing a contribution to tissue shear by topological transitions, *Δ****T***, given by:

0=ΔQ+ΔT.

That is why the shear by a topological change is given by the negative of the elongation change during a topological change (see Appendix 1.3, [Disp-formula equ22]). We have revised the text to clarify this point (subsection “A method to quantify cellular contributions to wing blade deformation”).

The original manuscript referred to a completely different contribution to the tissue shear, the correlation term ***D***. To clarify the discussion of this term we rewrote the corresponding paragraph.

*7) While using shear strain is okay (and is natural for a physics audience) it is not as natural for a solid mechanician. All the reviewers agree that the description of what is meant by “shear” is confusing at present in the manuscript, and could benefit from making more connections to the engineering literature. When using Hooke's law, it and other constitutive equations should be qualified with a statement that points out it is a simple and appropriate starting point, but that nonlinear elasticity may play a role*.

See reply to point one.

*8) The authors are missing references to other cell mechanics models, including those by Brodland et al. and comparison with previous cell-based computational models and constitutive equations that address how tissue deformation is related to mitosis, cell rearrangement and growth, and other factors*.

We have now added the relevant references.

*The paper has the potential to bring much understanding of tissue reshaping, and defining the deformations and their contributions in terms of elongation (normal strain) rather than shear (shear strain) would represent a tremendous improvement to its presentation, rigor and accessibility. It would also remove many of the hand-waving machinations that clutter an otherwise lovely story. It furthermore seems a shame that an article that could serve as a major reference to future researchers does not bring clarity and simplicity to the mechanical analysis and lead future investigators in that direction*.

We hope that our improved explanation of what we mean by shear has now resolved this problem.

*9) Wing area is determined by mass accumulation, tissue (cell) height and cell loss, not by other parameters. This should be made clear and discussed (in the subsection “Cellular contributions to wing area changes” and in the Discussion), i.e. there is probably no growth (*Figure 3*)*.

We are talking about tissue area change instead of actual tissue growth. To make this more clear, we changed each occurrence of the term tissue growth into the term tissue area expansion.

*10) It is surprising to see how similar tissue tension is in the wild type and Dumpy mutant tissue, as measured by both the isotropic and anisotropic components of wound opening following a cut. This suggests that the tissue is able to re-establish the force balance when uncoupled from the cuticle*.

*This is surprising and should be discussed more fully in the text (subsection “Dumpy-dependent physical constraints at the margin maintain epithelial tension in EDH the wing”)*.

*Is it correct that cell shape anisotropy is better aligned with tension anisotropy in the mutant as suggested by*
Figure 2
*and*
Figure 2—figure supplement 3?

In fact, anisotropic tension is significantly different in WT and dumpy wing blades. It is difficult to appreciate this by looking at Figure 2. However, it is more clear in Figure 9, where the same data are shown in a different format. Figure 9 and the new Figure 9—figure supplement 1 shows that dumpy cells are significantly less elongated than WT cells. The slope of the lines in Figure 9 also shows that their shear elastic modulus is the same as that of WT cells. Therefore, anisotropic stress is less in dumpy mutants. We have added a panel (Figure 9—figure supplement 1) comparing the PD component of average cell elongation in dumpy and WT wing blades and have discussed this issue more explicitly in the main text (subsection headed “Relationship between tissue stress and cell elongation”).

From the behavior of the isotropic component of the wound opening (Figure 9) it is difficult to judge whether isotropic tissue tension is reduced in dumpy mutants. As explained in the main text the data is too noisy to extract material properties possibly because average cell area varies little while individual cell area varies strongly. However, our model calculations predict that this is indeed the case.

*11) There are several striking findings that are not that fully explained in the text in the wild type, cell division is shut down by 22.5h APF. This is not the case in the mutant. Interestingly, this is exactly when the shift in the T1 cell contribution to shear occurs (*Figure 8*). Is it possible that this is related to DILP8 function? If not, what is it that happens at 22.5h*?

We had limited our discussion of these cell divisions to what we knew for certain: that they are not initiated by the development of anisotropic stresses in the wing epithelium—instead, there appears to be some element of autonomous control. But of course what initiates these divisions and what makes them stop are extremely interesting questions and we have expanded the Discussion in the manuscript to address the issue in more depth. Below, we outline our current thinking about these divisions, and discuss the reviewers’ proposal that Dilp signaling and T1 transitions may regulate them.

Although pupal cell divisions start at the same time as hinge contraction, it is clear that they are not initiated by the resulting epithelial stresses because releasing such stresses prolongs rather than prevents them. We agree there could be some sort of hormonal initiating signal. While it is possible that Dilp6 could contribute, pupal cell divisions (unlike Dilp6) don’t appear to increase the size of the wing; they are almost perfectly balanced by a decrease in cell area. (Dilp8 was probably a typo, since Dilp6, not Dilp8, increases body size in post-feeding animals). The burst of ecdysone that occurs at the prepupal to pupal transition might be a better candidate to initiate these divisions.

However, we don’t think that a single external signal, like Dilp or ecdysone, can completely explain the number or pattern of pupal cell divisions (though such a signal could help initiate them), because divisions can be prolonged by wing autonomous perturbations like laser ablation. Also (although we have not included this in the paper) pupal divisions occur in specific vein and intervein regions at reproducibly different times (consistent with Garcia-Bellido’s studies in the 90’s). This is hard to reconcile with control by a single hormonal signal. It suggests that local signals must also play a role, possibly signals associated with veins.

The reviewers note that cell divisions stop near the time that T1 transitions reorient. This is an interesting correlation, and seems to hold true for the perturbed wings as well. Since cell divisions do not appear to be promoted by epithelial stresses, it is unlikely that these T1’s inhibit cell divisions by relieving stress in the epithelium. But it is certainly possible that stress-induced T1 transitions inhibit cell division by some other mechanism. Probing a possible causal relationship between T1 transitions and cell divisions, and its directionality, will require us to analyze wings where each process is specifically blocked. We are pursuing this goal, but it is outside the current scope of the manuscript.

Another possibility is that the prolongation of cell division in Dumpy mutants, or upon laser ablation constitutes a wound healing response that occurs when stress is reduced. Scratching MDCK cell monolayers (thus releasing stress in the epithelium) causes a wave of proliferation that contributes to wound healing. In fact, we have noticed that pupal wing epithelial cells robustly heal wounds caused by laser ablation—the wing must be completely severed both dorsally and ventrally to prevent the resealing of the epithelium.

*12) The biggest behaviour change in the mutant is the increase in the rate of cell extrusion and in area compression. This implies that area compression and extrusion are mechanistically related (i.e. extrusion is driven by compression) and are subject to extrinsic regulation. If this is the case (i.e. if the rate of compression is proportional to rate of extrusion with a delay), this should be made clear. In*
Figure 3—figure supplement 1
*and*
Figure 3—figure supplement 2
*it appears that extrusion varies most between wildtype animals and perturbations*.

*Again, this suggests that extrusion is not subject to intrinsic control but is regulated by tissue mechanics, as previously suggested. Division appears largely insensitive to mechanics, despite what is stated in the text (in the last paragraph of the subsection headed “Cellular contributions to wing area changes”). Interestingly, extrusion does not contribute to shear. This may suggest that it is driven by isotropic forces, e.g. isotropic compression. Please discuss (e.g. in the Discussion: “This suggests that cells may measure epithelial tension to communicate with each other and reproducibly control tissue area despite variable contributions of cell division, cell extrusion and cell area)*.

Our data indeed suggest that extrusions depend on isotropic stress. We have begun to analyze this quantitatively as suggested by the reviewer. However, we find that the details of the relationship are not straightforward and will require more analysis. We think that this point is outside the scope of the present paper. We have now rewritten the corresponding parts to make the qualitative relationship more clear and we now cite previous work consistent with this idea ([38]; in the Results).

*13) The authors suggest that the precision of the wildtype tissue is greater than that of individual cellular processes. This is a very important point. While the data suggest that this may the case, it would be good to come up with a quantitative measure of precision and variation by which to test this idea with, i.e. compare cell number/cell area/division/extrusion precision/variation with that of tissue area and/or tissue shape precision/variation*.

In the previous manuscript, we used the standard deviation as a quantitative measure for the variation of cellular contributions to shear and area change. These standard deviations can be seen in Figures 3 and 6 as shaded regions surrounding the different curves. Now, to determine whether different cellular contributions influence each other, we have calculated the variances (square of the standard deviation) of the cumulative cellular contributions to area change and shear at the final time point. We then compared the sum of these variances to the variance of overall relative area change and overall shear deformation at the final time point. If cellular contributions were completely independent of each other, the sum of their variances would equal the variance of the sum and their ratio would be 1. However, we find that this is not the case but the ratio is about 25 and 20 for shear and area change respectively.

*Importantly, if there is no growth during the experiment, area precision may simply reflect the absence of cell mass accumulation. If this is the case, it is a trivial result*.

The fact that releasing connections to the apical extracellular matrix reduces the area of the wing suggests that area homeostasis is not a trivial consequence of lack of growth. Note also that in the two-dimensional arrangement of cells, area can change in the absence of growth if the height changes or if cells are extruded.